# POMP: A Theoretical Approach to Mitigate Forgetting in Finetuning Multi-Modal Models

## Abstract

Catastrophic forgetting is a major challenge when adapting pretrained models to new tasks in multi-modal contrastive learning (MMCL). We provide a theoretical analysis of finetuning by introducing a *contrastive target matrix* that reformulates the linearized contrastive objective as a matrix least-squares problem. This formulation yields closed-form solutions for direct finetuning, weight-space regularization, and self-distillation, providing a geometric interpretation of how each strategy manages pretrained knowledge. Our analysis reveals that self-distillation preserves knowledge in the subspace orthogonal to the finetuning data while forming a convex combination of the pretrained and new solutions within the task subspace. We extend this analysis to a dynamic self-distillation framework with a weighted moving average (WMA) teacher. We prove that, unlike standard Exponential Moving Average (EMA) teachers which eventually collapse onto the student, the WMA teacher maintains a persistent, non-vanishing regularizing force throughout training by integrating the full optimization trajectory. These theoretical insights motivate our method, **POMP** (Preserve-Orthogonal-Mix-Parallel), which operationalizes this framework. POMP uses a composite distillation loss guided by the WMA teacher to achieve state-of-the-art out-of-distribution robustness and calibration when finetuning CLIP.

## 1 Introduction

Pretrained models such as CLIP (Radford et al., 2021) have revolutionized machine learning through their remarkable zero-shot transfer and adaptive capabilities. These models derive their robustness from large-scale multi-modal pretraining (Fang et al., 2022; Xu et al., 2024b), enabling diverse applications from visual recognition (Shen et al., 2022b; Zhang et al., 2022b) to generative modeling (Betker et al., 2023; Pi et al., 2024) and serving as backbones for large multimodal models (Alayrac et al., 2022; Liu et al., 2023; Zhu et al., 2024).

Despite these successes, adapting these pretrained models to downstream tasks via finetuning presents a fundamental challenge: while finetuning improves in-distribution (ID) performance, it often degrades out-of-distribution (OOD) robustness (Radford et al., 2021). This trade-off manifests as catastrophic forgetting of pretrained knowledge (Wortsman et al., 2022b), where models sacrifice their general-purpose representations to optimize for task-specific patterns, potentially overfitting to spurious correlations in the finetuning data.

Several empirical strategies have emerged to mitigate this trade-off. For example, LP-FT (Kumar et al., 2022) addresses the problem of randomly initialized heads distorting pretrained features by first learning a linear probe on frozen features before full finetuning. FLYP (Goyal et al., 2023) extends this idea by reusing CLIP's pretrained text encoder as the classification head, maintaining consistency with the pretraining objective. Post-hoc methods like WiSE-FT (Wortsman et al., 2022b) and Model Stock (Jang et al., 2024) perform weight averaging between pretrained and finetuned models to recover lost robustness. Regularization-based approaches including $L_2$–SP (Li et al., 2018) and self-distillation with dynamic teachers (Oh et al., 2024) introduce constraints to preserve pretrained knowledge. However, these dynamic teacher methods typically rely on an Exponential Moving Average (EMA), whose regularizing influence we prove diminishes as the teacher converges to the student, creating an opportunity for a more robust approach.

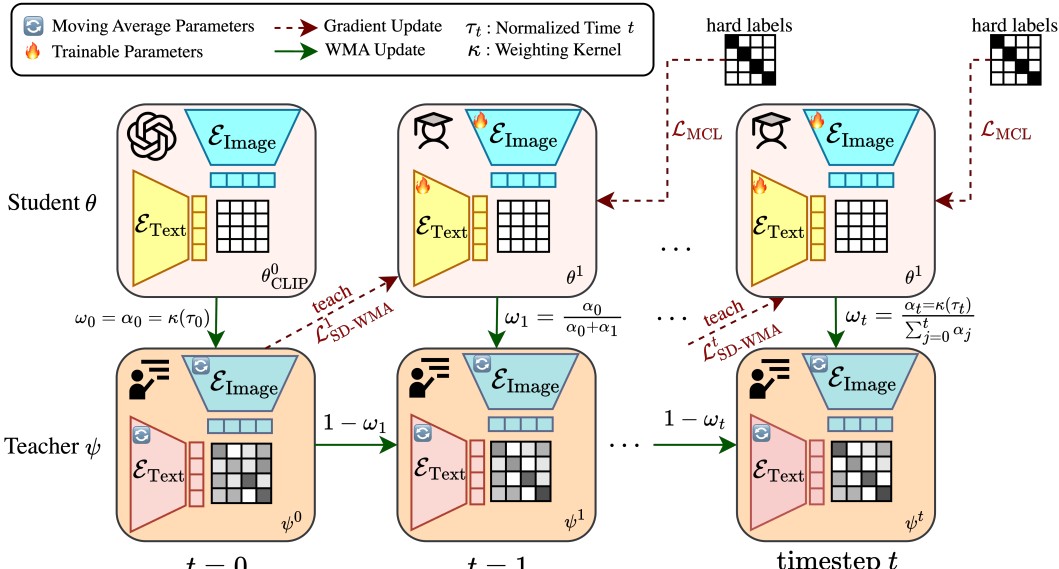

Figure 1: **Overview of POMP.** The base contrastive objective is combined with a dynamic self-distillation loss from a Weighted Moving Average (WMA) teacher to preserve orthogonal pretrained knowledge while adaptively mixing within the task subspace.

Moreover, despite the proliferation of these methods, a theoretical understanding of *what* changes during contrastive finetuning and *where* forgetting occurs remains elusive. We address this gap by developing a theoretical framework that reveals the geometric structure of how different finetuning strategies modify pretrained representations. We find that the linearized contrastive finetuning objective can be reformulated as a matrix least-squares problem through what we call the *contrastive target matrix*. This reformulation enables closed-form solutions for common finetuning strategies, exposing their fundamentally different geometric behaviors.

Building on these geometric insights, we extend our analysis to dynamic self-distillation with a weighted moving average (WMA) teacher (see Figure 1). We prove that this adaptive anchor achieves bias-free convergence to the task-optimal solution within the finetuning subspace, i.e., eliminating the persistent anchor bias of static methods, while maintaining preservation in orthogonal directions.

Our theoretical insights lead to the design of **POMP** (Preserve-Orthogonal-Mix-Parallel), a practical finetuning method that implements our geometric principles. As shown in Figure 1, POMP combines contrastive learning with dynamic self-distillation, achieving state-of-the-art results on ImageNet and its distribution shifts. Across multiple CLIP architectures, POMP consistently improves the ID-OOD trade-off, demonstrating that our theoretical framework translates into tangible empirical gains.

In summary, our work makes the following main contributions: **(i)** We introduce the contrastive target matrix formulation that reduces multi-modal contrastive finetuning to a tractable least-squares problem, yielding closed-form solutions for common finetuning strategies; **(ii)** We provide a geometric decomposition that characterizes where different methods preserve or modify pretrained knowledge, explaining their distinct forgetting behaviors and motivating the need for a dynamic teacher; **(iii)** We analyze dynamic self-distillation with a weighted moving-average teacher. We prove it overcomes a key limitation of EMA-based teachers by inducing a non-vanishing regularizing gradient that prevents late-stage overfitting. We further prove this achieves bias-free task convergence while maintaining orthogonal preservation, and we demonstrate its effectiveness through the POMP method.

## 2 RELATED WORK

Contrastive language–image pretraining (Radford et al., 2021; Jia et al., 2021; Ilharco et al., 2021; Zhai et al., 2023) enables strong zero-shot transfer but naive finetuning can harm OOD robustness (Taori et al., 2020; Wortsman et al., 2022b). Robust finetuning explores weight interpolation/averaging (Wortsman et al., 2022b;a; Jang et al., 2024), weight- or output-space regularization (Li

et al., 2018; Li and Hoiem, 2018), and contrastive variants aligned to text prompts or energies (Goyal et al., 2023; Mao et al., 2024; Nam et al., 2024; Shu et al., 2023). CaRot (Oh et al., 2024) couples contrastive training with new regularizers to jointly improve OOD accuracy and calibration.

Self-distillation and dynamic teachers stabilize learning and preserve knowledge (Hinton et al., 2015; Zhang et al., 2019; Mobahi et al., 2020; Laine and Aila, 2017; Tarvainen and Valpola, 2017). Momentum/EMA teachers are effective yet can introduce persistent bias toward initialization. Our *WMA* teacher generalizes EMA by weighting the entire trajectory on normalized time, enabling endpoint-aware curricula (e.g., arcsine/Beta kernels) and, as we prove, bias-free task-subspace convergence. Our theory complements linearized analyses of contrastive learning (Ji et al., 2023; Tian, 2022; Nakada et al., 2023; Xue et al., 2024) and explains forgetting via an explicit geometric decomposition. An extended literature review appears in §A.

# 3 FINETUNING LOSS REFORMULATION AND CORRESPONDING ANALYSIS

Finetuning multi-modal foundation models like CLIP to new tasks often leads to a fundamental trade-off: improved in-distribution (ID) performance at the expense of degraded out-of-distribution (OOD) robustness, a phenomenon known as catastrophic forgetting. To address this critical challenge, we develop a theoretical framework that sheds light on the underlying dynamics of finetuning.

## 3.1 FINETUNING LOSS REFORMULATION

We consider finetuning with paired image-text data $\{(\mathbf{x}_I^i, \mathbf{x}_T^i)\}_{i=1}^n$. Following linearized analyses (e.g., Ji et al. (2023); Tian (2022); Nakada et al. (2023)), image and text encoders are linear projections, $g_I(\mathbf{x}) = \mathbf{W}_I \mathbf{x}$ and $g_T(\mathbf{x}) = \mathbf{W}_T \mathbf{x}$. We analyze finetuning where the image encoder $\mathbf{W}_I$ is adapted from a pretrained state $\mathbf{W}_I^0$, while the text encoder $\mathbf{W}_T$ is frozen to its pretrained state $\mathbf{W}_T^0$. For a batch of $n$ image features $\mathbf{X}_I \in \mathbb{R}^{d_I \times n}$ (where $d_I$ is the feature dimension), the linearized multi-modal contrastive learning (MMCL) objective can be rewritten using a novel construct:

**Definition 3.1** (Contrastive Target Matrix). Given $\mathbf{W}_T^0$ and finetuning texts $\mathbf{X}_T$, we define the *contrastive target matrix* as $\mathbf{Y}_{\text{FT}} = \mathbf{W}_T^0 \mathbf{X}_T (n\mathbf{I}_n - \mathbf{J}_n) \in \mathbb{R}^{p \times n}$. Each column $\mathbf{y}_i$ is constructed to attract image $\mathbf{x}_I^i$ towards its paired text $\mathbf{x}_T^i$ and repel it from other texts in the batch (detailed in §C.1).

This definition allows us to reformulate the linearized contrastive finetuning objective as the following:

$$\min_{\mathbf{W}_I} \frac{1}{2} \|\mathbf{W}_I \mathbf{X}_I - \mathbf{Y}_{\text{FT}}\|_{\text{F}}^2. \tag{1}$$

This formulation is crucial as it enables closed-form solutions for various finetuning strategies under gradient descent, offering insights into their behavior (see §C.3 for a full derivation and proofs). Using this reformulation, we analyze how different finetuning strategies mitigate forgetting by preserving or adapting pretrained knowledge, and revealing the geometric structure of updates.

## 3.2 UNIFIED CLOSED-FORM SOLUTIONS

This subsection provides closed-form solutions for various finetuning strategies following our reformulation (Equation 1), revealing a *geometric decomposition*: finetuning involves (i) preserving pretrained knowledge in directions *orthogonal* to the finetuning data, and (ii) adapting or mixing knowledge *within* the task-relevant subspace. We first present the closed-form solutions below.

**Theorem 3.2** (Unified Framework for Contrastive Finetuning Solutions). *Let* $\mathcal{P}_I := \mathbf{X}_I (\mathbf{X}_I^\top \mathbf{X}_I)^+ \mathbf{X}_I^\top$ *be the orthogonal projector onto* $\text{range}(\mathbf{X}_I)$. *Gradient descent initialized at* $\mathbf{W}_I^0$ *on the objective* $\mathcal{L}(\mathbf{W}_I) = \frac{1}{2} \|\mathbf{W}_I \mathbf{X}_I - \mathbf{Y}_{FT}\|_F^2 + \mathcal{R}(\mathbf{W}_I)$ *converges to the following solutions:*

| *Strategy* | $\mathcal{R}(\mathbf{W}_I)$ | *Solution* |
|---|---|---|
| *Direct Finetuning* | $0$ | $\mathbf{W}_{FT} = \mathbf{W}_I^0(\mathbf{I} - \mathcal{P}_I) + \mathbf{Y}_{FT}\mathbf{X}_I^\top(\mathbf{X}_I\mathbf{X}_I^\top)^+$ |
| *$L_2$ Regularization (L2-SP (Li et al., 2018))* | $\frac{\lambda}{2}\|\mathbf{W}_I - \mathbf{W}_I^0\|_F^2$ | $\mathbf{W}_{L_2} = (\mathbf{Y}_{FT}\mathbf{X}_I^\top + \lambda\mathbf{W}_I^0)(\mathbf{X}_I\mathbf{X}_I^\top + \lambda\mathbf{I})^{-1}$ |
| *Static Self-Distillation (SD (Furlanello et al., 2018))* | $\frac{\lambda}{2}\|\mathbf{W}_I\mathbf{X}_I - \mathbf{W}_I^0\mathbf{X}_I\|_F^2$ | $\mathbf{W}_{SD} = \mathbf{W}_I^0(\mathbf{I} - \frac{1}{1+\lambda}\mathcal{P}_I) + \frac{1}{1+\lambda}\mathbf{Y}_{FT}\mathbf{X}_I^\top(\mathbf{X}_I\mathbf{X}_I^\top)^+$ |

*Here,* $^+$ *denotes the Moore-Penrose pseudoinverse and* $\lambda > 0$ *is the regularization parameter.*

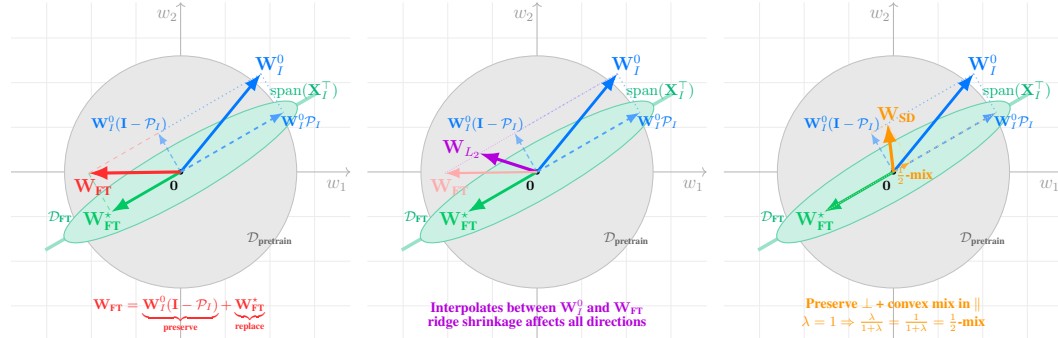

(a) **Direct FT:** Preserves orthogonal, replaces parallel component. (b) **L2-SP:** Blends all directions, no structure preservation. (c) **SD:** Preserves orthogonal, mixes parallel components.

Figure 2: **Geometric interpretation of finetuning strategies in 2D weight space.** The green line represents $\text{span}(\mathbf{X}_I^\top)$, the subspace where finetuning data concentrates. Starting from pretrained weights $\mathbf{W}_I^0$ (blue), each method combines the orthogonal component $\mathbf{W}_I^0(\mathbf{I} - \mathcal{P}_I)$ and the new task solution $\mathbf{W}_{\text{FT}}^\star = \mathbf{Y}_{\text{FT}}\mathbf{X}_I^\top(\mathbf{X}_I\mathbf{X}_I^\top)^+$ (green) differently: **(a)** Direct FT preserves the orthogonal component and replaces the parallel component entirely; **(b)** L2-SP creates a global blend without clean structural decomposition; **(c)** Static Self-Distillation preserves the orthogonal component and forms a convex combination of the parallel components (shown with $\lambda = 1$ giving equal weighting).

**Geometric Interpretation:** As visualized in Figure 2, Direct Finetuning discards pretrained knowledge within the finetuning data subspace, replacing it with the new task solution, while preserving orthogonal components. $L_2$ regularization shrinks the entire solution towards the pretrained weights, leading to a complex, non-surgical blend. **Self-Distillation achieves a nuanced trade-off**: it preserves pretrained knowledge in the subspace orthogonal to the finetuning data ($\mathbf{W}_I^0(\mathbf{I} - \mathcal{P}_I)$), and within the task-relevant subspace, it computes a convex combination of the projected pretrained weights and the optimal solution for the new task. This enables control over knowledge retention and adaptation (further details in §C.4).

### 3.3 DYNAMIC SELF-DISTILLATION WITH WEIGHTED MOVING AVERAGE (WMA) TEACHER

Although static SD overcomes the problems of direct FT and $L_2$-SP (see Fig. 2), it can introduce bias because of using a fixed anchor ($\mathbf{W}_I^0$). We resolve this by employing a *dynamic* teacher (Fig. 1) that evolves as a WMA of the student's trajectory. This enables bias-free convergence in the task subspace while maintaining orthogonal preservation. We first give the definition of WMA teacher below.

**Definition 3.3** (WMA Teacher). The WMA teacher $\mathbf{W}_{\text{Teacher}}^t$ averages student states $\mathbf{W}_I^k$ over time $k = 0, \ldots, t$, weighted by a kernel $\kappa(\tau_k)$ on normalized time $\tau_k = \frac{k+c_1}{T+c_2}$. The online recursion is:

$$\omega_t = \frac{\kappa(\tau_t)}{\sum_{j=0}^t \kappa(\tau_j)}, \qquad \mathbf{W}_{\text{Teacher}}^t = (1 - \omega_t)\,\mathbf{W}_{\text{Teacher}}^{t-1} + \omega_t\,\mathbf{W}_I^t, \qquad \mathbf{W}_{\text{Teacher}}^0 = \mathbf{W}_I^0. \quad (2)$$

**Persistent Regularization and Bias-Free Convergence:** Unlike an Exponential Moving Average (EMA) teacher, whose regularizing influence vanishes as it converges to the student, the WMA teacher (especially with a U-shaped kernel like Beta(0.5,0.5)) maintains a persistent regularizing force (see §C.5.2). This force continuously pulls the student towards its robust pretrained initialization. We prove that this adaptive anchoring achieves bias-free convergence to the task-optimal solution within the finetuning subspace:

**Theorem 3.4** (Bias-Free Convergence in the Task Subspace). *Let $\mathbf{W}_{FT}^\star$ be the optimal direct finetuning solution. The WMA teacher's projection onto the data subspace converges to $\mathbf{W}_{FT}^\star\mathcal{P}_I$, and consequently, the student's projection also converges to $\mathbf{W}_{FT}^\star\mathcal{P}_I$.*

The above theorem shows that using dynamic teachers can help eliminate the persistent anchor bias of static methods while preserving orthogonal knowledge (formal proof in §C.5).

## 4 PROPOSED APPROACH: PRESERVE-ORTHOGONAL-MIX-PARALLEL (POMP)

Guided by these geometric insights and the proven benefits of a WMA teacher in the above section, we propose **POMP** (Preserve-Orthogonal-Mix-Parallel), a novel finetuning method for multi-modal models, in this section. POMP combines the standard symmetric InfoNCE loss with dynamic self-distillation guided by a WMA teacher, as illustrated in Figure 1.

The total training objective for POMP is:

$$\mathcal{L}_{\text{POMP}} = \mathcal{L}_{\text{MMCL}} + \lambda_{\text{SD}} \, \mathcal{L}_{\text{SD-WMA}}. \tag{3}$$

**Multi-Modal Contrastive Loss ($\mathcal{L}_{\text{MMCL}}$):**  This is the primary finetuning loss, typically a symmetric InfoNCE objective. In our implementation, we also include a cross-Frobenius regularizer to prevent embedding collapse (standard CLIP finetuning recipe). This component drives the student model to learn new task-specific alignments.

**Dynamic Self-Distillation Loss ($\mathcal{L}_{\text{SD-WMA}}$):**  This is the core mechanism for robust knowledge preservation and adaptive mixing. It ensures the student retains generalizable features by learning from an evolving teacher model. As detailed in §C.6, $\mathcal{L}_{\text{SD-WMA}}$ is a composite distillation loss that includes several perspectives: **(i) Feature Distillation (FD):** Directly aligns student and teacher embeddings. **(ii) Contrastive Relational Distillation (CRD):** Matches batch-wise similarity distributions between student and teacher. **(iii) Interactive Contrastive Learning (ICL):** Encourages student-teacher cross-modal alignment. **(iv) Cross Knowledge Distillation (Cross-KD):** Aligns cross-modal logits to transfer relational structure. This multi-perspective approach operationalizes the theoretical insight of preserving distinct aspects of pretrained knowledge.

**Weighted Moving Average (WMA) Teacher:**  The teacher model is a central component of POMP. Unlike an EMA teacher, which gradually collapses onto the student, our WMA teacher is a weighted average of the *entire* student trajectory up to time $t$, using a carefully chosen weighting kernel (e.g., a Beta kernel with $\beta_1 = \beta_2 = 0.5$ as shown in Figure 1 and detailed in §C.5.1). This ensures that the initial robust pretrained knowledge always contributes to the teacher with a non-vanishing weight. This persistent regularization provides a continuous restoring force, preventing the student from over-specializing on spurious correlations in the finetuning data, and leading to bias-free convergence in the task subspace while maintaining robust orthogonal knowledge.

In Figure 1, $\theta^0_{\text{CLIP}}$ represents the initial pretrained CLIP model. $\theta^t$ denotes the student model at time $t$, with its image and text encoder ($\mathcal{E}_{\text{Image}}$ and $\mathcal{E}_{\text{Text}}$) being trained (marked by fire). The student receives gradient updates (red dashed arrows) from $\mathcal{L}_{\text{MMCL}}$. The WMA teacher model $\psi^t$ (with its parameters indicated by refresh symbol) is updated from the student's parameters (green solid arrows). The teacher then provides a teaching signal $\mathcal{L}_{\text{SD-WMA}}$ (red dashed arrow labeled 'teach') to regularize the student. This interplay allows POMP to adapt to new tasks while preserving pretrained knowledge.

## 5 EMPIRICAL ANALYSIS

This section evaluates POMP against strong baselines on ImageNet and natural distribution shifts, and includes a controlled toy study to validate theoretical predictions. We first describe our experimental setup, then define the evaluation metrics, followed by the main results and ablations. We conduct comprehensive ablations across distillation components, distillation strength ($\lambda_{\text{SD}}$), teacher update frequency, and teacher Beta-kernel shape; extended protocols and notes are provided in §B, with related loss definitions in §C.6 and teacher details in §C.5.1.

### 5.1 SYNTHETIC EMPIRICAL VALIDATION

To validate our theory (§3), we design a controlled toy experiment demonstrating catastrophic forgetting and its mitigation in a realistic spurious correlation setting (Arjovsky et al., 2019). The behaviors of Direct Finetuning, L2 Regularization, and Self-Distillation in a non-linear architecture align with our closed-form predictions and geometric interpretation.

### 5.1.1 EXPERIMENTAL SETUP

Our experiment consists of a pretraining phase to learn a general task, followed by a finetuning phase on a related but distinct task designed to induce forgetting.

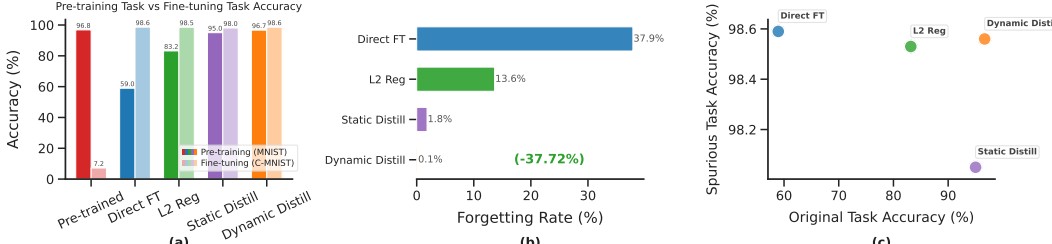

Figure 3: **Toy Experiment.** The experiment compares a pretrained model against four finetuning methods on a finetuning task. **(a)** Performance on the original MNIST and new Colored MNIST (C-MNIST) task. All finetuning methods successfully learn the new task. Direct FT and L2 Reg suffer severe performance degradation (catastrophic forgetting). **(b)** Catastrophic forgetting rate, quantified as the percentage drop in accuracy on the original task. Self-Distillation methods are more effective at preserving knowledge. **(c)** The performance trade-off between the original task ($x$-axis) and the spurious task ($y$-axis). Distillation methods achieve a much better trade-off, retaining high original task accuracy while mastering the new task.

**Datasets.** We use two variants of the MNIST dataset (LeCun et al., 1998; Deng, 2012). (i) **Original Pretraining Task:** We create a multi-modal version of MNIST, where each grayscale digit image is paired with a simple text description (e.g., an image of a '7' is paired with the text "the digit 7"). The model is pretrained on this dataset to learn robust, general-purpose representations for digit recognition. (ii) **Finetuning Task:** We create a dataset to introduce a spurious correlation. Images of digits 0-4 are colored red with 95% probability, while digits 5-9 are colored blue with 95% probability. This setup forces the model during finetuning to learn an easy-to-exploit but non-causal feature (color) to solve the new task, creating a direct conflict with the original digit recognition knowledge.

**Model Architecture.** We employ lightweight, non-linear models to show that our theory extends beyond the linear case. The architecture consists of a 'LightViT' (Dosovitskiy et al., 2021) image encoder and a 'LightTextTransformer' (Vaswani et al., 2017) text encoder. Both models project their inputs into a shared 128-dimensional embedding space, where a standard InfoNCE contrastive loss is applied during pretraining.

**Pretraining.** The multi-modal model is first pretrained on the MNIST dataset using a standard contrastive objective (similar to CLIP (Radford et al., 2021)) for 10 epochs. This initial model, denoted by weights $\mathbf{W}_I^0$, achieves high accuracy on the original digit recognition task but performs poorly on the color-based task.

**Finetuning Strategies.** We finetune the pretrained image encoder on the ColoredMNIST (Arjovsky et al., 2019; Zhang et al., 2022a) task for 10 epochs while keeping the text encoder frozen, mirroring our theoretical setup. We compare the following methods: (i) **Pretrained:** The baseline model without any finetuning. (ii) **Direct Finetuning (Direct FT):** The image encoder is finetuned on the new task, as analyzed in §C.3. (iii) $L_2$ **Regularization (L2 Reg):** We add a penalty term $\frac{\lambda}{2} \left\| \mathbf{W}_I - \mathbf{W}_I^0 \right\|_F^2$ to the finetuning loss, corresponding to our analysis of $L_2$ regularization. (iv) **Static Distillation (Static Distill):** We use the initial pretrained model $\mathbf{W}_I^0$ as a fixed teacher and add a distillation loss term to the finetuning objective, as analyzed for $\mathbf{W}_{SD}$. (v) **Dynamic Distillation (Dynamic Distill):** We use a teacher model whose weights are moving average of the student's weights, corresponding to our analysis of the WMA teacher.

### 5.1.2 RESULTS AND DISCUSSION

The results of our experiment, visualized in Figure 3, provide strong empirical support for our theoretical analysis.

**Analysis of Forgetting.** As predicted by our theory, **Direct Finetuning** exhibits severe catastrophic forgetting. It achieves near-perfect accuracy (98.5%) on the new color-based task by overwriting its original knowledge, causing its performance on the original MNIST test set to degrade from 96.8% to 59.0%—a forgetting rate of 37.9%. $L_2$ **Regularization** offers an improvement, but still forgets 13.6% of the original task's performance. In contrast, both **Static and Dynamic Distillation** demonstrate remarkable resilience to forgetting. They also master the new task but retain a larger portion of the original knowledge, with forgetting rates of only 1.8% and 0.1%, respectively. This result empirically

confirms our geometric *interpretation*: by interpolating between old and new knowledge within the task-relevant subspace while preserving knowledge in the orthogonal subspace, self-distillation methods achieve a better balance.

**The Performance Trade-off.** The scatter plot in Figure 3 visualizes the core trade-off. The ideal model would reside in the top-right corner, excelling at both tasks. While all finetuned models reach the top of the plot (high spurious task accuracy), the distillation-based methods (Static and Dynamic Distill) are positioned much further to the right. This indicates that they achieve a superior Pareto frontier, retaining more original task accuracy for the same level of new task performance. Dynamic Distillation, by using an adaptive teacher, finds a slightly better solution than its static counterpart, aligning with our theoretical argument for its superiority. These empirical results on non-linear models support our theoretical framework.

## 5.2 Main Results and Ablations

Table 1: **ImageNet accuracy**. We report the accuracy on ImageNet and its distribution shift variants by finetuning CLIP ViT-B/16 with six methods. In each column, the best value is bold and the second-best is underlined.

| Method | IN↑ | IN-V2↑ | IN-R↑ | IN-A↑ | IN-S↑ | ObjectNet↑ | Avg. shifts↑ |
|---|---|---|---|---|---|---|---|
| ZS | 68.33 | 61.93 | 77.71 | 49.95 | 48.26 | 54.17 | 58.39 |
| LP-FT | 82.47 | 72.71 | 72.84 | 49.31 | 50.28 | 54.45 | 59.92 |
| FLYP | 82.69 | 72.73 | 71.35 | 48.52 | 49.84 | 54.86 | 59.40 |
| Lipsum-FT | **83.30** | 73.60 | 75.90 | 49.90 | 51.40 | 54.38 | 61.04 |
| CaRot | 83.13 | **74.11** | 77.71 | 51.60 | 52.71 | 56.60 | 62.55 |
| POMP (Ours) | 82.79 | **74.11** | **79.36** | **54.89** | **53.72** | **58.23** | **64.06** |

Table 2: **ImageNet ECE.** We report the ECE on ImageNet and its distribution shifts to compare with five other finetuning methods, which demonstrates our out-of-distribution (OOD) calibration performance. In each column, the best value is bold and the second-best is underlined.

| Method | IN↓ | IN-V2↓ | IN-R↓ | IN-A↓ | IN-S↓ | ObjectNet↓ | Avg. shifts↓ |
|---|---|---|---|---|---|---|---|
| ZS | 0.0570 | 0.0548 | 0.0541 | **0.0967** | 0.0850 | **0.0780** | 0.0736 |
| LP-FT | 0.0505 | 0.0894 | 0.0613 | 0.2051 | 0.1659 | 0.2124 | 0.1468 |
| FLYP | 0.0635 | 0.1171 | 0.0967 | 0.2435 | 0.2200 | 0.2383 | 0.1836 |
| Lipsum-FT | **0.0384** | 0.0516 | 0.0426 | 0.1290 | 0.1023 | 0.1315 | 0.0914 |
| CaRot | 0.0470 | **0.0367** | 0.0575 | 0.1240 | **0.0699** | 0.1075 | 0.0791 |
| POMP (Ours) | 0.0446 | 0.0394 | **0.0412** | 0.1041 | 0.0784 | 0.1030 | **0.0732** |

### 5.2.1 Experimental Setup

**Objective.** Our experiments are designed to validate our theoretical claims and demonstrate that POMP, as a practical implementation of our framework, achieves SOTA performance in robust finetuning. We focus on evaluating both accuracy and calibration under distribution shifts.

**Datasets and Evaluation.** We use ImageNet-1K (IN) (Deng et al., 2009; Russakovsky et al., 2015) as our in-distribution (ID) downstream task. To measure OOD robustness, we evaluate all finetuned models on a standard suite of five distribution shift datasets: ImageNet-V2 (IN-V2) (Recht et al., 2019), ImageNet-Rendition (IN-R) (Hendrycks et al., 2021a), ImageNet-Adversarial (IN-A) (Hendrycks et al., 2021b), ImageNet-Sketch (IN-S) (Wang et al., 2019), and ObjectNet (Barbu et al., 2019). We report the average performance across these five datasets as "Avg. shifts" or "OOD".

**Baselines.** We compare POMP against a comprehensive set of baselines, including zero-shot (ZS), direct full finetuning (FT), linear probing then finetuning (LP-FT), finetune-like-you-pretrain (FLYP), and recent state-of-the-art robust finetuning methods like CaRot.

**Metrics.** We report top-1 accuracy and Expected Calibration Error (ECE). ECE measures the gap between predicted confidence and empirical accuracy across confidence bins. Let $\{B_m\}_{m=1}^M$ partition

Table 3: **ImageNet Accuracy.** (except ObjectNet) with additional baselines.

| Method | IN↑ | IN-V2↑ | IN-R↑ | IN-A↑ | IN-S↑ | Avg. shifts↑ |
|---|---|---|---|---|---|---|
| ZS | 68.33 | 61.93 | 77.71 | 49.95 | 48.26 | 59.46 |
| Direct FT | 82.80 | 72.60 | 68.50 | 39.20 | 48.00 | 57.08 |
| L2-SP (Li et al., 2018) | 82.90 | 72.60 | 68.80 | 39.70 | 48.20 | 57.33 |
| Static SD (Hinton et al., 2015) | 82.10 | 73.10 | 72.90 | 42.30 | 49.90 | 59.55 |
| LP-FT (Kumar et al., 2022) | 82.17 | 72.06 | 70.47 | 46.29 | 48.68 | 59.38 |
| FLYP (Goyal et al., 2023) | 82.69 | 72.73 | 71.35 | 48.52 | 49.84 | 60.61 |
| CAR-FT (Mao et al., 2024) | 83.30 | 74.00 | 75.40 | 49.50 | 53.00 | 62.98 |
| Lipsum-FT (Nam et al., 2024) | 83.30 | 73.60 | 75.90 | 49.90 | 51.40 | 62.70 |
| Model Stock (Jang et al., 2024) | **84.10** | **74.80** | 71.80 | 51.20 | 51.80 | 62.40 |
| ARF (Han et al., 2024) | 82.70 | 72.80 | 75.60 | 50.30 | 51.80 | 62.63 |
| CaRot (Oh et al., 2024) | 83.13 | 74.11 | 77.71 | 51.60 | 52.71 | 64.03 |
| POMP (Ours) | 82.79 | 74.09 | **79.33** | **54.69** | **53.72** | **65.46** |

Table 4: **ImageNet accuracy and ECE on different backbones**. We provide summarized results on CLIP ResNet50 and ViT-L/14. The best and the second-best in each column are underlined. (See Table 6 and 7 for details.)

| | Method | ID Acc.↑ | ID ECE↓ | OOD Acc.↑ | OOD ECE↓ | | ID Acc.↑ | ID ECE↓ | OOD Acc.↑ | OOD ECE↓ |
|---|---|---|---|---|---|---|---|---|---|---|
| RN50 | ZS | 59.83 | 0.0624 | 42.52 | **0.0955** | ViT-L/14 | 75.55 | 0.0590 | 70.93 | **0.0711** |
| | FT | 76.21 | 0.0983 | 41.97 | 0.2804 | | 85.26 | 0.0993 | 65.98 | 0.2036 |
| | LP-FT | 76.25 | 0.1042 | 41.62 | 0.3274 | | 84.74 | 0.1056 | 64.11 | 0.2521 |
| | FLYP | 76.16 | 0.0516 | 42.70 | 0.2127 | | 86.19 | 0.0729 | 71.44 | 0.1470 |
| | CaRot | 76.12 | 0.0471 | 42.71 | 0.2109 | | **86.95** | **0.0349** | 74.13 | 0.0737 |
| | POMP (Ours) | **76.48** | **0.0470** | **42.73** | 0.1807 | | 86.27 | 0.0507 | **75.32** | 0.0732 |

examples by confidence, with $\mathrm{acc}(B_m)$ and $\mathrm{conf}(B_m)$ denoting accuracy and mean confidence in bin $m$. Then

$$\mathrm{ECE} = \sum_{m=1}^{M} \frac{|B_m|}{N} \big| \mathrm{acc}(B_m) - \mathrm{conf}(B_m) \big|.$$

Lower ECE is better and indicates more reliable probabilities, which is critical under shift where overconfidence is common. We report averages over the five OOD datasets (IN-V2, IN-R, IN-A, IN-S, ObjectNet) to summarize robustness and calibration.

**Implementation Details and Experimental Setup.** We finetune CLIP variants on ImageNet-1K (IN) and evaluate on five OOD datasets: IN-V2, IN-R, IN-A, IN-S, and ObjectNet, following Taori et al. (2020). For all methods, we finetune for 10 epochs using the AdamW optimizer with a learning rate of $1 \times 10^{-5}$ and a weight decay of $0.01$. The batch size is set to 224 for ViT-L/14 and 512 for ViT-B/16 and ResNet50. For POMP, the WMA teacher uses a $\mathrm{Beta}(0.5, 0.5)$ weighting kernel and combines symmetric InfoNCE with the composite SD loss (§C.6).

### 5.2.2 RESULTS AND ANALYSIS

**OOD accuracy and calibration on ViT-B/16.** As shown in Table 1, POMP demonstrates superior OOD performance. On the ViT-B/16 backbone, our method achieves SOTA accuracy across the five distribution shift datasets (including IN-V2, IN-R, IN-A, IN-S, and ObjectNet), outperforming all other methods. Notably, POMP achieves the best results on the most challenging shifts, particularly ObjectNet and IN-A. While Direct FT improves ID accuracy significantly, its OOD performance is even worse than the zero-shot model (Table 3), empirically confirming the catastrophic forgetting problem our work addresses. In Table 2, POMP achieves the lowest average OOD ECE, indicating probabilistic reliability under shift. With additional baselines (Table 3), POMP remains the top OOD performer while staying competitive on IN. Additionally, the cross-backbone experiments (Table 4) confirm POMP's generality: it achieves the best OOD accuracy for both RN50 and ViT-L/14 while keeping ECE competitive.

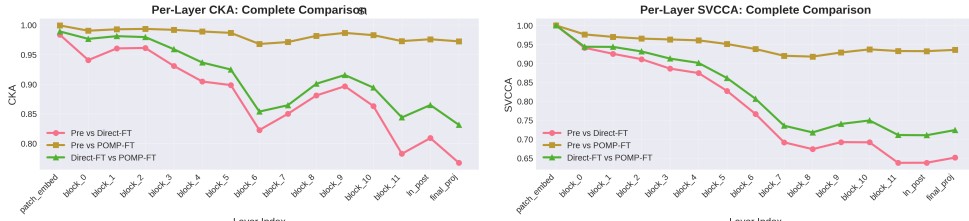

Figure 4: **Layer-wise Representational Similarity.** We compare the internal representations of the Pretrained model against `Direct FT` and POMP using CKA (left) and SVCCA (right) across all layers of the CLIP ViT-B/16 image encoder. POMP (gold) preserves the geometric structure of the pretrained knowledge significantly better than `Direct FT` (pink), particularly in deeper layers.

**Empirical Validation of Geometric Preservation.** To verify our theoretical claim that POMP preserves knowledge in the orthogonal subspace, we conduct a layer-wise representational similarity analysis using Centered Kernel Alignment (CKA) (Kornblith et al., 2019) on the CLIP ViT-B/16 image encoder. As shown in Figure 4, `Direct FT` exhibits a precipitous drop in similarity in the deeper layers (Blocks 6–11) relative to the pretrained model. This confirms that catastrophic forgetting manifests as a **Feature Distortion** of high-level semantic representations. In contrast, POMP maintains near-perfect similarity ($> 0.97$) across all layers. This provides strong empirical evidence for our geometric interpretation: POMP successfully anchors the optimization to the pretrained geometry, performing surgical updates that adapt to the task without overwriting robust feature extractors.

**Computational Efficiency and Complexity.** In addition to superior robustness, POMP offers significant efficiency advantages over prior SOTA methods like CaRot (Oh et al., 2024). CaRot relies on spectral regularization requiring matrix orthogonality constraints, which scale cubically with the projection dimension ($\mathcal{O}(d^3)$). Conversely, POMP's composite distillation operates on batch similarity matrices, scaling with batch size ($\mathcal{O}(B^2)$), where typically $B \ll d$. As detailed in Table 5, this theoretical advantage translates to a reduction in training time per epoch on ImageNet compared to CaRot.

Table 5: **Computational Efficiency Comparison.** Average training time per epoch on ImageNet-1K using CLIP ViT-B/16 on an NVIDIA H100 GPU.

| Method | Dominant Cost | Time / Epoch | Relative Overhead | Avg. OOD Acc. |
|---|---|---|---|---|
| Direct FT | $\mathcal{O}(P)$ (Backprop) | $\sim 16$ min | $1.00\times$ | 57.08% |
| CaRot (Oh et al., 2024) | $\mathcal{O}(d^3)$ (Matrix Reg.) | $\sim 29$ min | $1.81\times$ | 62.55% |
| POMP (Ours) | $\mathcal{O}(B^2)$ (Batch Distill) | $\sim 22$ min | $1.38\times$ | **64.06%** |

**Validating Teacher Dynamics and Regularization Strength** Our theory posits that the WMA teacher in POMP provides a more persistent regularizing signal than the EMA teacher used in methods like CaRot. To validate this, we track the KL divergence between teacher and student, throughout training on ImageNet. As shown in Figure 5, for the EMA teacher, the KL decays steadily, indicating that the teacher is rapidly collapsing onto the student and its regularizing influence is diminishing. In contrast, the WMA teacher maintains a higher and more stable KL throughout the entire training process. This sustained divergence confirms that the WMA teacher provides a persistent "restoring force," as predicted by our analysis in §C.5.2. This prevents the student from converging to a narrow task-specific minimum and is key to POMP's superior OOD robustness and calibration.

Furthermore, this stability translates into algorithmic simplicity. While EMA-based methods often require complex, sparse update schedules (e.g., updating only every 500 steps with linear ramping) to prevent collapse, POMP is robust to update frequency. As shown in Table 10, POMP maintains consistent SOTA performance ($\sim 64.0 - 64.2\%$ OOD accuracy) whether the teacher is updated every step or every 500 steps, eliminating the need for brittle hyperparameter tuning. Figure 6 explicitly illustrates this failure mode in prior methods: without careful tuning, the EMA teacher in CaRot

collapses immediately when updated at every step, whereas POMP's WMA teacher remains stable even under the simplest dense update schedule.

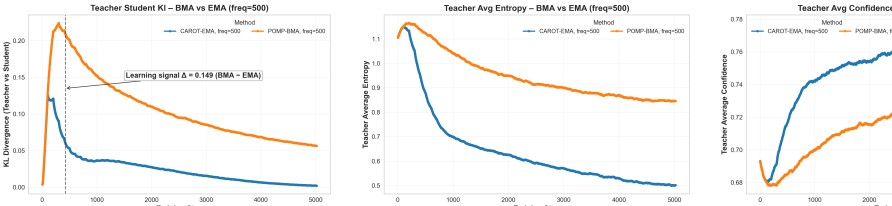

Figure 5: **Teacher-Student Knowledge Gap during training**. Compared to the EMA teacher (blue), which shows rapidly vanishing KL divergence and thus a weakening regularization signal (left), the WMA teacher (orange) sustains a higher and more stable KL gap. This stability is supported by higher teacher entropy (middle) and moderated confidence (right), preventing overfitting. Together, these trends confirm that WMA provides a stronger and more persistent self-distillation signal than EMA.

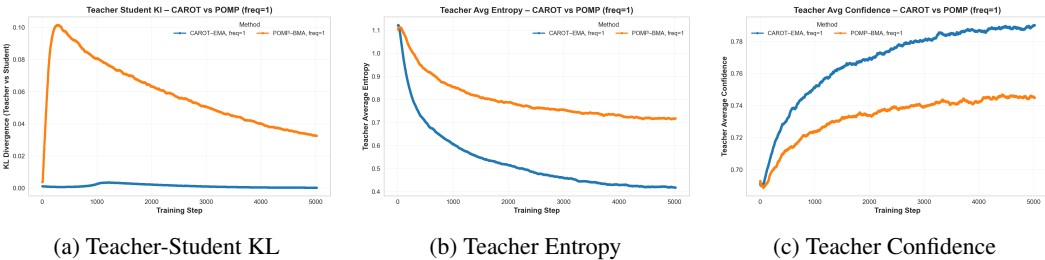

(a) Teacher-Student KL          (b) Teacher Entropy          (c) Teacher Confidence

Figure 6: **Comparison of Teacher Dynamics (Update Frequency = 1).** We track the evolution of the teacher model for CaRot (EMA) and POMP (WMA) when updated at every step. The EMA teacher (blue) rapidly collapses onto the student (KL $\rightarrow$ 0), losing its regularizing capability. The WMA teacher (orange) maintains a persistent, stable gap, providing continuous regularization without needing brittle update schedules.

**Ablation Studies.** Beyond the primary results, we conducted comprehensive ablation studies (detailed in §B) to further validate POMP's design. We found that combining all four multi-perspective distillation components (FD, CRD, ICL, Cross-KD) is crucial for optimal performance, demonstrating their complementary roles in preserving diverse aspects of knowledge. The distillation strength ($\lambda_{SD}$) and the WMA Beta-kernel shape also significantly impact the ID-OOD trade-off, with moderate $\lambda_{SD}$ and arcsine-like weighting (Beta$(0.5, 0.5)$) proving most effective for robust and calibrated performance, aligning with our theoretical insights into persistent and endpoint-aware regularization.

## 6 CONCLUSION

We proved that POMP's trajectory-averaging WMA teacher, unlike its EMA counterpart, maintains a persistent, non-vanishing regularizing force. This force continuously anchors the model to its robust pretrained initialization, preventing late-stage overfitting and explaining POMP's state-of-the-art out-of-distribution performance. Our work bridges the geometry of finetuning with the practical design of robust methods, and these principles motivate future extensions to parameter-efficient methods and continual learning.

## 7 REPRODUCIBILITY STATEMENT

The empirical results presented in this paper, including those from the synthetic validation and the ImageNet experiments are fully reproducible. Our implementation, based on PyTorch and leveraging the OpenAI CLIP library, will be made publicly available. This includes all model architectures, training procedures, data processing steps, and evaluation protocols. Detailed descriptions of hyperparameters, and environment specification for running experiments, are provided in §E.

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

APPENDIX

# A EXTENDED RELATED WORK

## A.1 CONTRASTIVE LANGUAGE-IMAGE PRETRAINING

Initial advancements in contrastive learning between vision and language modalities were made by Virtex (Desai and Johnson, 2021), ICMLM (Sariyildiz et al., 2020), and ConVIRT (Zhang et al., 2022c). These early approaches laid the groundwork for later models like CLIP (Radford et al., 2021; Ilharco et al., 2021) and ALIGN (Jia et al., 2021), which scaled contrastive techniques to larger datasets and model architectures. Subsequent work explores improved cross-modal interaction and training recipes (Yuan et al., 2021; Yu et al., 2022; Fang et al., 2023). Following these, several open-weight contrastive models have been introduced to improve CLIP's performance and robustness (Sun et al., 2023; Zhai et al., 2023; Li et al., 2023a; Fang et al., 2024; Xu et al., 2024a; Schuhmann et al., 2022). For example, SigLIP (Zhai et al., 2023; Tschannen et al., 2025) modifies the contrastive loss by using a sigmoid function instead of softmax, and FLIP (Li et al., 2023c) integrates masking strategies to accelerate training.

## A.2 THEORY OF CONTRASTIVE LEARNING

A rich theoretical literature analyzes contrastive learning from first principles, characterizing when and why contrastive objectives recover useful features and class structure (Saunshi et al., 2019). The alignment–uniformity lens formalizes how pulling positives together while spreading embeddings uniformly on the sphere drives representation quality (Wang and Isola, 2020). For tractability, many works study *linearized* or simplified contrastive losses that replace log-exp with linear functions and show that their gradients align with those of standard objectives up to reweighting, enabling closed-form analysis and geometric insight (Ji et al., 2023; Tian, 2022; Nakada et al., 2023; Xue et al., 2024). This linearized viewpoint has proven effective in theoretical analyses across self-supervised contrastive learning (CL) (Ji et al., 2023; HaoChen et al., 2022; HaoChen and Ma, 2023; Shen et al., 2022a), multi-modal contrastive learning (MMCL) (Nakada et al., 2023), non-contrastive methods (Liu et al., 2022), and supervised CL (Xue et al., 2023). Complementing these results, large-scale empirical studies suggest that many design choices of popular losses (e.g., log-exp, cosine similarity) are not essential for effective representation learning (Garrido et al., 2023).

## A.3 FINETUNING, FORGETTING, AND REGULARIZATION

Catastrophic forgetting—adapting to new data at the expense of prior knowledge—has long been recognized as a central challenge in sequential and transfer learning (McCloskey and Cohen, 1989; French, 1999). Mitigation strategies include: (i) regularization, which constrains parameter updates via importance penalties or output consistency (Kirkpatrick et al., 2017; Zenke et al., 2017; Li and Hoiem, 2018); (ii) replay, which mixes current data with stored or synthesized memories (Robins, 1995; Rebuffi et al., 2017; Aljundi et al., 2019); and (iii) architectural growth, which expands capacity and distills across modules (Rusu et al., 2016; Yan et al., 2021; Wang et al., 2022a). L2-SP (Li et al., 2018) tethers the solution to the pretrained initialization via weight-space regularization, while output-space regularizers distill prior behaviors during adaptation (Li and Hoiem, 2018). Additionally, parameter-efficient finetuning methods such as adapters (Houlsby et al., 2019) and prefix tuning (Li and Liang, 2021) enable task adaptation without full model updates, thus mitigating forgetting. Among these, Low-Rank Adaptation (LoRA) (Hu et al., 2022) has gained prominence for finetuning large language models by injecting trainable low-rank matrices into existing weights, achieving competitive performance with reduced parameter updates and minimal forgetting. Further work explores functional regularization (Titsias et al., 2020) and knowledge-preserving contrastive losses (Jung et al., 2020) to encourage feature stability. As model sizes grow, scalable and minimally invasive adaptation techniques—balancing plasticity and stability—remain critical to continual and transfer learning paradigms.

## A.4 ROBUST FINETUNING OF CLIP

Robustness evaluates how well models maintain performance under distribution shifts, which can include synthetic corruptions (Hendrycks and Dietterich, 2019) as well as real-world variations in viewpoint, style, and time (Barbu et al., 2019; Hendrycks et al., 2021a; Wang et al., 2019; Recht et al., 2019). A standard protocol for evaluating CLIP-like models, proposed by Taori et al. (2020), involves finetuning on ImageNet and measuring transfer performance on a suite of realistic OOD sets (ImageNet-V2, -A, -R, -Sketch, and ObjectNet), which is now standard practice. This evaluation highlights a central challenge: naive finetuning methods like Linear Probing (LP), which only trains a classification head, or **Direct Full finetuning**, which updates all parameters, often create a trade-off between in-distribution (ID) performance and OOD robustness. To address this, a diverse array of robust finetuning techniques has been developed. A prominent line of work involves post-hoc averaging or interpolating model weights. For instance, `WiSE-FT` (Wortsman et al., 2022b) averages the weights of the zero-shot and a fully finetuned model, while `Model Soup` (Wortsman et al., 2022a) averages the weights of multiple models found through a hyperparameter search. This concept is extended by `Model Stock` (Jang et al., 2024), which efficiently builds and averages a diverse set of minimally adapted models. Other post-hoc methods include `TPGM` (Tian et al., 2023a) and its efficient successor `Fast TPGM` (Tian et al., 2023b), which project finetuned weights back towards the initial weights, and `DaWin` (Oh et al., 2025), which introduces a training-free, dynamic interpolation where the mixing coefficient is decided on a per-sample basis using predictive entropy.

Beyond post-hoc modifications, many methods introduce regularization during the finetuning process itself. These can constrain the model in weight-space, such as `L2-SP` (Li et al., 2018) which penalizes weight deviation, or by maintaining an `EMA` of model parameters to find smoother, more robust solutions. Others operate in the output-space, where **Knowledge Distillation (KD)** (Hinton et al., 2015) aligns the student's predictions with the robust zero-shot teacher. A particularly relevant strategy for Vision-Language Models is using the text modality for guidance. This includes continuing contrastive learning with supervised image-text pairs as in Finetune-Like-You-Pretrain (`FLYP`) (Goyal et al., 2023), aligning with fixed context-specific prompts in `CAR-FT` (Mao et al., 2024), regularizing the model's energy function using random texts to preserve broad semantic alignment in `Lipsum-FT` (Nam et al., 2024), or improving discrimination with both positive and negative prompts as in `CLIPood` (Shu et al., 2023). Alternative strategies modify the training pipeline, such as the two-stage `LP-FT` approach (Kumar et al., 2022) which first finds a good head via linear probing before full finetuning. More advanced methods like `CaRot` (Oh et al., 2024) aim to simultaneously improve OOD accuracy and confidence calibration through a principled combination of contrastive learning and novel regularization terms.

## A.5 KNOWLEDGE DISTILLATION AND SELF-DISTILLATION

Knowledge Distillation (KD) was initially introduced for compression, where a smaller student learns from a larger teacher's outputs (Hinton et al., 2015). The same principle underpins continual and transfer learning, where a pretrained model guides finetuning to preserve capabilities, often termed Learning without Forgetting (LwF) (Li and Hoiem, 2018). Self-distillation (SD) is a special case where the model learns from its own initial state (Zhang et al., 2019; Mobahi et al., 2020). Beyond single-modality SD, multi-modal KD aligns internal signals and outputs to preserve cross-modal structure (Fang et al., 2021; Wang et al., 2022b; Li et al., 2023b; Liang et al., 2023; Li et al., 2024), with recent work demonstrating effective CLIP distillation via affinity matching and weight inheritance (Wu et al., 2023; Yang et al., 2024).

## A.6 DYNAMIC TEACHERS, WEIGHT AVERAGING, AND MODE CONNECTIVITY

Temporal ensembling and EMA teachers stabilize training and improve targets (Laine and Aila, 2017; Tarvainen and Valpola, 2017), and they underpin momentum-encoder methods in self-supervised learning (He et al., 2020; Grill et al., 2020; Caron et al., 2021). Separately, model averaging and linear mode connectivity suggest that interpolations and averages often lie in flat, low-loss regions and improve robustness (Izmailov et al., 2018; Frankle and Carbin, 2019). Wise-FT leverages interpolation between pretrained and finetuned weights to strengthen OOD performance (Wortsman et al., 2022b;a).

## B  ABLATION STUDIES AND ADDITIONAL EXPERIMENTAL RESULTS

**Experimental Protocol.**   We use the same seeds, and hyperparameter configurations as in the main experiments, varying only the stated factor per ablation.

**Additional Experimental Results.**   To further demonstrate the generalizability of our method, we present results using the CLIP RN50 and ViT-L/14 backbones. A summary of these experiments is provided in Table 4, with detailed results reported below.

Table 6: **ImageNet results on CLIP ResNet50**

| Method | IN | IN-V2 | IN-R | IN-A | IN-S | ObjectNet | Avg. shifts |
|---|---|---|---|---|---|---|---|
| | | Acc.↑ | | | | | |
| ZS | 59.83 | 52.90 | 60.72 | 23.25 | 35.45 | 40.27 | 42.52 |
| FT | 76.21 | 64.87 | 50.66 | 18.11 | 33.90 | 42.32 | 41.97 |
| LP-FT | 76.25 | 64.48 | 49.55 | 18.60 | 33.33 | 42.13 | 41.62 |
| FLYP | 76.16 | 65.10 | 51.55 | 20.08 | 34.24 | 42.53 | 42.70 |
| CaRot | 76.12 | 65.36 | 52.16 | 19.32 | 34.05 | 42.67 | 42.71 |
| POMP (Ours) | 76.48 | 65.58 | 51.54 | 19.52 | 34.34 | 42.66 | 42.73 |
| | | ECE↓ | | | | | |
| ZS | 0.0624 | 0.0559 | 0.0530 | 0.2048 | 0.0740 | 0.0899 | 0.0955 |
| FT | 0.0983 | 0.1623 | 0.1860 | 0.4692 | 0.2824 | 0.3023 | 0.2804 |
| LP-FT | 0.1042 | 0.1759 | 0.2709 | 0.5184 | 0.3520 | 0.3197 | 0.3274 |
| FLYP | 0.0516 | 0.0872 | 0.1439 | 0.3872 | 0.2021 | 0.2432 | 0.2127 |
| CaRot | 0.0471 | 0.0601 | 0.0948 | 0.3435 | 0.3435 | 0.2127 | 0.2109 |
| POMP (Ours) | 0.0470 | 0.0564 | 0.1176 | 0.3456 | 0.1741 | 0.2097 | 0.1807 |

Table 7: **ImageNet results on CLIP ViT-L/14**

| Method | IN | IN-V2 | IN-R | IN-A | IN-S | ObjectNet | Avg. shifts |
|---|---|---|---|---|---|---|---|
| | | Acc.↑ | | | | | |
| ZS | 75.55 | 69.85 | 87.85 | 70.76 | 59.61 | 66.59 | 70.93 |
| FT | 84.74 | 75.32 | 75.36 | 55.65 | 54.44 | 59.76 | 64.11 |
| LP-FT | 85.26 | 76.76 | 80.21 | 55.95 | 56.84 | 60.12 | 65.98 |
| FLYP | 86.19 | 78.21 | 83.81 | 68.85 | 60.20 | 66.15 | 71.44 |
| CaRot | 86.95 | 79.28 | 87.96 | 72.68 | 62.66 | 68.05 | 74.13 |
| POMP (Ours) | 86.27 | 78.54 | 89.70 | 74.87 | 63.71 | 69.76 | 75.32 |
| | | ECE↓ | | | | | |
| ZS | 0.0590 | 0.0686 | 0.0339 | 0.0640 | 0.1037 | 0.0852 | 0.0711 |
| FT | 0.1056 | 0.1741 | 0.1613 | 0.3151 | 0.3234 | 0.2865 | 0.2521 |
| LP-FT | 0.0993 | 0.1531 | 0.0872 | 0.2593 | 0.2613 | 0.2572 | 0.2036 |
| FLYP | 0.0729 | 0.1219 | 0.0621 | 0.1443 | 0.2164 | 0.1903 | 0.1470 |
| CaRot | 0.0349 | 0.0634 | 0.0353 | 0.0732 | 0.0914 | 0.1051 | 0.0737 |
| POMP (Ours) | 0.0507 | 0.0581 | 0.0442 | 0.0665 | 0.1052 | 0.0918 | 0.0732 |

**Ablation 1: Multi-perspective distillation.** The ablation study on multi-perspective distillation (Table 8) quantifies the contribution of each perspective to out-of-distribution (OOD) accuracy and calibration. Results show that CRD and FD emerge as the strongest individual components for OOD accuracy and ECE, respectively, while combining all four perspectives yields the best overall performance and remains among the top performers on OOD metrics. These findings highlight the complementary nature of the terms: FD stabilizes features, CRD preserves batch-level relational structure, ICL enriches mutual information in the teacher's space, and CrossKD blends relational and interactive cues.

Table 8: Ablation of POMP components across ImageNet (IN) and distribution shifts. For each setting (row), accuracy (Acc.↑) is reported in %, and expected calibration error (ECE↓) in $[0, 1]$. OOD Avg. is the mean over {IN-V2, IN-R, IN-A, IN-S, ObjectNet}. Method names encode the presence of losses ($\mathcal{L}_{\text{FD}}, \mathcal{L}_{\text{CrossKD}}, \mathcal{L}_{\text{ICL}}, \mathcal{L}_{\text{CRD}}$) as ✓ or −. Rows with only one loss term active are in gray.

| $\mathcal{L}_{\text{FD}}$ | $\mathcal{L}_{\text{CrossKD}}$ | $\mathcal{L}_{\text{ICL}}$ | $\mathcal{L}_{\text{CRD}}$ | IN | IN-V2 | Acc.↑ IN-R | IN-A | IN-S | ObjectNet | Avg. shifts | Avg. All |
|---|---|---|---|---|---|---|---|---|---|---|---|
| − | − | − | − | 82.69 | 72.73 | 71.35 | 48.52 | 49.84 | 54.86 | 59.40 | 63.33 |
| − | − | − | ✓ | 83.17 | 74.29 | 77.75 | 53.09 | 53.03 | 57.46 | 63.12 | 66.47 |
| − | − | ✓ | − | 82.50 | 73.13 | 72.12 | 49.23 | 50.03 | 55.26 | 59.95 | 63.71 |
| − | − | ✓ | ✓ | 83.23 | 74.31 | 76.50 | 52.39 | 52.53 | 57.00 | 62.55 | 65.99 |
| − | ✓ | − | − | 83.19 | 74.04 | 74.67 | 50.65 | 51.39 | 56.40 | 61.43 | 65.06 |
| − | ✓ | − | ✓ | 83.08 | 74.40 | 78.68 | 53.53 | 53.37 | 57.45 | 63.49 | 66.75 |
| − | ✓ | ✓ | − | 83.03 | 73.95 | 74.11 | 50.60 | 51.28 | 55.77 | 61.14 | 64.79 |
| − | ✓ | ✓ | ✓ | 83.27 | 74.48 | 77.60 | 52.76 | 52.94 | 57.28 | 63.01 | 66.39 |
| ✓ | − | − | − | 83.06 | 74.16 | 78.14 | 54.39 | 53.14 | 57.79 | 63.52 | 66.78 |
| ✓ | − | − | ✓ | 82.45 | 73.91 | 79.67 | 54.88 | 53.93 | 58.02 | **64.08** | 67.14 |
| ✓ | − | ✓ | − | 83.08 | 74.39 | 77.59 | 53.84 | 53.10 | 57.59 | 63.30 | 66.60 |
| ✓ | − | ✓ | ✓ | 82.92 | 74.40 | 79.21 | 54.61 | 53.75 | 57.99 | 63.99 | 67.15 |
| ✓ | ✓ | − | − | 83.01 | 74.21 | 78.91 | 54.09 | 53.28 | 58.04 | 63.71 | 66.92 |
| ✓ | ✓ | − | ✓ | 82.27 | 73.71 | 79.81 | 54.65 | 53.82 | 58.14 | 64.03 | 67.07 |
| ✓ | ✓ | ✓ | − | 83.06 | 74.38 | 78.38 | 54.07 | 53.25 | 57.86 | 63.59 | 66.83 |
| ✓ | ✓ | ✓ | ✓ | 82.81 | 73.94 | 79.55 | 54.83 | 53.96 | 58.02 | 64.06 | **67.19** |

| $\mathcal{L}_{\text{FD}}$ | $\mathcal{L}_{\text{CrossKD}}$ | $\mathcal{L}_{\text{ICL}}$ | $\mathcal{L}_{\text{CRD}}$ | IN | IN-V2 | ECE.↓ IN-R | IN-A | IN-S | ObjectNet | Avg. shifts | Avg. All |
|---|---|---|---|---|---|---|---|---|---|---|---|
| − | − | − | − | 0.0635 | 0.1171 | 0.0967 | 0.2435 | 0.2200 | 0.2383 | 0.1836 | 0.1632 |
| − | − | − | ✓ | 0.0415 | 0.0412 | 0.0413 | 0.1328 | 0.0860 | 0.1211 | 0.0845 | 0.0773 |
| − | − | ✓ | − | 0.0585 | 0.1000 | 0.0817 | 0.2117 | 0.1974 | 0.2168 | 0.1615 | 0.1444 |
| − | − | ✓ | ✓ | **0.0393** | 0.0523 | 0.0429 | 0.1534 | 0.1111 | 0.1441 | 0.1008 | 0.0905 |
| − | ✓ | − | − | 0.0483 | 0.0830 | 0.0662 | 0.2007 | 0.1660 | 0.1918 | 0.1415 | 0.1260 |
| − | ✓ | − | ✓ | 0.0453 | 0.0374 | 0.0434 | 0.1141 | 0.0753 | 0.1096 | 0.0760 | 0.0709 |
| − | ✓ | ✓ | − | 0.0507 | 0.0824 | 0.0691 | 0.2007 | 0.1684 | 0.1968 | 0.1435 | 0.1280 |
| − | ✓ | ✓ | ✓ | 0.0401 | 0.0442 | 0.0392 | 0.1345 | 0.0897 | 0.1260 | 0.0867 | 0.0790 |
| ✓ | − | − | − | 0.0430 | 0.0674 | 0.0479 | 0.1592 | 0.1383 | 0.1661 | 0.1158 | 0.1037 |
| ✓ | − | − | ✓ | 0.0474 | 0.0380 | 0.0455 | 0.1034 | 0.0720 | 0.0975 | 0.0713 | **0.0673** |
| ✓ | − | ✓ | − | 0.0436 | 0.0684 | 0.0482 | 0.1632 | 0.1384 | 0.1691 | 0.1175 | 0.1052 |
| ✓ | − | ✓ | ✓ | 0.0419 | 0.0454 | 0.0397 | 0.1157 | 0.0853 | 0.1162 | 0.0805 | 0.0740 |
| ✓ | ✓ | − | − | 0.0399 | 0.0537 | 0.0398 | 0.1340 | 0.1082 | 0.1374 | 0.0946 | 0.0855 |
| ✓ | ✓ | − | ✓ | 0.0531 | 0.0404 | 0.0500 | 0.0936 | 0.0703 | 0.0888 | **0.0686** | 0.0660 |
| ✓ | ✓ | ✓ | − | 0.0402 | 0.0572 | 0.0418 | 0.1420 | 0.1176 | 0.1499 | 0.1017 | 0.0915 |
| ✓ | ✓ | ✓ | ✓ | 0.0446 | 0.0416 | 0.0430 | 0.1027 | 0.0757 | 0.1054 | 0.0737 | 0.0688 |

**Ablation 2: Distillation strength $\lambda_{SD}$.** The ablation on distillation strength $\lambda_{SD}$ (Table 9) examines the balance between teacher influence and task adaptation. We sweep $\lambda_{SD} \in \{0.1, 0.2, 0.3, 0.4, 0.5, 0.7, 1.0, 1.2, 1.5, 2.0, 3.0, 4.0, 5.0, 10.0\}$. Results indicate that moderate values of $\lambda_{SD}$ ($\approx 1.0$–$2.0$) achieve the best OOD accuracy, while larger values improve calibration by lowering ECE but slightly reduce in-distribution (ID) accuracy. This aligns with our theory that stronger distillation enhances calibration through teacher anchoring, whereas moderate strength provides the optimal trade-off between adaptation and preservation for OOD performance.

Table 9: Ablation of distillation coefficient $\lambda_{SD}$ across ImageNet (IN) and distribution shifts. For each setting (row), accuracy (Acc.$\uparrow$) is reported in %, and expected calibration error (ECE$\downarrow$) in $[0, 1]$. OOD Avg. is the mean over {IN-V2, IN-R, IN-A, IN-S, ObjectNet}.

| $\lambda_{SD}$ | Acc.$\uparrow$ (%) | | | | | | |
|---|---|---|---|---|---|---|---|
| | IN | IN-V2 | IN-R | IN-A | IN-S | ObjectNet | OOD Avg. |
| 10.0 | 80.52 | 72.08 | 79.50 | 54.07 | 53.27 | 57.45 | 63.27 |
| 5.0 | 81.08 | 72.54 | 79.79 | 54.37 | 53.64 | 57.65 | 63.60 |
| 4.0 | 81.25 | 72.67 | 79.78 | 54.75 | 53.74 | 57.66 | 63.72 |
| 3.0 | 81.58 | 73.12 | 79.90 | 54.83 | 53.84 | 57.75 | 63.89 |
| 2.0 | 81.94 | 73.49 | **80.03** | 54.64 | **54.02** | 57.88 | 64.01 |
| 1.5 | 82.27 | 73.52 | 79.72 | **55.28** | 53.99 | 58.08 | **64.12** |
| 1.2 | 82.50 | 73.74 | 79.55 | 55.20 | 53.94 | **58.14** | 64.11 |
| 1.0 | 82.70 | 74.07 | 79.64 | 54.87 | 53.85 | 58.08 | 64.10 |
| 0.7 | 82.90 | 74.41 | 79.21 | 54.72 | 53.73 | 58.03 | 64.02 |
| 0.5 | 83.16 | 74.31 | 78.76 | 54.13 | 53.52 | 57.76 | 63.70 |
| 0.4 | 83.26 | 74.48 | 78.19 | 53.64 | 53.27 | 57.82 | 63.48 |
| 0.3 | 83.28 | **74.52** | 77.53 | 53.55 | 52.91 | 57.44 | 63.19 |
| 0.2 | **83.29** | 74.30 | 76.80 | 52.61 | 52.58 | 57.04 | 62.67 |
| 0.1 | 83.25 | 74.08 | 75.34 | 51.52 | 51.89 | 56.49 | 61.86 |

| $\lambda_{SD}$ | ECE$\downarrow$ | | | | | | |
|---|---|---|---|---|---|---|---|
| | IN | IN-V2 | IN-R | IN-A | IN-S | ObjectNet | OOD Avg. |
| 10.0 | 0.0637 | 0.0475 | 0.0621 | 0.0839 | 0.0725 | 0.0795 | 0.0691 |
| 5.0 | 0.0631 | 0.0467 | 0.0600 | **0.0812** | 0.0700 | **0.0772** | **0.0670** |
| 4.0 | 0.0606 | 0.0457 | 0.0583 | 0.0817 | 0.0705 | 0.0801 | 0.0673 |
| 3.0 | 0.0590 | 0.0466 | 0.0566 | 0.0866 | 0.0701 | 0.0814 | 0.0683 |
| 2.0 | 0.0547 | 0.0422 | 0.0528 | 0.0885 | **0.0682** | 0.0863 | 0.0676 |
| 1.5 | 0.0511 | **0.0396** | 0.0490 | 0.0911 | 0.0707 | 0.0897 | 0.0680 |
| 1.2 | 0.0484 | 0.0408 | 0.0455 | 0.0963 | 0.0713 | 0.0957 | 0.0699 |
| 1.0 | 0.0465 | 0.0410 | 0.0445 | 0.1001 | 0.0742 | 0.1010 | 0.0722 |
| 0.7 | 0.0419 | 0.0445 | 0.0416 | 0.1120 | 0.0841 | 0.1144 | 0.0793 |
| 0.5 | 0.0409 | 0.0466 | **0.0390** | 0.1259 | 0.0953 | 0.1295 | 0.0873 |
| 0.4 | **0.0394** | 0.0502 | 0.0415 | 0.1353 | 0.1030 | 0.1363 | 0.0933 |
| 0.3 | 0.0397 | 0.0554 | 0.0424 | 0.1459 | 0.1161 | 0.1502 | 0.1020 |
| 0.2 | 0.0421 | 0.0626 | 0.0463 | 0.1628 | 0.1341 | 0.1660 | 0.1144 |
| 0.1 | 0.0470 | 0.0798 | 0.0601 | 0.1890 | 0.1594 | 0.1895 | 0.1356 |

**Ablation 3: Teacher update frequency.** The ablation on teacher update frequency (Table 10) investigates the trade-off between stability and plasticity in the dynamic teacher. We vary the frequency from 1 to 2500 steps ($\approx$ 1 epoch). The results show that updating every 50–100 steps yields the highest OOD accuracy, whereas slower update schedules lead to lower OOD ECE. These findings align with the dynamic-teacher analysis: slower updates preserve early robustness and calibration, while faster updates allow the teacher to better track the evolving task solution and enhance accuracy.

Table 10: Ablation of teacher update frequency in POMP distillation across ImageNet (IN) and distribution shifts. For each setting (row), accuracy (Acc.↑) is reported in %, and expected calibration error (ECE↓) in $[0, 1]$. OOD Avg. is the mean over {IN-V2, IN-R, IN-A, IN-S, ObjectNet}. The update frequency denotes the number of training steps between each teacher model update from the student; lower frequencies (e.g., 2-10 steps) result in a teacher that closely follows the student's trajectory providing fine-grained regularization, while higher frequencies (e.g., 500-2500 steps) maintain a more stable teacher that changes less frequently, providing stronger regularization from earlier checkpoints and the initial pretrained model.

| Update Freq. | Acc.↑ (%) | | | | | | |
| --- | --- | --- | --- | --- | --- | --- | --- |
| | IN | IN-V2 | IN-R | IN-A | IN-S | ObjectNet | OOD Avg. |
| 2500 | 81.38 | 72.97 | **79.83** | 55.05 | 53.88 | 58.17 | 63.98 |
| 1000 | 81.90 | 73.27 | 79.71 | 54.93 | 54.21 | 58.26 | 64.08 |
| 500 | 82.13 | 73.53 | 79.80 | 54.72 | **54.23** | 58.30 | 64.12 |
| 100 | 82.54 | 73.92 | 79.76 | **55.09** | 54.00 | **58.31** | **64.22** |
| 50 | 82.62 | 73.84 | 79.69 | 54.96 | 53.96 | 58.12 | 64.11 |
| 10 | 82.57 | 74.01 | 79.58 | 54.96 | 53.88 | 58.00 | 64.09 |
| 5 | 82.70 | 73.98 | 79.57 | 54.81 | 53.93 | 58.07 | 64.07 |
| 2 | 82.73 | **74.12** | 79.57 | 54.51 | 53.99 | 58.09 | 64.06 |
| 1 | **82.79** | 74.11 | 79.36 | 54.89 | 53.72 | 58.23 | 64.06 |

| Update Freq. | ECE↓ | | | | | | |
| --- | --- | --- | --- | --- | --- | --- | --- |
| | IN | IN-V2 | IN-R | IN-A | IN-S | ObjectNet | OOD Avg. |
| 2500 | 0.0614 | 0.0426 | 0.0632 | **0.0839** | 0.0722 | **0.0764** | **0.0677** |
| 1000 | 0.0562 | 0.0434 | 0.0581 | 0.0908 | 0.0698 | 0.0825 | 0.0689 |
| 500 | 0.0524 | 0.0419 | 0.0548 | 0.0935 | **0.0677** | 0.0868 | 0.0689 |
| 100 | 0.0475 | 0.0410 | 0.0487 | 0.0985 | 0.0694 | 0.0920 | 0.0699 |
| 50 | 0.0481 | 0.0413 | 0.0471 | 0.0992 | 0.0713 | 0.0949 | 0.0708 |
| 10 | 0.0473 | 0.0408 | 0.0468 | 0.0983 | 0.0714 | 0.0978 | 0.0710 |
| 5 | 0.0480 | **0.0400** | 0.0451 | 0.1001 | 0.0738 | 0.0975 | 0.0713 |
| 2 | 0.0471 | 0.0409 | 0.0440 | 0.1034 | 0.0733 | 0.0990 | 0.0721 |
| 1 | **0.0446** | 0.0394 | **0.0412** | 0.1041 | 0.0784 | 0.1030 | 0.0732 |

**Ablation 4: Beta kernel shape.** The ablation on the Beta kernel shape (Table 11) evaluates the role of endpoint-aware curricula. We vary $\beta \in \{0.2, 0.5, 0.7, 0.9, 1.0, 1.5\}$. Results show that smaller $\beta$ values (0.2–0.5), which emphasize endpoints, enhance both OOD accuracy and ECE by reinforcing strong early anchoring and late solution emphasis. In contrast, larger $\beta$ values favor mid-trajectory weighting, yielding marginal ID improvements at the cost of reduced OOD gains. These findings suggest that arcsine-like weighting is particularly effective for robust finetuning. Additional kernel families are discussed in §C.5.1.

Table 11: Ablation of $\beta$ value in Beta($\beta$, $\beta$) distribution for teacher weighting in POMP distillation across ImageNet (IN) and distribution shifts. For each setting (row), accuracy (Acc.↑) is reported in %, and expected calibration error (ECE↓) in $[0, 1]$. OOD Avg. is the mean over {IN-V2, IN-R, IN-A, IN-S, ObjectNet}. The $\beta$ value controls the shape of the distribution used for sampling teacher ensemble weights. Lower $\beta$ values ($< 1$) assign higher weights to the pretrained model and early training steps, $\beta = 1$ corresponds to uniform weighting, while higher $\beta$ values ($> 1$) emphasize intermediate training steps and down-weight both the initial pretrained model and final training steps (See Figure 7).

| $\beta$ | IN | IN-V2 | IN-R | IN-A | IN-S | ObjectNet | OOD Avg. |
|---|---|---|---|---|---|---|---|
| | | | | Acc.↑ (%) | | | |
| 1.5 | **83.19** | 74.38 | 77.15 | 52.43 | 52.95 | 57.06 | 62.79 |
| 1.0 | 83.09 | **74.39** | 78.10 | 52.93 | 53.25 | 57.41 | 63.22 |
| 0.9 | 83.14 | 74.26 | 78.28 | 53.59 | 53.35 | 57.41 | 63.38 |
| 0.7 | 82.97 | 74.32 | 78.89 | 54.28 | 53.58 | 57.80 | 63.77 |
| 0.5 | 82.79 | 74.11 | 79.36 | 54.89 | 53.72 | 58.23 | 64.06 |
| 0.2 | 81.91 | 73.46 | **79.96** | **55.27** | **54.08** | **58.34** | **64.22** |

| $\beta$ | IN | IN-V2 | IN-R | IN-A | IN-S | ObjectNet | OOD Avg. |
|---|---|---|---|---|---|---|---|
| | | | | ECE↓ | | | |
| 1.5 | 0.0403 | 0.0485 | 0.0418 | 0.1433 | 0.1069 | 0.1418 | 0.0965 |
| 1.0 | 0.0407 | 0.0478 | 0.0393 | 0.1317 | 0.0957 | 0.1286 | 0.0886 |
| 0.9 | **0.0401** | 0.0477 | 0.0402 | 0.1280 | 0.0926 | 0.1262 | 0.0869 |
| 0.7 | 0.0416 | 0.0456 | **0.0388** | 0.1148 | 0.0856 | 0.1161 | 0.0802 |
| 0.5 | 0.0446 | 0.0394 | 0.0412 | 0.1041 | 0.0784 | 0.1030 | 0.0732 |
| 0.2 | 0.0548 | **0.0424** | 0.0542 | **0.0917** | **0.0670** | **0.0843** | **0.0679** |

## C Theoretical Analysis: Extended Details

This section provides the full derivations, proofs, and detailed geometric interpretations for the theoretical analysis presented in the main paper.

### C.1 Derivation of $\mathcal{L}_{\text{CL}}$

We start with the linearized multi-modal contrastive learning (MMCL) loss function, which balances positive and negative pairs across a batch, as commonly used in theoretical analyses (Ji et al., 2023; Tian, 2022; Nakada et al., 2023; Xue et al., 2024). The original formulation is given by:

$$\mathcal{L}_{\text{MMCL}}(\mathbf{W}_I, \mathbf{W}_T) = \frac{1}{2n(n-1)} \sum_i \sum_{j \neq i} (s_{ij} - s_{ii}) + \frac{1}{2n(n-1)} \sum_i \sum_{j \neq i} (s_{ji} - s_{ii}) + \frac{\rho}{2} \|\mathbf{W}_I^\top \mathbf{W}_T\|_F^2$$

where $s_{ij} = (\mathbf{W}_I \mathbf{x}_I^i)^\top (\mathbf{W}_T \mathbf{x}_T^j)$ represents the similarity score between image $i$ and text $j$.

**Step 1: Expanding the first term.**

$$\frac{1}{2n(n-1)} \sum_i \sum_{j \neq i} (s_{ij} - s_{ii}) = \frac{1}{2n(n-1)} \left[ \sum_i \sum_{j \neq i} s_{ij} - \sum_i \sum_{j \neq i} s_{ii} \right]$$

Since for each $i$, there are $(n-1)$ values of $j \neq i$, the second sub-sum simplifies:

$$= \frac{1}{2n(n-1)} \left[ \sum_i \sum_{j \neq i} s_{ij} - (n-1) \sum_i s_{ii} \right]$$

**Step 2: Expanding the second term.**

$$\frac{1}{2n(n-1)} \sum_i \sum_{j \neq i} (s_{ji} - s_{ii}) = \frac{1}{2n(n-1)} \left[ \sum_i \sum_{j \neq i} s_{ji} - \sum_i \sum_{j \neq i} s_{ii} \right]$$

Similarly, this becomes:

$$= \frac{1}{2n(n-1)} \left[ \sum_i \sum_{j \neq i} s_{ji} - (n-1) \sum_i s_{ii} \right]$$

**Step 3: Combining both terms.**   Adding the first and second terms yields:

$$\frac{1}{2n(n-1)} \left[ \sum_i \sum_{j \neq i} s_{ij} + \sum_i \sum_{j \neq i} s_{ji} - 2(n-1) \sum_i s_{ii} \right]$$

**Step 4: Analyzing negative similarity terms.**   Note that $\sum_i \sum_{j \neq i} s_{ji}$ is simply a re-indexing of $\sum_j \sum_{i \neq j} s_{ij}$, which is equivalent to $\sum_i \sum_{j \neq i} s_{ij}$. Therefore:

$$\sum_i \sum_{j \neq i} s_{ij} + \sum_i \sum_{j \neq i} s_{ji} = 2 \sum_i \sum_{j \neq i} s_{ij}$$

**Step 5: Substituting back into $\mathcal{L}_{\text{MMCL}}$.**

$$\mathcal{L}_{\text{MMCL}} = \frac{1}{2n(n-1)} \left[ 2 \sum_i \sum_{j \neq i} s_{ij} - 2(n-1) \sum_i s_{ii} \right] + \frac{\rho}{2} \|\mathbf{W}_I^\top \mathbf{W}_T\|_F^2$$

$$= \frac{1}{n(n-1)} \left[ \sum_i \sum_{j \neq i} s_{ij} - (n-1) \sum_i s_{ii} \right] + \frac{\rho}{2} \|\mathbf{W}_I^\top \mathbf{W}_T\|_F^2$$

**Step 6: Defining $\mathcal{L}_{\text{CL}}$.** We define the core contrastive alignment term as:

$$\mathcal{L}_{\text{CL}} = \sum_{i=1}^{n} \sum_{j \neq i} s_{ij} - (n-1) \sum_{i=1}^{n} s_{ii}. \tag{4}$$

And the regularization term as $R(\mathbf{W}_I, \mathbf{W}_T) = \frac{\rho}{2} \|\mathbf{W}_I^\top \mathbf{W}_T\|_F^2$. Thus, the total MMCL loss can be written as:

$$\mathcal{L}_{\text{MMCL}} = \frac{1}{n(n-1)} \mathcal{L}_{\text{CL}} + R(\mathbf{W}_I, \mathbf{W}_T)$$

## C.2 RE-FORMULATION OF THE LEAST-SQUARES OBJECTIVE

We demonstrate how the contrastive alignment term $\mathcal{L}_{\text{CL}}$ can be re-expressed as a matrix least-squares problem, which is the foundation of our theoretical analysis. Recall $\mathcal{L}_{\text{CL}} = \sum_{i=1}^{n} \sum_{j \neq i} s_{ij} - (n-1) \sum_{i=1}^{n} s_{ii}$. Let $\mathbf{H}_I = \mathbf{W}_I \mathbf{X}_I$ and $\mathbf{H}_T = \mathbf{W}_T^0 \mathbf{X}_T$. Then $s_{ij} = (\mathbf{H}_I)_i^\top (\mathbf{H}_T)_j$. Let $\mathbf{S} = \mathbf{H}_I^\top \mathbf{H}_T$. The sum of all similarities is $\mathbf{1}^\top \mathbf{S} \mathbf{1} = \sum_{i,j} s_{ij}$. The sum of diagonal similarities is $\text{Tr}(\mathbf{S}) = \sum_i s_{ii}$. Then, $\sum_{i=1}^{n} \sum_{j \neq i} s_{ij} = \sum_{i,j} s_{ij} - \sum_i s_{ii} = \mathbf{1}^\top \mathbf{S} \mathbf{1} - \text{Tr}(\mathbf{S})$. Substituting this into $\mathcal{L}_{\text{CL}}$:

$$\begin{aligned}
\mathcal{L}_{\text{CL}} &= (\mathbf{1}^\top \mathbf{S} \mathbf{1} - \text{Tr}(\mathbf{S})) - (n-1) \text{Tr}(\mathbf{S}) \\
&= \mathbf{1}^\top \mathbf{S} \mathbf{1} - n \text{Tr}(\mathbf{S}) \\
&= \text{Tr}(\mathbf{1}\mathbf{1}^\top \mathbf{S}) - n \text{Tr}(\mathbf{S}) \\
&= \text{Tr}((\mathbf{J}_n - n\mathbf{I}_n)^\top \mathbf{S}) \\
&= \text{Tr}((\mathbf{J}_n - n\mathbf{I}_n)^\top \mathbf{H}_I^\top \mathbf{H}_T) \\
&= \text{Tr}(\mathbf{H}_T (\mathbf{J}_n - n\mathbf{I}_n) \mathbf{H}_I^\top) \\
&= \text{Tr}(\mathbf{W}_T^0 \mathbf{X}_T (\mathbf{J}_n - n\mathbf{I}_n)(\mathbf{W}_I \mathbf{X}_I)^\top) \\
&= -\text{Tr}\left((\mathbf{W}_T^0 \mathbf{X}_T (n\mathbf{I}_n - \mathbf{J}_n))^\top \mathbf{W}_I \mathbf{X}_I\right).
\end{aligned}$$

Let $\mathbf{Y}_{\text{FT}} = \mathbf{W}_T^0 \mathbf{X}_T (n\mathbf{I}_n - \mathbf{J}_n)$, as defined in Definition 3.1. Then $\mathcal{L}_{\text{CL}} = -\text{Tr}(\mathbf{Y}_{\text{FT}}^\top \mathbf{W}_I \mathbf{X}_I)$. Ignoring constant terms, minimizing $-\text{Tr}(\mathbf{Y}_{\text{FT}}^\top \mathbf{W}_I \mathbf{X}_I)$ is equivalent to minimizing $\frac{1}{2} \|\mathbf{W}_I \mathbf{X}_I - \mathbf{Y}_{\text{FT}}\|_F^2$. This can be shown by expanding the Frobenius norm:

$$\begin{aligned}
\frac{1}{2} \|\mathbf{W}_I \mathbf{X}_I - \mathbf{Y}_{\text{FT}}\|_F^2 &= \frac{1}{2} \text{Tr}((\mathbf{W}_I \mathbf{X}_I - \mathbf{Y}_{\text{FT}})^\top (\mathbf{W}_I \mathbf{X}_I - \mathbf{Y}_{\text{FT}})) \\
&= \frac{1}{2} \text{Tr}(\mathbf{X}_I^\top \mathbf{W}_I^\top \mathbf{W}_I \mathbf{X}_I - \mathbf{X}_I^\top \mathbf{W}_I^\top \mathbf{Y}_{\text{FT}} - \mathbf{Y}_{\text{FT}}^\top \mathbf{W}_I \mathbf{X}_I + \mathbf{Y}_{\text{FT}}^\top \mathbf{Y}_{\text{FT}}) \\
&= \frac{1}{2} \|\mathbf{W}_I \mathbf{X}_I\|_F^2 - \text{Tr}(\mathbf{Y}_{\text{FT}}^\top \mathbf{W}_I \mathbf{X}_I) + \frac{1}{2} \|\mathbf{Y}_{\text{FT}}\|_F^2.
\end{aligned}$$

Minimizing this expression with respect to $\mathbf{W}_I$ is equivalent to minimizing $\frac{1}{2} \|\mathbf{W}_I \mathbf{X}_I\|_F^2 - \text{Tr}(\mathbf{Y}_{\text{FT}}^\top \mathbf{W}_I \mathbf{X}_I)$, as the term $\frac{1}{2} \|\mathbf{Y}_{\text{FT}}\|_F^2$ is constant for a fixed teacher $\mathbf{W}_T^0$ and dataset $\mathbf{X}_T$. The term $\frac{1}{2} \|\mathbf{W}_I \mathbf{X}_I\|_F^2$ acts as a data-dependent regularization. When we optimize for $\mathbf{W}_I$, it naturally arises.

## C.3 UNIFIED FRAMEWORK FOR CONTRASTIVE FINETUNING: PROOFS AND DETAILS

Our proofs rely on the following lemma for gradient descent on a matrix quadratic program.

*Lemma* C.1 (Gradient Descent for Matrix Quadratic Programs). Let $\mathcal{Q} : \mathbb{R}^{p \times d} \to \mathbb{R}^{p \times d}$ be a positive semi-definite (PSD) linear operator and $\mathbf{P} \in \mathbb{R}^{p \times d}$. Consider the quadratic objective

$$f(\mathbf{W}) = \frac{1}{2} \langle \mathbf{W}, \mathcal{Q}(\mathbf{W}) \rangle_F - \langle \mathbf{P}, \mathbf{W} \rangle_F, \tag{5}$$

where $\langle \cdot, \cdot \rangle_F$ denotes the Frobenius inner product. Let $\|\mathcal{Q}\|_{\text{op}}$ denote the operator norm of $\mathcal{Q}$ induced by the Frobenius norm. If $\mathbf{P} \in \text{Range}(\mathcal{Q})$, then gradient descent initialized at $\mathbf{W}_0$ with step size $\gamma \in (0, 2/\|\mathcal{Q}\|_{\text{op}})$ converges to

$$\mathbf{W}_\infty = (\mathbf{I} - \Pi_\mathcal{Q})(\mathbf{W}_0) + \mathcal{Q}^+(\mathbf{P}), \tag{6}$$

where $\Pi_\mathcal{Q}$ is the orthogonal projector onto $\text{Range}(\mathcal{Q})$ and $\mathcal{Q}^+$ is the Moore-Penrose pseudoinverse of $\mathcal{Q}$.

*Proof.* The gradient of $f$ is given by $\nabla f(\mathbf{W}) = \mathcal{Q}(\mathbf{W}) - \mathbf{P}$, yielding the gradient descent update

$$\mathbf{W}_{t+1} = \mathbf{W}_t - \gamma(\mathcal{Q}(\mathbf{W}_t) - \mathbf{P}). \tag{7}$$

Since $\mathcal{Q}$ is PSD, we have the orthogonal decomposition

$$\mathbb{R}^{p \times d} = \mathrm{Range}(\mathcal{Q}) \oplus \mathrm{Null}(\mathcal{Q}). \tag{8}$$

Let $\Pi_{\mathcal{Q}}$ and $\Pi_{\mathcal{Q}^\perp} = \mathbf{I} - \Pi_{\mathcal{Q}}$ denote the orthogonal projectors onto the range and null space of $\mathcal{Q}$, respectively.

**Analysis of the null space component.** Projecting the gradient descent update onto $\mathrm{Null}(\mathcal{Q})$ yields

$$\Pi_{\mathcal{Q}^\perp}(\mathbf{W}_{t+1}) = \Pi_{\mathcal{Q}^\perp}(\mathbf{W}_t) - \gamma\Pi_{\mathcal{Q}^\perp}(\mathcal{Q}(\mathbf{W}_t)) + \gamma\Pi_{\mathcal{Q}^\perp}(\mathbf{P})$$
$$= \Pi_{\mathcal{Q}^\perp}(\mathbf{W}_t),$$

where we used that $\mathcal{Q}(\mathbf{W}_t) \in \mathrm{Range}(\mathcal{Q})$ implies $\Pi_{\mathcal{Q}^\perp}(\mathcal{Q}(\mathbf{W}_t)) = \mathbf{0}$, and our assumption $\mathbf{P} \in \mathrm{Range}(\mathcal{Q})$ implies $\Pi_{\mathcal{Q}^\perp}(\mathbf{P}) = \mathbf{0}$. Thus, the null space component remains invariant throughout the optimization:

$$\Pi_{\mathcal{Q}^\perp}(\mathbf{W}_t) = \Pi_{\mathcal{Q}^\perp}(\mathbf{W}_0) \quad \forall t \geq 0. \tag{9}$$

**Analysis of the range component.** Let $\mathbf{W}_t' = \Pi_{\mathcal{Q}}(\mathbf{W}_t)$ denote the projection onto $\mathrm{Range}(\mathcal{Q})$. The dynamics of this component follow

$$\mathbf{W}_{t+1}' = (\mathbf{I} - \gamma\mathcal{Q})\mathbf{W}_t' + \gamma\mathbf{P}. \tag{10}$$

The restriction of $\mathcal{Q}$ to its range, denoted $\mathcal{Q}_R : \mathrm{Range}(\mathcal{Q}) \to \mathrm{Range}(\mathcal{Q})$, is positive definite (since for any non-zero $x \in \mathrm{Range}(\mathcal{Q})$, we must have $\mathcal{Q}(x) \neq 0$, otherwise $x$ would be in $\mathrm{Null}(\mathcal{Q})$). For $\gamma \in (0, 2/\|\mathcal{Q}\|_{\mathrm{op}})$, the operator $\mathbf{I} - \gamma\mathcal{Q}_R$ has spectral radius less than 1, making it a contraction mapping. By the Banach fixed-point theorem, the sequence $\{\mathbf{W}_t'\}$ converges to the unique fixed point $\mathbf{W}_\infty' \in \mathrm{Range}(\mathcal{Q})$ satisfying

$$\mathbf{W}_\infty' = (\mathbf{I} - \gamma\mathcal{Q})\mathbf{W}_\infty' + \gamma\mathbf{P}. \tag{11}$$

Rearranging gives $\mathcal{Q}(\mathbf{W}_\infty') = \mathbf{P}$, which has the unique solution $\mathbf{W}_\infty' = \mathcal{Q}^+(\mathbf{P})$ in $\mathrm{Range}(\mathcal{Q})$.

**Synthesis.** Combining the analyses of both components, we obtain

$$\mathbf{W}_\infty = \lim_{t \to \infty} \left(\Pi_{\mathcal{Q}^\perp}(\mathbf{W}_t) + \mathbf{W}_t'\right)$$
$$= \Pi_{\mathcal{Q}^\perp}(\mathbf{W}_0) + \mathcal{Q}^+(\mathbf{P})$$
$$= (\mathbf{I} - \Pi_{\mathcal{Q}})(\mathbf{W}_0) + \mathcal{Q}^+(\mathbf{P}),$$

completing the proof. $\qquad\square$

*Theorem* C.2 (Unified Framework for Contrastive Finetuning Solutions (Full Proof)). Let $\mathcal{P}_I \coloneqq \mathbf{X}_I(\mathbf{X}_I^\top\mathbf{X}_I)^+\mathbf{X}_I^\top$ denote the orthogonal projection onto the subspace spanned by the finetuning data $\mathbf{X}_I$. Consider the general finetuning objective:

$$\mathcal{L}(\mathbf{W}_I) = \frac{1}{2}\|\mathbf{W}_I\mathbf{X}_I - \mathbf{Y}_{\mathrm{FT}}\|_F^2 + \mathcal{R}(\mathbf{W}_I) \tag{12}$$

where $\mathcal{R}(\mathbf{W}_I)$ represents different regularization strategies. Gradient descent initialized at $\mathbf{W}_I^0$ with sufficiently small learning rate converges to the following solutions:

| Strategy | $\mathcal{R}(\mathbf{W}_I)$ | Solution |
|---|---|---|
| Direct Finetuning | $0$ | $\mathbf{W}_{\mathrm{FT}} = \mathbf{W}_I^0(\mathbf{I} - \mathcal{P}_I) + \mathbf{Y}_{\mathrm{FT}}\mathbf{X}_I^\top(\mathbf{X}_I\mathbf{X}_I^\top)^+$ |
| $L_2$ Regularization (L2-SP (Li et al., 2018)) | $\frac{\lambda}{2}\|\mathbf{W}_I - \mathbf{W}_I^0\|_F^2$ | $\mathbf{W}_{L_2} = (\mathbf{Y}_{\mathrm{FT}}\mathbf{X}_I^\top + \lambda\mathbf{W}_I^0)(\mathbf{X}_I\mathbf{X}_I^\top + \lambda\mathbf{I})^{-1}$ |
| Self-Distillation (SD (Furlanello et al., 2018)) | $\frac{\lambda}{2}\|\mathbf{W}_I\mathbf{X}_I - \mathbf{W}_I^0\mathbf{X}_I\|_F^2$ | $\mathbf{W}_{SD} = \mathbf{W}_I^0(\mathbf{I} - \frac{1}{1+\lambda}\mathcal{P}_I) + \frac{1}{1+\lambda}\mathbf{Y}_{\mathrm{FT}}\mathbf{X}_I^\top(\mathbf{X}_I\mathbf{X}_I^\top)^+$ |

Here, $^+$ denotes the Moore-Penrose pseudoinverse and $\lambda > 0$ is the regularization parameter.

*Proof.* Let $\mathbf{C}_I = \mathbf{X}_I\mathbf{X}_I^\top$.

**Direct Finetuning.** The objective is $\mathcal{L}(\mathbf{W}_I) = \frac{1}{2} \|\mathbf{W}_I \mathbf{X}_I - \mathbf{Y}_{\text{FT}}\|_{\text{F}}^2$. We rewrite this in the quadratic form of Lemma C.1:

$$\mathcal{L}(\mathbf{W}_I) = \frac{1}{2} \langle \mathbf{W}_I \mathbf{X}_I - \mathbf{Y}_{\text{FT}}, \mathbf{W}_I \mathbf{X}_I - \mathbf{Y}_{\text{FT}} \rangle_F$$

$$= \frac{1}{2} \langle \mathbf{W}_I, \mathbf{W}_I (\mathbf{X}_I \mathbf{X}_I^\top) \rangle_F - \langle \mathbf{W}_I, \mathbf{Y}_{\text{FT}} \mathbf{X}_I^\top \rangle_F + \frac{1}{2} \|\mathbf{Y}_{\text{FT}}\|_{\text{F}}^2$$

$$= \frac{1}{2} \langle \mathbf{W}_I, \mathbf{W}_I \mathbf{C}_I \rangle_F - \langle \mathbf{W}_I, \mathbf{Y}_{\text{FT}} \mathbf{X}_I^\top \rangle_F + \text{const.}$$

This matches the form $f(\mathbf{W}) = \frac{1}{2} \langle \mathbf{W}, \mathcal{Q}(\mathbf{W}) \rangle_F - \langle \mathbf{P}, \mathbf{W} \rangle_F$ with $\mathcal{Q}(\mathbf{W}) = \mathbf{W}\mathbf{C}_I$ and $\mathbf{P} = \mathbf{Y}_{\text{FT}} \mathbf{X}_I^\top$.

The operator $\mathcal{Q}$ is linear and positive semi-definite, as $\langle \mathbf{W}_I, \mathcal{Q}(\mathbf{W}_I) \rangle_F = \|\mathbf{W}_I \mathbf{X}_I\|_{\text{F}}^2 \geq 0$. The condition $\mathbf{P} \in \text{Range}(\mathcal{Q})$ holds because the rows of $\mathbf{P} = \mathbf{Y}_{\text{FT}} \mathbf{X}_I^\top$ are linear combinations of the rows of $\mathbf{X}_I^\top$, which form the row space of $\mathbf{C}_I$.

By Lemma C.1, gradient descent converges to $\mathbf{W}_\infty = \Pi_{\mathcal{Q}^\perp}(\mathbf{W}_I^0) + \mathcal{Q}^+(\mathbf{P})$.

1. **Null Space Component:** The null space of $\mathcal{Q}$ consists of matrices $\mathbf{A}$ such that $\mathcal{Q}(\mathbf{A}) = \mathbf{A}\mathbf{C}_I = \mathbf{0}$. This holds if and only if the rows of $\mathbf{A}$ are in the null space of $\mathbf{C}_I$. The orthogonal projector onto this component of the initial matrix $\mathbf{W}_I^0$ is $\Pi_{\mathcal{Q}^\perp}(\mathbf{W}_I^0) = \mathbf{W}_I^0(\mathbf{I} - \mathcal{P}_I)$, where $\mathcal{P}_I = \mathbf{C}_I \mathbf{C}_I^+$ is the projector onto the row space of $\mathbf{X}_I$. This component is preserved.

2. **Range Component:** The pseudoinverse $\mathcal{Q}^+$ finds the minimum Frobenius norm solution to $\mathcal{Q}(\mathbf{W}) = \mathbf{P}$ that lies in $\text{Range}(\mathcal{Q})$. This is the solution to $\mathbf{W}\mathbf{C}_I = \mathbf{Y}_{\text{FT}}\mathbf{X}_I^\top$, which is $\mathcal{Q}^+(\mathbf{P}) = (\mathbf{Y}_{\text{FT}}\mathbf{X}_I^\top)\mathbf{C}_I^+$.

Combining the components gives the final solution:

$$\mathbf{W}_{\text{FT}} = \mathbf{W}_I^0(\mathbf{I} - \mathcal{P}_I) + \mathbf{Y}_{\text{FT}}\mathbf{X}_I^\top(\mathbf{X}_I\mathbf{X}_I^\top)^+.$$

$L_2$ **Regularization.** The objective $\mathcal{L}(\mathbf{W}_I) = \frac{1}{2} \|\mathbf{W}_I\mathbf{X}_I - \mathbf{Y}_{\text{FT}}\|_{\text{F}}^2 + \frac{\lambda}{2} \|\mathbf{W}_I - \mathbf{W}_I^0\|_{\text{F}}^2$. This objective is strongly convex for $\lambda > 0$. The unique minimizer is found by setting the gradient to zero:

$$\nabla_{\mathbf{W}_I}\mathcal{L} = (\mathbf{W}_I\mathbf{X}_I - \mathbf{Y}_{\text{FT}})\mathbf{X}_I^\top + \lambda(\mathbf{W}_I - \mathbf{W}_I^0) = 0$$

$$\mathbf{W}_I\mathbf{X}_I\mathbf{X}_I^\top + \lambda\mathbf{W}_I = \mathbf{Y}_{\text{FT}}\mathbf{X}_I^\top + \lambda\mathbf{W}_I^0$$

$$\mathbf{W}_I(\mathbf{X}_I\mathbf{X}_I^\top + \lambda\mathbf{I}) = \mathbf{Y}_{\text{FT}}\mathbf{X}_I^\top + \lambda\mathbf{W}_I^0$$

Since $\mathbf{X}_I\mathbf{X}_I^\top$ is PSD, the matrix $(\mathbf{X}_I\mathbf{X}_I^\top + \lambda\mathbf{I})$ is positive definite and thus invertible. The solution is:

$$\mathbf{W}_{L_2} = (\mathbf{Y}_{\text{FT}}\mathbf{X}_I^\top + \lambda\mathbf{W}_I^0)(\mathbf{X}_I\mathbf{X}_I^\top + \lambda\mathbf{I})^{-1}.$$

A more detailed analysis of the limit behavior of this solution as $\lambda \to 0$ and $\lambda \to \infty$ is provided in §C.4.

**Self-Distillation.** The objective is $\mathcal{L}(\mathbf{W}_I) = \frac{1}{2} \|\mathbf{W}_I\mathbf{X}_I - \mathbf{Y}_{\text{FT}}\|_{\text{F}}^2 + \frac{\lambda}{2} \|\mathbf{W}_I\mathbf{X}_I - \mathbf{W}_I^0\mathbf{X}_I\|_{\text{F}}^2$. Expanding and grouping terms reveals the quadratic structure:

$$\mathcal{L}(\mathbf{W}_I) = \frac{1}{2} \|\mathbf{W}_I\mathbf{X}_I\|_{\text{F}}^2 - \text{Tr}(\mathbf{Y}_{\text{FT}}^\top\mathbf{W}_I\mathbf{X}_I) + \frac{1}{2} \|\mathbf{Y}_{\text{FT}}\|_{\text{F}}^2$$

$$+ \frac{\lambda}{2} \|\mathbf{W}_I\mathbf{X}_I\|_{\text{F}}^2 - \lambda \text{Tr}((\mathbf{W}_I^0\mathbf{X}_I)^\top\mathbf{W}_I\mathbf{X}_I) + \frac{\lambda}{2} \|\mathbf{W}_I^0\mathbf{X}_I\|_{\text{F}}^2$$

$$= \frac{1+\lambda}{2} \|\mathbf{W}_I\mathbf{X}_I\|_{\text{F}}^2 - \text{Tr}((\mathbf{Y}_{\text{FT}}^\top + \lambda(\mathbf{W}_I^0\mathbf{X}_I)^\top)\mathbf{W}_I\mathbf{X}_I) + \text{const.}$$

$$= \frac{1+\lambda}{2} \langle \mathbf{W}_I, \mathbf{W}_I\mathbf{C}_I \rangle_F - \langle \mathbf{W}_I, \mathbf{Y}_{\text{FT}}\mathbf{X}_I^\top + \lambda\mathbf{W}_I^0\mathbf{C}_I \rangle_F + \text{const.}$$

This matches the form of Lemma C.1 with $\mathcal{Q}_{SD}(\mathbf{W}_I) = (1+\lambda)\mathbf{W}_I\mathbf{C}_I$ and $\mathbf{P}_{SD} = \mathbf{Y}_{\text{FT}}\mathbf{X}_I^\top + \lambda\mathbf{W}_I^0\mathbf{C}_I$.

The operator $\mathcal{Q}_{SD}$ is PSD. Its range and null space are identical to those of $\mathcal{Q}$ from the Direct Finetuning case. The terms $\mathbf{Y}_{\text{FT}}\mathbf{X}_I^\top$ and $\lambda\mathbf{W}_I^0\mathbf{C}_I$ are both in $\text{Range}(\mathcal{Q}_{SD})$ (as shown before for $\mathbf{Y}_{\text{FT}}\mathbf{X}_I^\top$, and $\mathbf{W}_I^0\mathbf{C}_I$ by definition). Thus, their sum $\mathbf{P}_{SD}$ is also in the range.

We apply Lemma C.1 to find the limit $\mathbf{W}_\infty = \Pi_{\mathcal{Q}_{SD}^\perp}(\mathbf{W}_I^0) + \mathcal{Q}_{SD}^+(\mathbf{P}_{SD})$.

1. **Null Space Component:** $\text{Null}(\mathcal{Q}_{SD}) = \text{Null}(\mathcal{Q})$, so the invariant component is again $\Pi_{\mathcal{Q}_{SD}^\perp}(\mathbf{W}_I^0) = \mathbf{W}_I^0(\mathbf{I} - \mathcal{P}_I)$.

2. **Range Component:** The pseudoinverse is $\mathcal{Q}_{SD}^+ = \frac{1}{1+\lambda}\mathcal{Q}^+$, where $\mathcal{Q}^+$ corresponds to the direct finetuning case. Applying it to $\mathbf{P}_{SD}$:

$$
\begin{aligned}
\mathcal{Q}_{SD}^+(\mathbf{P}_{SD}) &= \frac{1}{1+\lambda}\mathcal{Q}^+\left(\mathbf{Y}_{\text{FT}}\mathbf{X}_I^\top + \lambda\mathbf{W}_I^0\mathbf{C}_I\right) \\
&= \frac{1}{1+\lambda}\left((\mathbf{Y}_{\text{FT}}\mathbf{X}_I^\top)\mathbf{C}_I^+ + \lambda\mathcal{Q}^+(\mathcal{Q}(\mathbf{W}_I^0))\right) \\
&= \frac{1}{1+\lambda}\left((\mathbf{Y}_{\text{FT}}\mathbf{X}_I^\top)\mathbf{C}_I^+ + \lambda\Pi_\mathcal{Q}(\mathbf{W}_I^0)\right) \\
&= \frac{1}{1+\lambda}\left(\mathbf{Y}_{\text{FT}}\mathbf{X}_I^\top(\mathbf{X}_I\mathbf{X}_I^\top)^+ + \lambda\mathbf{W}_I^0\mathcal{P}_I\right).
\end{aligned}
$$

Combining the components for the final solution $\mathbf{W}_{SD}$:

$$
\begin{aligned}
\mathbf{W}_{SD} &= \mathbf{W}_I^0(\mathbf{I} - \mathcal{P}_I) + \frac{\lambda}{1+\lambda}\mathbf{W}_I^0\mathcal{P}_I + \frac{1}{1+\lambda}\mathbf{Y}_{\text{FT}}\mathbf{X}_I^\top(\mathbf{X}_I\mathbf{X}_I^\top)^+ \\
&= \mathbf{W}_I^0\left(\mathbf{I} - \mathcal{P}_I + \frac{\lambda}{1+\lambda}\mathcal{P}_I\right) + \frac{1}{1+\lambda}\mathbf{Y}_{\text{FT}}\mathbf{X}_I^\top(\mathbf{X}_I\mathbf{X}_I^\top)^+ \\
&= \mathbf{W}_I^0\left(\mathbf{I} - \frac{1}{1+\lambda}\mathcal{P}_I\right) + \frac{1}{1+\lambda}\mathbf{Y}_{\text{FT}}\mathbf{X}_I^\top(\mathbf{X}_I\mathbf{X}_I^\top)^+.
\end{aligned}
$$

This completes the proof. $\qquad\square$

### C.4 GEOMETRIC INTERPRETATION OF SOLUTIONS

The closed-form solutions presented in Theorem C.2 provide a geometric understanding of how different finetuning strategies modify pretrained representations. We decompose the solution for $\mathbf{W}_I$ into components acting on the subspace spanned by the finetuning data $\mathbf{X}_I$ (parallel component) and its orthogonal complement (orthogonal component).

**Direct Finetuning.** The solution $\mathbf{W}_{\text{FT}}$ is a sum of two orthogonal parts: **(1)** $\mathbf{W}_I^0(\mathbf{I} - \mathcal{P}_I)$: This is the projection of the pretrained weights onto the orthogonal complement of the finetuning data subspace ($\text{Null}(\mathbf{X}_I^\top)$). This component preserves the action of $\mathbf{W}_I^0$ on data vectors orthogonal to the finetuning examples. **(2)** $\mathbf{Y}_{\text{FT}}\mathbf{X}_I^\top(\mathbf{X}_I\mathbf{X}_I^\top)^+$: This is the minimum-norm solution that fits the new contrastive task within the finetuning data subspace. This component lies entirely within the range of $\mathbf{X}_I^\top$.

*Interpretation:* Direct finetuning completely replaces (forgets) any pretrained knowledge related to features present in the finetuning data, substituting it with the new task-specific solution. It only preserves knowledge in directions entirely unrelated to the finetuning examples.

$L_2$ **Regularization.** The solution $\mathbf{W}_{L_2}$ is the standard matrix ridge regression solution. It creates a complex blend of the new task solution and the initial weights. There is no clean separation of orthogonal and parallel components as in direct finetuning or self-distillation. The key insight is that $L_2$ regularization modifies the data covariance matrix $\mathbf{X}_I\mathbf{X}_I^\top$ by adding $\lambda\mathbf{I}$, which acts as a *ridge* that prevents overfitting by shrinking the solution along all eigendirections of the data. Unlike direct finetuning and self-distillation, which primarily modify weights in the subspace spanned by $\mathbf{X}_I$, $L_2$ regularization affects all directions in the weight space, blending the old and new across the entire parameter space.

**Detailed Analysis of the $L_2$ Regularization Solution.** The solution for $L_2$ regularization is given by:

$$\mathbf{W}_{L_2} = \left(\mathbf{Y}_{\text{FT}}\mathbf{X}_I^\top + \lambda\mathbf{W}_I^0\right)\left(\mathbf{X}_I\mathbf{X}_I^\top + \lambda\mathbf{I}\right)^{-1}.$$

To analyze its behavior, we consider the eigendecomposition of the data covariance matrix $\mathbf{C}_I := \mathbf{X}_I\mathbf{X}_I^\top$. Since $\mathbf{C}_I$ is a real, symmetric, positive semi-definite (PSD) matrix, it has an eigendecomposition $\mathbf{C}_I = \mathbf{U}\mathbf{\Lambda}\mathbf{U}^\top$, where $\mathbf{U}$ is an orthogonal matrix of eigenvectors and $\mathbf{\Lambda}$ is a diagonal matrix of non-negative eigenvalues. Using this decomposition, the inverse term in the solution becomes:

$$(\mathbf{C}_I + \lambda\mathbf{I})^{-1} = (\mathbf{U}\mathbf{\Lambda}\mathbf{U}^\top + \lambda\mathbf{U}\mathbf{U}^\top)^{-1} = (\mathbf{U}(\mathbf{\Lambda} + \lambda\mathbf{I})\mathbf{U}^\top)^{-1} = \mathbf{U}(\mathbf{\Lambda} + \lambda\mathbf{I})^{-1}\mathbf{U}^\top.$$

The matrix $(\mathbf{\Lambda} + \lambda\mathbf{I})$ is diagonal with entries $\lambda_k + \lambda$, so its inverse has entries $1/(\lambda_k + \lambda)$.

**Analysis of the Limit as $\lambda \to 0$.** Let $r = \text{rank}(\mathbf{C}_I)$. We partition the eigenvectors $\mathbf{U}$ and eigenvalues $\mathbf{\Lambda}$ into components corresponding to non-zero and zero eigenvalues. Let $\mathbf{U}_r \in \mathbb{R}^{d_I \times r}$ contain eigenvectors for $r$ positive eigenvalues ($\mathbf{\Lambda}_r$), and $\mathbf{U}_0 \in \mathbb{R}^{d_I \times (d_I - r)}$ for zero eigenvalues. The projectors onto the range and null space of $\mathbf{C}_I$ are $\mathcal{P}_{\text{range}} = \mathbf{U}_r\mathbf{U}_r^\top$ and $\mathcal{P}_{\text{null}} = \mathbf{U}_0\mathbf{U}_0^\top$, respectively. Note that $\mathcal{P}_{\text{range}} = \mathcal{P}_I$ and $\mathcal{P}_{\text{null}} = \mathbf{I} - \mathcal{P}_I$.

The inverse term can be split:

$$(\mathbf{C}_I + \lambda\mathbf{I})^{-1} = \mathbf{U}_r(\mathbf{\Lambda}_r + \lambda\mathbf{I}_r)^{-1}\mathbf{U}_r^\top + \frac{1}{\lambda}\mathbf{U}_0\mathbf{U}_0^\top.$$

Substituting this back into $\mathbf{W}_{L_2}$:

$$\mathbf{W}_{L_2} = \left(\mathbf{Y}_{\text{FT}}\mathbf{X}_I^\top + \lambda\mathbf{W}_I^0\right)\left[\mathbf{U}_r(\mathbf{\Lambda}_r + \lambda\mathbf{I}_r)^{-1}\mathbf{U}_r^\top + \frac{1}{\lambda}\mathbf{U}_0\mathbf{U}_0^\top\right]$$

$$= \underbrace{\left(\mathbf{Y}_{\text{FT}}\mathbf{X}_I^\top + \lambda\mathbf{W}_I^0\right)\mathbf{U}_r(\mathbf{\Lambda}_r + \lambda\mathbf{I}_r)^{-1}\mathbf{U}_r^\top}_{\text{Term 1}} + \underbrace{\left(\mathbf{Y}_{\text{FT}}\mathbf{X}_I^\top + \lambda\mathbf{W}_I^0\right)\frac{1}{\lambda}\mathbf{U}_0\mathbf{U}_0^\top}_{\text{Term 2}}.$$

For Term 2, since $\mathbf{X}_I^\top\mathbf{U}_0 = \mathbf{0}$ (columns of $\mathbf{U}_0$ are in the null space of $\mathbf{C}_I$), it simplifies to:

$$\text{Term 2} = \frac{1}{\lambda}\mathbf{Y}_{\text{FT}}\underbrace{\mathbf{X}_I^\top\mathbf{U}_0}_{\mathbf{0}}\mathbf{U}_0^\top + \mathbf{W}_I^0\mathbf{U}_0\mathbf{U}_0^\top = \mathbf{W}_I^0\mathcal{P}_{\text{null}} = \mathbf{W}_I^0(\mathbf{I} - \mathcal{P}_I).$$

As $\lambda \to 0$, Term 1 converges to:

$$\lim_{\lambda \to 0} \text{Term 1} = \left(\mathbf{Y}_{\text{FT}}\mathbf{X}_I^\top\right)\mathbf{U}_r\mathbf{\Lambda}_r^{-1}\mathbf{U}_r^\top = \mathbf{Y}_{\text{FT}}\mathbf{X}_I^\top\mathbf{C}_I^+,$$

where $\mathbf{C}_I^+ = \mathbf{U}_r\mathbf{\Lambda}_r^{-1}\mathbf{U}_r^\top$ is the Moore-Penrose pseudoinverse of $\mathbf{C}_I$. Combining the limits of both terms, we get:

$$\lim_{\lambda \to 0} \mathbf{W}_{L_2} = \mathbf{Y}_{\text{FT}}\mathbf{X}_I^\top(\mathbf{X}_I\mathbf{X}_I^\top)^+ + \mathbf{W}_I^0(\mathbf{I} - \mathcal{P}_I).$$

This is precisely the direct finetuning solution, $\mathbf{W}_{\text{FT}}$.

**Analysis of the Limit as $\lambda \to \infty$.** For the limit as $\lambda \to \infty$, we factor out $\lambda$:

$$\mathbf{W}_{L_2} = \left(\mathbf{Y}_{\text{FT}}\mathbf{X}_I^\top + \lambda\mathbf{W}_I^0\right)\frac{1}{\lambda}\left(\frac{1}{\lambda}\mathbf{C}_I + \mathbf{I}\right)^{-1}$$

$$= \left(\frac{1}{\lambda}\mathbf{Y}_{\text{FT}}\mathbf{X}_I^\top + \mathbf{W}_I^0\right)\left(\frac{1}{\lambda}\mathbf{C}_I + \mathbf{I}\right)^{-1}.$$

As $\lambda \to \infty$, the term $\frac{1}{\lambda} \to 0$. Therefore, the expression converges to:

$$\lim_{\lambda \to \infty} \mathbf{W}_{L_2} = \left(\mathbf{0} + \mathbf{W}_I^0\right)\left(\mathbf{0} + \mathbf{I}\right)^{-1} = \mathbf{W}_I^0.$$

Thus, the regularization parameter $\lambda$ smoothly interpolates the solution between two meaningful extremes: pure task adaptation and pure preservation of pretrained weights.

**Self-Distillation.** The solution $\mathbf{W}_{SD}$ provides the most sophisticated and effective compromise. We can rewrite it to reveal its structure:

$$
\begin{aligned}
\mathbf{W}_{SD} &= \mathbf{W}_I^0 - \frac{1}{1+\lambda}\mathbf{W}_I^0\mathcal{P}_I + \frac{1}{1+\lambda}\mathbf{Y}_{\mathrm{FT}}\mathbf{X}_I^\top(\mathbf{X}_I\mathbf{X}_I^\top)^+ \\
&= \mathbf{W}_I^0(\mathbf{I}-\mathcal{P}_I) + \mathbf{W}_I^0\mathcal{P}_I - \frac{1}{1+\lambda}\mathbf{W}_I^0\mathcal{P}_I + \frac{1}{1+\lambda}\left(\mathbf{Y}_{\mathrm{FT}}\mathbf{X}_I^\top(\mathbf{X}_I\mathbf{X}_I^\top)^+\right) \\
&= \underbrace{\mathbf{W}_I^0(\mathbf{I}-\mathcal{P}_I)}_{\substack{\text{Component orthogonal to finetuning data} \\ \textbf{(Preserved)}}} + \underbrace{\frac{\lambda}{1+\lambda}\left(\mathbf{W}_I^0\mathcal{P}_I\right) + \frac{1}{1+\lambda}\left(\mathbf{Y}_{\mathrm{FT}}\mathbf{X}_I^\top(\mathbf{X}_I\mathbf{X}_I^\top)^+\right)}_{\substack{\text{Component within finetuning data subspace} \\ \textbf{(Convex Combination)}}}
\end{aligned}
$$

*Interpretation*: Self-Distillation operates with surgical precision: 1. **Outside the finetuning subspace**, it acts as an identity function, preserving the components of the pretrained model that are irrelevant to the new task. 2. **Inside the finetuning subspace**, it does not discard the pretrained knowledge. Instead, it computes a convex combination of the projected pretrained weights and the optimal solution for the new contrastive task. The hyperparameter $\lambda$ smoothly controls this trade-off. This demonstrates that Self-Distillation achieves a "best of both worlds" scenario: preserving general capabilities while adapting to new information where necessary.

## C.5 Self-Distillation with a Dynamic Teacher: WMA Details and Convergence

We extend the analysis of static self-distillation to a dynamic teacher, specifically a Weighted Moving Average (WMA) teacher, which adapts its regularization throughout training. This section provides the detailed definitions, dynamics, and convergence proofs.

**Definition C.3** (SD–WMA Objective (Repeated from Main Text)). At step $t$, the student weights $\mathbf{W}_I^t$ solve

$$
\mathcal{L}_{\text{SD-WMA}}(\mathbf{W}_I) = \frac{1}{2}\left\|\mathbf{W}_I\mathbf{X}_I - \mathbf{Y}_{\mathrm{FT}}\right\|_{\mathrm{F}}^2 + \frac{\lambda}{2}\left\|\mathbf{W}_I\mathbf{X}_I - \mathbf{W}_{\mathrm{Teacher}}^{t-1}\mathbf{X}_I\right\|_{\mathrm{F}}^2, \qquad \text{initialized from } \mathbf{W}_I^{t-1}. \tag{13}
$$

**Definition C.4** (Weighted Moving Average (WMA) Teacher (Repeated from Main Text)). Let the normalized time grid be

$$
\tau_k = \frac{k+c_1}{T+c_2} \in (0,1), \qquad c_1, c_2 > 0.
$$

Choose any nonnegative *weighting kernel* $\kappa : [0,1] \to \mathbb{R}_{\geq 0}$ and define unnormalized weights $\alpha_k = \kappa(\tau_k)$ The *online* normalization and teacher recursion are

$$
\omega_t = \frac{\alpha_t}{\sum_{j=0}^{t}\alpha_j}, \qquad \mathbf{W}_{\mathrm{Teacher}}^t = (1-\omega_t)\mathbf{W}_{\mathrm{Teacher}}^{t-1} + \omega_t\mathbf{W}_I^t, \qquad \mathbf{W}_{\mathrm{Teacher}}^0 = \mathbf{W}_I^0. \tag{14}
$$

*Remark* C.5 (Teacher as a normalized history average). Unrolling equation 14 yields a normalized convex average of the student's history:

$$
\mathbf{W}_{\mathrm{Teacher}}^t = \sum_{k=0}^{t} \underbrace{\frac{\alpha_k}{\sum_{j=0}^{t}\alpha_j}}_{\omega_{k|t}} \mathbf{W}_I^k, \qquad \omega_{k|t} \geq 0, \quad \sum_{k=0}^{t}\omega_{k|t} = 1.
$$

Thus the teacher is an *expectation* with respect to the discrete distribution $\mathrm{Categorical}(\omega_{0|t}, \dots, \omega_{t|t})$: $\mathbf{W}_{\mathrm{Teacher}}^t = \mathbb{E}_{K \sim \omega_{\cdot|t}}[\mathbf{W}_I^K]$.

### C.5.1 WMA vs. EMA Teachers

This section contrasts the proposed *Weighted Moving Average* (WMA) teacher with the standard *Exponential Moving Average* (EMA), which underlies mean-teacher approaches.

**EMA (mean-teacher).** EMA maintains an exponentially decaying average:

$$\mathbf{W}_{\text{EMA}}^t \;=\; \rho\,\mathbf{W}_{\text{EMA}}^{t-1} + (1-\rho)\,\mathbf{W}_I^t, \qquad \rho \in (0,1), \;\; \mathbf{W}_{\text{EMA}}^0 = \mathbf{W}_I^0. \tag{15}$$

Unrolling this recursion gives a geometric kernel over *lag*:

$$\mathbf{W}_{\text{EMA}}^t \;=\; \rho^t\mathbf{W}_I^0 + (1-\rho)\sum_{k=1}^{t}\rho^{\,t-k}\,\mathbf{W}_I^k \;=\; \sum_{k=0}^{t}\underbrace{\omega_{k|t}^{\text{EMA}}}_{\text{depends on }t-k}\,\mathbf{W}_I^k,$$

with $\omega_{0|t}^{\text{EMA}} = \rho^t$, $\omega_{k|t}^{\text{EMA}} = (1-\rho)\rho^{\,t-k}$ for $k \geq 1$, and $\sum_{k=0}^{t}\omega_{k|t}^{\text{EMA}} = 1$. The kernel is *stationary in lag*: weights depend only on recency $t-k$.

**WMA (normalized-time kernel).** In contrast, WMA assigns weights via a *kernel over normalized time* $\tau_k = (k + c_1)/(T + c_2)$:

$$\mathbf{W}_{\text{WMA}}^t \;=\; \sum_{k=0}^{t}\underbrace{\omega_{k|t}^{\text{WMA}}}_{\propto\,\kappa(\tau_k)}\,\mathbf{W}_I^k, \qquad \omega_{k|t}^{\text{WMA}} = \frac{\alpha_k}{\sum_{j=0}^{t}\alpha_j}, \quad \alpha_k = \kappa(\tau_k).$$

Here the kernel is *position-aware* in absolute (normalized) time, not just lag. The symmetric Beta kernel ($\beta_1 = \beta_2$) permits simultaneous emphasis of *both* endpoints (early stability and late convergence), a pattern that is *not* attainable with any single-parameter EMA.

**Key differences.**

- **Shape control.** EMA imposes a monotone geometric decay from the present; WMA can be early-peaked, late-peaked, flat (uniform), bimodal (e.g., arcsine), etc.

- **Invariance to schedule granularity.** WMA weights are defined on normalized time: if the training is retimed or step granularity changes while preserving the path over $[0, 1]$, the kernel $\kappa$ need not be retuned. EMA depends on the absolute decay $\rho$ and typically requires retuning when $T$ or logging cadence changes.

- **Endpoint behavior.** With $\beta_1 = \beta_2 = \frac{1}{2}$ (arcsine), BMA places substantial weight near $k \approx 0$ and $k \approx t$, preserving early information *and* emphasizing late iterates; EMA cannot simultaneously upweight both ends.

- **Recovering classical averages.** Choosing $\kappa$ uniform (Beta$(1, 1)$) yields the simple running average (Polyak/Ruppert; SWA (Izmailov et al., 2018)). EMA cannot realize an exactly uniform window without time-varying $\rho_t$.

- **Online normalization.** Both EMA and WMA are online and convex at each step; WMA's $\omega_t = \alpha_t / \sum_{j \leq t}\alpha_j$ admits arbitrary nonnegative $\alpha_t$ induced by $\kappa$.

**Mean-teacher within the WMA recursion (exact recovery).** In SD–WMA (Definition C.4), the step weight is $\omega_t = \alpha_t / \sum_{j=0}^{t}\alpha_j$, which is generally time-varying. To *recover EMA exactly* with constant $\omega \equiv 1 - \rho$, choose any $\alpha_0 > 0$ and set, for $t \geq 1$,

$$\alpha_t \;=\; \frac{\omega}{(1-\omega)^t}\,\alpha_0 \qquad \Longleftrightarrow \qquad \alpha_t \;=\; \frac{1-\rho}{\rho^{\,t}}\,\alpha_0, \tag{16}$$

which yields $\omega_t \equiv \omega$ and makes the WMA recursion identical to equation 15. If one insists on $\alpha_t = \kappa(\tau_t)$ with $\tau_t = (t + c_1)/(T + c_2)$, EMA corresponds to an exponential kernel over normalized time, $\kappa(\tau) = C\,(1-\omega)^{-(T+c_2)\tau + c_1'}$, for suitable constants $C, c_1'$ (fixed per run), which reproduces $\omega_t \equiv \omega$ via equation 16.

**Practical guidance for kernel choice.**

- **Arcsine (Beta($\frac{1}{2}, \frac{1}{2}$)) kernel.** Strong endpoint emphasis: stabilizes early training and accelerates near-convergence. This is our default choice in POMP.

- **Uniform (Beta($1, 1$)) kernel.** Equivalent to a running average (Polyak/SWA), often strong for flat-minima exploration.

- **Early-peaked ($\beta_1 > \beta_2$).** Emphasizes performance near the start of training, giving more weight to the pretrained model parameters.

- **Late-peaked ($\beta_1 < \beta_2$).** Emphasizes performance near the end of training without discarding the early anchor.

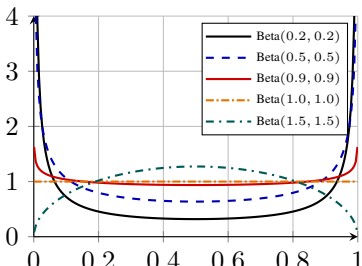

Figure 7: Beta PDFs with different parameters

The offsets $c_1, c_2 > 0$ function as finite baseline weight near the endpoints and remove singularities for $\beta_1, \beta_2 \leq 1$.

### C.5.2 THE PERSISTENT REGULARIZER OF THE WMA TEACHER

A key advantage of the WMA teacher over the more common EMA teacher lies in the dynamics of the regularization it provides. The self-distillation loss, $\mathcal{L}_{\text{SD}}$, induces a **regularizing gradient field**, $\mathbf{g}_R(\mathbf{W}_I^t) = \nabla_{\mathbf{W}_I} \mathcal{L}_{\text{SD}}(\mathbf{W}_T^t, \mathbf{W}_I^t)$, that pulls the student towards the teacher. The persistence of this field is critical for preventing the student from over-specializing on the finetuning task.

**The Vanishing Regularizer of EMA.** An EMA teacher is a low-pass filter of the student's trajectory: $\mathbf{W}_{\text{EMA}}^t = \rho \, \mathbf{W}_{\text{EMA}}^{t-1} + (1 - \rho) \, \mathbf{W}_I^t$. As the student's updates converge ($\|\mathbf{W}_I^{t+1} - \mathbf{W}_I^t\| \to 0$), the teacher necessarily converges to the student's final parameters ($\lim_{t\to\infty} \|\mathbf{W}_{\text{EMA}}^t - \mathbf{W}_I^t\| = 0$). Consequently, the regularizing gradient vanishes:

$$\lim_{t\to\infty} \|\mathbf{g}_R(\mathbf{W}_I^t; \mathbf{W}_{\text{EMA}}^t)\| = 0$$

This allows the optimization to be dominated entirely by the task loss $\mathcal{L}_{\text{MMCL}}$ at the end of training, risking overfitting.

**The Non-Vanishing Force of WMA.** The WMA teacher, by contrast, is a weighted average of the *entire* student history: $\mathbf{W}_{\text{WMA}}^t = \sum_{k=0}^{t} \omega_{k|t} \mathbf{W}_I^k$. Using a U-shaped kernel (e.g. Beta(0.5, 0.5)) ensures that the initial model $\mathbf{W}_I^0$ always contributes to the teacher with non-vanishing weight. As the student converges to a final state $\mathbf{W}_I^\infty \neq \mathbf{W}_I^0$, the teacher converges to a point $\mathbf{W}_{\text{WMA}}^\infty$ that is a convex combination of the entire path, and thus $\mathbf{W}_{\text{WMA}}^\infty \neq \mathbf{W}_I^\infty$.

*Theorem* C.6 (Non-Vanishing WMA Regularizing Gradient). Let the WMA teacher be constructed with a kernel that assigns non-zero weight to the initial time step (e.g., a Beta($\beta_1, \beta_2$) kernel with $\beta_1 \leq 1$). If the student finetuning trajectory moves from an initial state $\mathbf{W}_I^0$ to a convergent state $\mathbf{W}_I^\infty \neq \mathbf{W}_I^0$, the regularizing gradient induced by the WMA teacher converges to a persistent, non-zero vector:

$$\lim_{t\to\infty} \mathbf{g}_R(\mathbf{W}_I^t; \mathbf{W}_{\text{WMA}}^t) = \mathbf{g}_R^\infty \neq \mathbf{0}$$

*Proof Sketch.* 1. **Structure of the Gradient:** We assume the self-distillation loss $\mathcal{L}_{\text{SD}}$ (e.g., KL divergence between teacher and student outputs) behaves locally like a quadratic function of parameter difference. For small parameter differences $\|\mathbf{W}_T - \mathbf{W}_I\|$, the regularizing gradient can be approximated by:

$$\mathbf{g}_R(\mathbf{W}_I) = \nabla_{\mathbf{W}_I} \mathcal{L}_{\text{SD}}(\mathbf{W}_T, \mathbf{W}_I) \approx \mathbf{F}(\mathbf{W}_I)(\mathbf{W}_I - \mathbf{W}_T)$$

where $\mathbf{F}(\mathbf{W}_I)$ is the Fisher Information Matrix (FIM) at $\mathbf{W}_I$. The FIM is positive semi-definite and represents the curvature of the manifold of predictive distributions. The WMA regularizing gradient

is:

$$\mathbf{g}_R(\mathbf{W}_I^t) \approx \mathbf{F}(\mathbf{W}_I^t)(\mathbf{W}_I^t - \mathbf{W}_{\text{WMA}}^t)$$

$$= \mathbf{F}(\mathbf{W}_I^t)\left(\mathbf{W}_I^t - \sum_{k=0}^{t}\omega_{k|t}\mathbf{W}_I^k\right) \quad \text{(by Definition C.4)}$$

$$= \mathbf{F}(\mathbf{W}_I^t)\left(\left(\sum_{k=0}^{t}\omega_{k|t}\right)\mathbf{W}_I^t - \sum_{k=0}^{t}\omega_{k|t}\mathbf{W}_I^k\right) \quad \left(\text{since } \sum\omega_{k|t} = 1\right)$$

$$= \mathbf{F}(\mathbf{W}_I^t)\sum_{k=0}^{t}\omega_{k|t}(\mathbf{W}_I^t - \mathbf{W}_I^k)$$

2. **Asymptotic Behavior:** As $t \to \infty$, we assume the student converges to $\mathbf{W}_I^\infty$. The normalized time grid becomes dense in $[0,1]$ and the sum can be approximated by an integral. The asymptotic weighting density $\omega(\tau)$ is proportional to the chosen kernel $\kappa(\tau)$. The gradient converges to:

$$\mathbf{g}_R^\infty = \mathbf{F}(\mathbf{W}_I^\infty)\int_0^1 \omega(\tau)(\mathbf{W}_I^\infty - \mathbf{W}_I(\tau))d\tau$$

where $\mathbf{W}_I(\tau)$ is the continuous-time representation of the training trajectory.

3. **Non-Vanishing Property:** For a kernel like $\text{Beta}(\beta_1, \beta_2)$ with $\beta_1 \leq 1$, the weighting density $\omega(\tau)$ places non-zero mass near the start of the trajectory ($\tau \to 0$), where $\mathbf{W}_I(\tau) \approx \mathbf{W}_I^0$. Since finetuning changes the model, we have $\mathbf{W}_I^\infty \neq \mathbf{W}_I^0$. Therefore, the integrand $(\mathbf{W}_I^\infty - \mathbf{W}_I(\tau))$ is non-zero for a substantial portion of the integration domain, especially near $\tau = 0$. Given that $\mathbf{F}(\mathbf{W}_I^\infty)$ is positive semi-definite (and typically positive definite for non-degenerate models), the integral of a non-zero, positively weighted vector field results in a non-zero vector. Thus, $\mathbf{g}_R^\infty \neq \mathbf{0}$.

4. **Directionality of the Force:** The term $(\mathbf{W}_I^t - \mathbf{W}_I^k)$ points from a past iterate to the current one. The gradient $\mathbf{g}_R(\mathbf{W}_I^t)$ points in the direction of $(\mathbf{W}_{\text{WMA}}^t - \mathbf{W}_I^t)$. Since $\mathbf{W}_{\text{WMA}}^t$ is a weighted average of past iterates including $\mathbf{W}_I^0$, it effectively lies "behind" $\mathbf{W}_I^t$ on the trajectory. Therefore, the gradient vector $\mathbf{g}_R$ exerts a "restoring force" that is anti-parallel to the overall finetuning direction $(\mathbf{W}_I^t - \mathbf{W}_I^0)$, continuously pulling the student back towards the robust features of its initial, more general model. $\square$

**Conclusion.** The final solution $\mathbf{W}^*$ is a stationary point where the task-specific gradient is balanced by this persistent regularizing force: $\nabla_{\mathbf{W}_I}\mathcal{L}_{\text{MMCL}}(\mathbf{W}^*) + \lambda_{\text{SD}}\mathbf{g}_R^\infty = \mathbf{0}$. Unlike the EMA case where $\mathbf{g}_R^\infty = \mathbf{0}$, the WMA-distilled solution is necessarily displaced from the pure task minimizer. It is forced to find a compromise in a region that retains the robust characteristics of its initialization, providing a theoretical basis for its improved OOD performance.

### C.5.3 Convergence Analysis

We first state the single-step solution and then derive global convergence in the task subspace.

*Proposition* C.7 (Single-Step Solution). Let $\mathbf{W}_{\text{FT}}^\star = \mathbf{Y}_{\text{FT}}\mathbf{X}_I^\top(\mathbf{X}_I\mathbf{X}_I^\top)^+$ be the minimum-norm solution for the direct finetuning task, and let $\mathcal{P}_I$ be the orthogonal projector onto $\text{range}(\mathbf{X}_I)$. The SD–WMA update at step $t$ yields

$$\mathbf{W}_I^t = \mathbf{W}_I^{t-1}(\mathbf{I} - \mathcal{P}_I) + \frac{\lambda}{1+\lambda}\mathbf{W}_{\text{Teacher}}^{t-1}\mathcal{P}_I + \frac{1}{1+\lambda}\mathbf{W}_{\text{FT}}^\star. \tag{17}$$

*Proof.* This proposition is immediate from applying Lemma C.1 to the objective in Definition C.3. The objective at step $t$ has the same structure as static self-distillation (analyzed in Theorem C.2), but with the pretrained weights $\mathbf{W}_I^0$ in the regularization term replaced by $\mathbf{W}_{\text{Teacher}}^{t-1}$, and the initialization for gradient descent being $\mathbf{W}_I^{t-1}$. Specifically, we find the minimizer of: $\min_{\mathbf{W}_I} \frac{1}{2}\|\mathbf{W}_I\mathbf{X}_I - \mathbf{Y}_{\text{FT}}\|_F^2 + \frac{\lambda}{2}\|\mathbf{W}_I\mathbf{X}_I - \mathbf{W}_{\text{Teacher}}^{t-1}\mathbf{X}_I\|_F^2$ This corresponds to the self-distillation case in Theorem C.2, where $\mathbf{W}_I^0$ is effectively replaced by $\mathbf{W}_{\text{Teacher}}^{t-1}$ for the purpose of defining the

fixed regularization target at this step. The solution form is then directly obtained by substituting $\mathbf{W}_{\text{Teacher}}^{t-1}$ for $\mathbf{W}_I^0$ in the $\mathbf{W}_{SD}$ formula, which yields:

$$\mathbf{W}_I^t = \mathbf{W}_I^{t-1}(\mathbf{I} - \mathcal{P}_I) + \frac{1}{1+\lambda}\mathbf{Y}_{\text{FT}}\mathbf{X}_I^\top(\mathbf{X}_I\mathbf{X}_I^\top)^+ + \frac{\lambda}{1+\lambda}\mathbf{W}_{\text{Teacher}}^{t-1}\mathcal{P}_I.$$

Recognizing $\mathbf{W}_{\text{FT}}^\star = \mathbf{Y}_{\text{FT}}\mathbf{X}_I^\top(\mathbf{X}_I\mathbf{X}_I^\top)^+$, we get the desired result. $\qquad\square$

The key advantage over static SD emerges from the teacher's evolution.

*Theorem* C.8 (Bias-Free Convergence in the Task Subspace). Let $a = \frac{\lambda}{1+\lambda}$ and define the teacher error $\mathbf{E}^t = (\mathbf{W}_{\text{Teacher}}^t - \mathbf{W}_{\text{FT}}^\star)\mathcal{P}_I$. Then for any online weights $\{\omega_t\}$ as in equation 14:

  (i) **Teacher contraction.** $\mathbf{E}^t = \left(1 - \frac{\omega_t}{1+\lambda}\right)\mathbf{E}^{t-1}$.

  (ii) **Student tracking.** $(\mathbf{W}_I^t - \mathbf{W}_{\text{FT}}^\star)\mathcal{P}_I = a\,\mathbf{E}^{t-1}$.

  (iii) **Convergence.** If $\sum_{t\geq 1}\omega_t = \infty$, then $\mathbf{W}_{\text{Teacher}}^t\mathcal{P}_I \to \mathbf{W}_{\text{FT}}^\star$ and $\mathbf{W}_I^t\mathcal{P}_I \to \mathbf{W}_{\text{FT}}^\star$.

*Proof.* Let $\mathbf{W}_{I,\|}^t = \mathbf{W}_I^t\mathcal{P}_I$ and $\mathbf{W}_{\text{Teacher},\|}^t = \mathbf{W}_{\text{Teacher}}^t\mathcal{P}_I$. From Proposition C.7, projecting onto the subspace $\text{range}(\mathbf{X}_I)$ gives:

$$\mathbf{W}_{I,\|}^t = \mathbf{W}_I^{t-1}(\mathbf{I} - \mathcal{P}_I)\mathcal{P}_I + \frac{\lambda}{1+\lambda}\,\mathbf{W}_{\text{Teacher}}^{t-1}\mathcal{P}_I + \frac{1}{1+\lambda}\,\mathbf{W}_{\text{FT}}^\star\mathcal{P}_I.$$

Since $(\mathbf{I} - \mathcal{P}_I)\mathcal{P}_I = \mathbf{0}$, and $\mathbf{W}_{\text{FT}}^\star$ is already in the parallel subspace (by definition), we have $\mathbf{W}_{\text{FT}}^\star\mathcal{P}_I = \mathbf{W}_{\text{FT}}^\star$. So,

$$\mathbf{W}_{I,\|}^t = a\,\mathbf{W}_{\text{Teacher},\|}^{t-1} + (1-a)\,\mathbf{W}_{\text{FT}}^\star, \qquad (18)$$

where $a = \frac{\lambda}{1+\lambda}$. (ii) Subtracting $\mathbf{W}_{\text{FT}}^\star$ from both sides of equation 18:

$$\mathbf{W}_{I,\|}^t - \mathbf{W}_{\text{FT}}^\star = a\,\mathbf{W}_{\text{Teacher},\|}^{t-1} + (1-a)\,\mathbf{W}_{\text{FT}}^\star - \mathbf{W}_{\text{FT}}^\star = a\,(\mathbf{W}_{\text{Teacher},\|}^{t-1} - \mathbf{W}_{\text{FT}}^\star) = a\,\mathbf{E}^{t-1}.$$

This proves part (ii).

(i) Now consider the teacher recursion (Definition C.4) projected onto $\mathcal{P}_I$:

$$\mathbf{W}_{\text{Teacher},\|}^t = (1 - \omega_t)\,\mathbf{W}_{\text{Teacher},\|}^{t-1} + \omega_t\,\mathbf{W}_{I,\|}^t.$$

Substitute equation 18 into this:

$$\mathbf{W}_{\text{Teacher},\|}^t = (1 - \omega_t)\,\mathbf{W}_{\text{Teacher},\|}^{t-1} + \omega_t\,(a\,\mathbf{W}_{\text{Teacher},\|}^{t-1} + (1-a)\,\mathbf{W}_{\text{FT}}^\star).$$

Rearranging terms to isolate $\mathbf{E}^t = \mathbf{W}_{\text{Teacher},\|}^t - \mathbf{W}_{\text{FT}}^\star$:

$$\begin{aligned}
\mathbf{W}_{\text{Teacher},\|}^t - \mathbf{W}_{\text{FT}}^\star &= (1-\omega_t)\,\mathbf{W}_{\text{Teacher},\|}^{t-1} + \omega_t\,a\,\mathbf{W}_{\text{Teacher},\|}^{t-1} + \omega_t\,(1-a)\,\mathbf{W}_{\text{FT}}^\star - \mathbf{W}_{\text{FT}}^\star \\
&= (1 - \omega_t + \omega_t a)\,\mathbf{W}_{\text{Teacher},\|}^{t-1} - (1 - \omega_t(1-a))\,\mathbf{W}_{\text{FT}}^\star \\
&= (1 - \omega_t(1-a))\,(\mathbf{W}_{\text{Teacher},\|}^{t-1} - \mathbf{W}_{\text{FT}}^\star).
\end{aligned}$$

Since $1 - a = 1 - \frac{\lambda}{1+\lambda} = \frac{1}{1+\lambda}$, we have:

$$\mathbf{E}^t = \left(1 - \frac{\omega_t}{1+\lambda}\right)\mathbf{E}^{t-1}.$$

This proves part (i).

(iii) Iterating the recurrence relation from part (i):

$$\|\mathbf{E}^t\|_F = \left(\prod_{k=1}^t \left(1 - \frac{\omega_k}{1+\lambda}\right)\right)\|\mathbf{E}^0\|_F.$$

For $\mathbf{E}^t$ to converge to 0, we need the product term to converge to 0. This occurs if and only if the sum $\sum_{k=1}^\infty \frac{\omega_k}{1+\lambda}$ diverges to $\infty$. Since $\lambda > 0$, $1 + \lambda$ is a finite constant. Thus, the condition

for convergence is $\sum_{k=1}^{\infty} \omega_k = \infty$. From Definition C.4, $\omega_t = \frac{\alpha_t}{\sum_{j=0}^{t} \alpha_j}$. If $\kappa(\tau_t)$ is a continuous function on $[0, 1]$ that is non-zero on a set of positive measure, then $\sum_k \alpha_k$ will diverge as $T \to \infty$ (assuming $t$ goes up to $T$), and thus $\sum_k \omega_k$ will diverge. For common kernels like Beta distributions (e.g., arcsine kernel), this condition holds. Since $\mathbf{E}^t \to \mathbf{0}$, we have $\mathbf{W}_{\text{Teacher}}^t \mathcal{P}_I \to \mathbf{W}_{\text{FT}}^{\star}$. From part (ii), as $\mathbf{E}^{t-1} \to \mathbf{0}$, it follows that $(\mathbf{W}_I^t - \mathbf{W}_{\text{FT}}^{\star})\mathcal{P}_I \to \mathbf{0}$, meaning $\mathbf{W}_I^t \mathcal{P}_I \to \mathbf{W}_{\text{FT}}^{\star}$. $\qquad\square$

*Corollary* C.9 (Linear rate under a bounded step weight). If $\omega_t \geq \omega_{\min} > 0$ for all $t \leq T$, then

$$\|(\mathbf{W}_{\text{Teacher}}^t - \mathbf{W}_{\text{FT}}^{\star})\mathcal{P}_I\|_F \;\leq\; \left(1 - \tfrac{\omega_{\min}}{1+\lambda}\right)^t \|(\mathbf{W}_{\text{Teacher}}^0 - \mathbf{W}_{\text{FT}}^{\star})\mathcal{P}_I\|_F.$$

Hence the training loss in the task subspace decays at least geometrically to the minimum, whereas static SD converges to a biased point for any fixed $\lambda > 0$.

*Proof.* This follows directly from Theorem C.8 part (i). If $\omega_t \geq \omega_{\min}$, then $1 - \frac{\omega_t}{1+\lambda} \leq 1 - \frac{\omega_{\min}}{1+\lambda}$. Since $0 < \omega_{\min} \leq 1$ and $\lambda > 0$, we have $0 < \frac{\omega_{\min}}{1+\lambda} < 1$, so $0 < 1 - \frac{\omega_{\min}}{1+\lambda} < 1$. Thus, the error contracts geometrically. Static SD, as derived in Theorem C.2, converges to a solution that is a convex combination of $\mathbf{W}_I^0 \mathcal{P}_I$ and $\mathbf{W}_{\text{FT}}^{\star}$. This is a biased point unless $\mathbf{W}_I^0 \mathcal{P}_I = \mathbf{W}_{\text{FT}}^{\star}$. $\qquad\square$

**Geometric Interpretation of Dynamic Self-Distillation.** We decompose the dynamics into orthogonal and parallel components with respect to $\text{range}(\mathbf{X}_I)$.

**Orthogonal Preservation.** Applying $(\mathbf{I} - \mathcal{P}_I)$ to Proposition C.7 and using the idempotency of projectors, $\mathcal{P}_I(\mathbf{I} - \mathcal{P}_I) = \mathbf{0}$, we get:

$$\mathbf{W}_I^t(\mathbf{I} - \mathcal{P}_I) \;=\; \mathbf{W}_I^{t-1}(\mathbf{I} - \mathcal{P}_I) \;=\; \cdots \;=\; \mathbf{W}_I^0(\mathbf{I} - \mathcal{P}_I),$$

This demonstrates that SD–WMA preserves pretrained knowledge orthogonal to the finetuning subspace, just like static self-distillation.

**Adaptive Task-Space Evolution.** Within the task subspace, the student update is given by:

$$\mathbf{W}_{I,\|}^t \;=\; \frac{\lambda}{1+\lambda}\, \mathbf{W}_{\text{Teacher},\|}^{t-1} \;+\; \frac{1}{1+\lambda}\, \mathbf{W}_{\text{FT}}^{\star}.$$

**Early training ($t$ small):** The teacher $\mathbf{W}_{\text{Teacher}}^{t-1}$ is still close to $\mathbf{W}_I^0$ (as $\omega_k$ for small $k$ is often high for U-shaped kernels, or simply because few updates have occurred). This means the teacher acts as a strong anchor, mitigating catastrophic forgetting during volatile updates.

**Late training ($t$ large):** As $t \to \infty$, Theorem C.8 shows that $\mathbf{W}_{\text{Teacher},\|}^{t-1}$ converges to $\mathbf{W}_{\text{FT}}^{\star}$. Substituting this into the student update:

$$\lim_{t\to\infty} \mathbf{W}_{I,\|}^t \;=\; \frac{\lambda}{1+\lambda}\, \mathbf{W}_{\text{FT}}^{\star} \;+\; \frac{1}{1+\lambda}\, \mathbf{W}_{\text{FT}}^{\star} \;=\; \mathbf{W}_{\text{FT}}^{\star}.$$

Thus, the dynamic teacher adapts, reducing anchor bias and enabling exact convergence to $\mathbf{W}_{\text{FT}}^{\star}$ in $\text{range}(\mathbf{X}_I)$.

*Proposition* C.10 (Dominance over Static SD). If $\|\mathbf{W}_{\text{Teacher},\|}^{t-1} - \mathbf{W}_{\text{FT}}^{\star}\|_F \leq \|\mathbf{W}_{I,\|}^0 - \mathbf{W}_{\text{FT}}^{\star}\|_F$, then for the same $\lambda$ the SD–WMA update attains lower squared error than static SD in the task subspace.

*Proof.* Let $\mathbf{W}_{\text{static SD}}^{\star}$ be the solution for static SD (from Theorem C.2). The squared error from $\mathbf{W}_{\text{FT}}^{\star}$ in the task subspace for static SD is proportional to $\|\frac{\lambda}{1+\lambda}\mathbf{W}_I^0 \mathcal{P}_I - \mathbf{W}_{\text{FT}}^{\star}\|_F^2$. For dynamic SD, the instantaneous target is proportional to $\|\frac{\lambda}{1+\lambda}\mathbf{W}_{\text{Teacher}}^{t-1} \mathcal{P}_I - \mathbf{W}_{\text{FT}}^{\star}\|_F^2$. If the teacher is closer to $\mathbf{W}_{\text{FT}}^{\star}$ in the parallel subspace than the initial model $\mathbf{W}_I^0$, i.e., $\|\mathbf{W}_{\text{Teacher},\|}^{t-1} - \mathbf{W}_{\text{FT}}^{\star}\|_F \leq \|\mathbf{W}_{I,\|}^0 - \mathbf{W}_{\text{FT}}^{\star}\|_F$, then the dynamic SD solution will be closer to $\mathbf{W}_{\text{FT}}^{\star}$ in that subspace, thus achieving lower error. The convergence result (Theorem C.8) guarantees that the teacher gets arbitrarily close to $\mathbf{W}_{\text{FT}}^{\star}$, eventually satisfying this condition. $\qquad\square$

## C.6 Distillation Loss Definitions in POMP

POMP employs a composite self-distillation loss $\mathcal{L}_{\text{SD-WMA}}$ from the WMA teacher, which consists of several complementary terms to transfer different aspects of knowledge. Let $\mathbf{T}$ denote the teacher model and $\mathbf{S}$ denote the student model. $\mathbf{h}_{I_i}^{\mathbf{T}}$ and $\mathbf{h}_{T_i}^{\mathbf{T}}$ are image and text embeddings from the teacher for the $i$-th example, and similarly for the student. $\tau$ denotes the temperature parameter.

**Feature Distillation (FD).** This loss directly minimizes the Mean Squared Error between the student's and teacher's embeddings for each corresponding image-text pair in a mini-batch of size $N$. It helps align the feature spaces.

$$\mathcal{L}_{\text{FD}} = \frac{1}{N} \sum_{i=1}^{N} \left( \left\| \mathbf{h}_{I_i}^{\mathbf{T}} - \mathbf{h}_{I_i}^{\mathbf{S}} \right\|_2^2 + \left\| \mathbf{h}_{T_i}^{\mathbf{T}} - \mathbf{h}_{T_i}^{\mathbf{S}} \right\|_2^2 \right) \tag{19}$$

**Contrastive Relational Distillation (CRD).** CRD aligns the student's contrastive similarity distribution with the teacher's. We first compute the image-to-text ($p$) and text-to-image ($q$) softmax distributions for both student and teacher across the mini-batch:

$$p_i^{\mathbf{T}}[j] = \frac{\exp(\mathbf{h}_{I_i}^{\mathbf{T}\top} \mathbf{h}_{T_j}^{\mathbf{T}}/\tau)}{\sum_{b=1}^{N} \exp(\mathbf{h}_{I_i}^{\mathbf{T}\top} \mathbf{h}_{T_b}^{\mathbf{T}}/\tau)}, \qquad p_i^{\mathbf{S}}[j] = \frac{\exp(\mathbf{h}_{I_i}^{\mathbf{S}\top} \mathbf{h}_{T_j}^{\mathbf{S}}/\tau)}{\sum_{b=1}^{N} \exp(\mathbf{h}_{I_i}^{\mathbf{S}\top} \mathbf{h}_{T_b}^{\mathbf{S}}/\tau)} \tag{20}$$

$$q_i^{\mathbf{T}}[j] = \frac{\exp(\mathbf{h}_{T_i}^{\mathbf{T}\top} \mathbf{h}_{I_j}^{\mathbf{T}}/\tau)}{\sum_{b=1}^{N} \exp(\mathbf{h}_{T_i}^{\mathbf{T}\top} \mathbf{h}_{I_b}^{\mathbf{T}}/\tau)}, \qquad q_i^{\mathbf{S}}[j] = \frac{\exp(\mathbf{h}_{T_i}^{\mathbf{S}\top} \mathbf{h}_{I_j}^{\mathbf{S}}/\tau)}{\sum_{b=1}^{N} \exp(\mathbf{h}_{T_i}^{\mathbf{S}\top} \mathbf{h}_{I_b}^{\mathbf{S}}/\tau)} \tag{21}$$

The distillation loss is the sum of the KL-divergences between these distributions, averaged over the batch.

$$\mathcal{L}_{\text{CRD}} = \frac{1}{N} \sum_{i=1}^{N} \left( D_{KL}(p_i^{\mathbf{T}} \| p_i^{\mathbf{S}}) + D_{KL}(q_i^{\mathbf{T}} \| q_i^{\mathbf{S}}) \right) \tag{22}$$

**Interactive Contrastive Learning (ICL).** ICL forces the student to learn within the teacher's embedding space by performing contrastive learning between the student's anchor embeddings and the teacher's key embeddings. The loss is a symmetric InfoNCE objective computed on these mixed-model pairs.

$$\mathcal{L}_{\text{ICL}} = -\frac{1}{2N} \sum_{i=1}^{N} \left( \log \frac{\exp(\mathbf{h}_{I_i}^{\mathbf{S}\top} \mathbf{h}_{T_i}^{\mathbf{T}}/\tau)}{\sum_{j=1}^{N} \exp(\mathbf{h}_{I_i}^{\mathbf{S}\top} \mathbf{h}_{T_j}^{\mathbf{T}}/\tau)} + \log \frac{\exp(\mathbf{h}_{T_i}^{\mathbf{S}\top} \mathbf{h}_{I_i}^{\mathbf{T}}/\tau)}{\sum_{j=1}^{N} \exp(\mathbf{h}_{T_i}^{\mathbf{S}\top} \mathbf{h}_{I_j}^{\mathbf{T}}/\tau)} \right) \tag{23}$$

**Cross Knowledge Distillation (Cross-KD).** This method acts as a hybrid of CRD and ICL. It aligns the student-to-teacher cross-modal similarity distribution with the teacher's self-modal distribution using KL-divergence. We define the student-to-teacher cross-modal distributions ($p^{\mathbf{S}\to\mathbf{T}}, q^{\mathbf{S}\to\mathbf{T}}$) as:

$$p_i^{\mathbf{S}\to\mathbf{T}}[j] = \frac{\exp(\mathbf{h}_{I_i}^{\mathbf{S}\top} \mathbf{h}_{T_j}^{\mathbf{T}}/\tau)}{\sum_{b=1}^{N} \exp(\mathbf{h}_{I_i}^{\mathbf{S}\top} \mathbf{h}_{T_b}^{\mathbf{T}}/\tau)} \tag{24}$$

$$q_i^{\mathbf{S}\to\mathbf{T}}[j] = \frac{\exp(\mathbf{h}_{T_i}^{\mathbf{S}\top} \mathbf{h}_{I_j}^{\mathbf{T}}/\tau)}{\sum_{b=1}^{N} \exp(\mathbf{h}_{T_i}^{\mathbf{S}\top} \mathbf{h}_{I_b}^{\mathbf{T}}/\tau)} \tag{25}$$

The loss then minimizes the divergence from these distributions to the teacher's own relational distributions, $p_i^{\mathbf{T}}$ and $q_i^{\mathbf{T}}$.

$$\mathcal{L}_{\text{CrossKD}} = \frac{1}{2N} \sum_{i=1}^{N} \left( D_{KL}(p_i^{\mathbf{T}} \| p_i^{\mathbf{S}\to\mathbf{T}}) + D_{KL}(q_i^{\mathbf{T}} \| q_i^{\mathbf{S}\to\mathbf{T}}) \right) \tag{26}$$

**Geometric bridge to composite distillation.** Our analysis decomposes learning into an orthogonal preservation term and an in-subspace mixing term (Equation 1). The composite distillation terms are chosen to preserve *structure* consistent with this geometry: (i) **FD** anchors pointwise embeddings, biasing updates toward the teacher component within $\text{range}(\mathbf{X}_I)$ while damping drift in orthogonal

directions; (ii) **CRD** aligns the teacher's batch-wise similarity *distributions*, preserving inter-example geometry (a probabilistic surrogate for preserving $\mathbf{S}=\mathbf{H}_I^\top \mathbf{H}_T$); (iii) **ICL** performs contrastive learning in the teacher's semantic space, encouraging the student to operate on the teacher's subspace and thus to mix along task-relevant directions; and (iv) **CrossKD** aligns cross-modal logits to transmit cross-modal relational structure that vanilla InfoNCE may underweight. Together with the **WMA** teacher, these terms operationalize the geometric principle at feature-, relation-, and cross-modal levels.

## C.7 Connection to Robustness via Inter-class Feature Sharing

The self-distillation approach, particularly with a dynamic WMA teacher, can be understood through the lens of recent theoretical work on multi-modal contrastive learning's robustness mechanisms. Xue et al. (2024) identify *inter-class feature sharing* as a key mechanism behind MMCL's superior robustness to distribution shift, where models learn to leverage information about features appearing across different classes to dissociate spurious correlations.

Building on the insight that self-distillation acts as instance-specific label smoothing (Zhang and Sabuncu, 2020), we argue that the self-distillation method provides a similar robustness benefit by acting as an **informed label smoothing mechanism** that preserves inter-class similarities learned during pretraining. To see this connection, recall the self-distillation solution from Theorem C.2:

$$\mathbf{W}_{SD} = \mathbf{W}_I^0 \left( \mathbf{I} - \frac{1}{1+\lambda}\mathcal{P}_I \right) + \frac{1}{1+\lambda}\mathbf{Y}_{\text{FT}}\mathbf{X}_I^\top (\mathbf{X}_I\mathbf{X}_I^\top)^+ \tag{27}$$

This solution exhibits three key properties that enhance robustness:

**Preservation of Cross-Class Knowledge.** The term $\mathbf{W}_I^0 \left( \mathbf{I} - \frac{1}{1+\lambda}\mathcal{P}_I \right)$ maintains the pretrained model's understanding of feature relationships across classes. Unlike direct finetuning which completely overwrites representations in the finetuning subspace, self-distillation retains a weighted contribution from the original cross-class feature covariances. This is analogous to how Xue et al. (2024) show that MMCL leverages features appearing in multiple contexts to learn their independence from class labels.

**Informed Smoothing via Pretrained Similarities.** By regularizing towards $\mathbf{W}_I^0\mathbf{X}_I$ rather than arbitrary targets, self-distillation performs label smoothing that is informed by the pretrained model's learned inter-class similarities. This extends the instance-specific label smoothing interpretation of Zhang and Sabuncu (2020) to the finetuning setting, where the smoothing is guided by pretrained knowledge. This regularization preserves the cross-covariance structure that Xue et al. (2024) identify as crucial for robustness—specifically, the covariance between features that appear independently across different classes.

**Robustness Through Feature Independence.** Within the finetuning subspace, self-distillation computes a convex combination:

$$\frac{\lambda}{1+\lambda}\left(\mathbf{W}_I^0\mathcal{P}_I\right) + \frac{1}{1+\lambda}\left(\mathbf{Y}_{\text{FT}}\mathbf{X}_I^\top(\mathbf{X}_I\mathbf{X}_I^\top)^+\right) \tag{28}$$

This combination maintains the pretrained understanding of feature independence while adapting to the new task. As Xue et al. (2024) demonstrate in their Data Model 2, when features can occur independently across classes (e.g., "trees without green leaves" appearing in non-tree classes), models that preserve these cross-class relationships achieve superior robustness. The self-distillation mechanism explicitly preserves these relationships through the weighted contribution of $\mathbf{W}_I^0\mathcal{P}_I$.

The hyperparameter $\lambda$ controls the strength of this inter-class knowledge preservation: larger values of $\lambda$ maintain more of the pretrained model's understanding of how features vary independently across different contexts, potentially enhancing robustness to distribution shift. This suggests that self-distillation's effectiveness stems not merely from preventing catastrophic forgetting, but from actively preserving the rich inter-class feature relationships that contribute to robustness—a mechanism that parallels the theoretical insights of Xue et al. (2024) on why MMCL achieves superior out-of-distribution generalization.

## D  POMP ALGORITHM

---

**Algorithm 1** POMP (Preserve-Orthogonal-Mix-Parallel) Finetuning

---

**Require:** Pretrained CLIP model $\theta_{\text{CLIP}}^0 = \{\mathcal{E}_{\text{Image}}^0, \mathcal{E}_{\text{Text}}^0\}$
**Require:** Finetuning dataset $\mathcal{D}_{\text{FT}} = \{(\mathbf{x}_I, \mathbf{x}_T)\}_{i=1}^N$
**Require:** Learning rate $\eta$, Weight decay $\delta$, Batch size $B$, Number of epochs $E$
**Require:** Distillation coefficient $\lambda_{\text{SD}}$
**Require:** WMA kernel $\kappa(\tau_k)$ (e.g., Beta$(\beta_1, \beta_2)$) and total steps $T_{\text{total}}$
**Require:** Temperature $\tau_{\text{NCE}}$ for InfoNCE losses
1: **Initialize Student Model:** $\theta_S \leftarrow \theta_{\text{CLIP}}^0$ (image encoder $\mathcal{E}_{\text{Image},S}$, text encoder $\mathcal{E}_{\text{Text},S}$)
2: **Initialize Teacher Model:** $\theta_T \leftarrow \text{copy}(\theta_S)$
3: **Initialize Optimizer:** $\text{Opt} \leftarrow \text{AdamW}(\theta_S.\text{parameters}(), \eta, \delta)$
4: **Initialize WMA state:** cumulative_alpha $\leftarrow 0$
5: global_step $\leftarrow 0$
6: **for** epoch $= 1$ to $E$ **do**
7:     **for** batch $= \{(\mathbf{x}_I, \mathbf{x}_T)\}_{i=1}^B$ in $\mathcal{D}_{\text{FT}}$ **do**
8:         global_step $\leftarrow$ global_step $+ 1$
                                              ▷ — Student Forward Pass —
9:         $\mathbf{h}_{I,S} \leftarrow \mathcal{E}_{\text{Image},S}(\mathbf{x}_I)$
10:       $\mathbf{h}_{T,S} \leftarrow \mathcal{E}_{\text{Text},S}(\mathbf{x}_T)$
11:       Normalize student embeddings: $\mathbf{h}_{I,S} \leftarrow \text{normalize}(\mathbf{h}_{I,S}), \mathbf{h}_{T,S} \leftarrow \text{normalize}(\mathbf{h}_{T,S})$
                 ▷ — Compute Multi-Modal Contrastive Loss ($\mathcal{L}_{\text{MMCL}}$) —
12:       $\text{logits}_{I\leftrightarrow T} \leftarrow \mathbf{h}_{I,S} \cdot \mathbf{h}_{T,S}^\top / \tau_{\text{NCE}}$
13:       $\mathcal{L}_{\text{MMCL}} \leftarrow \text{InfoNCE}(\text{logits}_{I\leftrightarrow T}) + \text{InfoNCE}(\text{logits}_{I\leftrightarrow T}^\top)$     ▷ Symmetric InfoNCE
                 ▷ — Teacher Forward Pass (with no gradient updates) —
14:       **with** torch.no_grad() :
15:       $\mathbf{h}_{I,T} \leftarrow \mathcal{E}_{\text{Image},T}(\mathbf{x}_I)$
16:       $\mathbf{h}_{T,T} \leftarrow \mathcal{E}_{\text{Text},T}(\mathbf{x}_T)$
17:       Normalize teacher embeddings: $\mathbf{h}_{I,T} \leftarrow \text{normalize}(\mathbf{h}_{I,T}), \mathbf{h}_{T,T} \leftarrow \text{normalize}(\mathbf{h}_{T,T})$
              ▷ — Compute Dynamic Self-Distillation Loss ($\mathcal{L}_{\text{SD-WMA}}$) —
18:       $\mathcal{L}_{\text{FD}} \leftarrow \frac{1}{B} \sum_{i=1}^B (\|\mathbf{h}_{I,T}[i] - \mathbf{h}_{I,S}[i]\|_2^2 + \|\mathbf{h}_{T,T}[i] - \mathbf{h}_{T,S}[i]\|_2^2)$
19:       $\mathcal{L}_{\text{CRD}} \leftarrow \text{KL}(\text{softmax}(\mathbf{h}_{I,T}\mathbf{h}_{T,T}^\top / \tau_{\text{NCE}}) \| \text{softmax}(\mathbf{h}_{I,S}\mathbf{h}_{T,S}^\top / \tau_{\text{NCE}}))$   ▷ + text-to-image
20:       $\mathcal{L}_{\text{ICL}} \leftarrow \text{InfoNCE}(\mathbf{h}_{I,S}, \mathbf{h}_{T,T}) + \text{InfoNCE}(\mathbf{h}_{T,S}, \mathbf{h}_{I,T})$
21:       $\mathcal{L}_{\text{CrossKD}} \leftarrow \text{KL}(\text{softmax}(\mathbf{h}_{I,T}\mathbf{h}_{T,T}^\top / \tau_{\text{NCE}}) \| \text{softmax}(\mathbf{h}_{I,S}\mathbf{h}_{T,T}^\top / \tau_{\text{NCE}}))$     ▷ +
    text-to-image
22:       $\mathcal{L}_{\text{SD-WMA}} \leftarrow \mathcal{L}_{\text{FD}} + \mathcal{L}_{\text{CRD}} + \mathcal{L}_{\text{ICL}} + \mathcal{L}_{\text{CrossKD}}$
                   ▷ — Total Loss and Optimization —
23:       $\mathcal{L}_{\text{Total}} \leftarrow \mathcal{L}_{\text{MMCL}} + \lambda_{\text{SD}} \cdot \mathcal{L}_{\text{SD-WMA}}$
24:       $\text{Opt.zero\_grad}()$
25:       $\mathcal{L}_{\text{Total}}.\text{backward}()$
26:       $\text{Opt.step}()$
                         ▷ — Update WMA Teacher —
27:       $\tau_{\text{current}} \leftarrow (\text{global\_step} + c_1)/(T_{\text{total}} + c_2)$     ▷ Normalized time
28:       $\alpha_{\text{current}} \leftarrow \kappa(\tau_{\text{current}})$
29:       cumulative_alpha $\leftarrow$ cumulative_alpha $+ \alpha_{\text{current}}$
30:       $\omega_{\text{current}} \leftarrow \alpha_{\text{current}}/\text{cumulative\_alpha}$
31:       **for** parameter $p_S$ in $\theta_S$ and $p_T$ in $\theta_T$ **do**
32:           $p_T \leftarrow (1 - \omega_{\text{current}}) \cdot p_T + \omega_{\text{current}} \cdot p_S$
33:       **end for**
34:     **end for**
35: **end for**
36: **return** $\theta_S$

---

## E  REPRODUCIBILITY DETAILS

To ensure full reproducibility, we detail our experimental setup, key hyperparameters, and implementation. All source code will be made publicly available.

### E.1  COMPUTATIONAL ENVIRONMENT

- **Operating System:** Linux kernel 5.14.0-427.42.1.el9_4.x86_64.
- **GPU Hardware:** NVIDIA H100 80GB HBM3.
- **NVIDIA Driver Version:** 550.144.03.
- **CUDA Version:** 12.4.
- **Python Version:** 3.10.4.
- **PyTorch Version:** 2.0.1+ (with CUDA support).

### E.2  IMPLEMENTATION AND TRAINING DETAILS

Our implementation extends the OpenAI CLIP framework.

- **Model Architectures:** We use pretrained CLIP models (ViT-B/16, ResNet50, ViT-L/14) from OpenAI's official `clip` library.
- **Total Loss:** $\mathcal{L}_{\text{POMP}} = \mathcal{L}_{\text{MMCL}} + \lambda_{\text{SD}}\,\mathcal{L}_{\text{SD-WMA}}$.
  - $\mathcal{L}_{\text{MMCL}}$: Symmetric InfoNCE loss, directly leveraging OpenAI CLIP's core loss implementation. Optional cross-Frobenius regularizer coefficient was set to 0.05.
  - $\mathcal{L}_{\text{SD-WMA}}$: A composite self-distillation loss. For POMP, this comprises Feature Distillation (FD), Contrastive Relational Distillation (CRD), Interactive Contrastive Learning (ICL), and Cross Knowledge Distillation (Cross-KD).
- **WMA Teacher:** A custom Weighted Moving Average (WMA) teacher implementation, whose weighting kernel is a Beta distribution with $\beta_1 = \beta_2 = 0.5$.

### E.3  KEY HYPERPARAMETERS

The following hyperparameters were used for POMP finetuning on ImageNet-1K:

- **Epochs:** 10.
- **Optimizer:** AdamW.
- **Learning Rate:** $1 \times 10^{-5}$.
- **Weight Decay:** 0.1.
- **Batch Size:** 512 (ViT-B/16, RN50), 224 (ViT-L/14).
- **Warmup Length:** 500 steps (cosine LR schedule).
- **Mixed Precision:** Enabled using `torch.amp.autocast` with `torch.bfloat16`.
- **Distillation Coefficient $\lambda_{\text{SD}}$:** 0.9.
- **WMA Beta Kernel Parameter:** 0.5 (for Beta(0.5,0.5) kernel, i.e., arcsine distribution).
- **Teacher Update Frequency:** 0 or 1 (update every step).

### E.4  DATA PROCESSING

Standard OpenAI CLIP image preprocessing was applied. Input images are sourced from ImageNet-1K, and finetuning captions from OpenAI class templates.

### E.5  CODE AVAILABILITY

The full codebase, will be made publicly available to facilitate direct reproduction.

## F    EXPERIMENTAL DETAILS FOR LAYER-WISE ANALYSIS

To empirically validate our theoretical claims regarding geometric preservation in deep non-linear networks, we conducted a layer-wise representational similarity analysis (results in main text Figure 4).

**Methodology.**    We extract feature maps from every layer of the CLIP ViT-B/16 image encoder (Patch Embeddings, Transformer Blocks 0–11, and the Final Projection) on the ImageNet validation set. We compare the internal representations of the finetuned models (Direct FT and POMP) against the original Pretrained model using two standard metrics:

- **Centered Kernel Alignment (CKA) (Kornblith et al., 2019):** Measures the similarity between two representational spaces, invariant to orthogonal transformation and isotropic scaling. We use the linear CKA variant.
- **Singular Vector Canonical Correlation Analysis (SVCCA) (Raghu et al., 2017):** Measures the correlation between the principal components of the activation matrices. We report the mean correlation coefficient of the top 20 singular vectors.

This analysis confirms that catastrophic forgetting in standard finetuning manifests as a significant distortion of high-level features in deeper layers, a phenomenon effectively mitigated by POMP's geometric regularization.

## G    EXTENDED COMPLEXITY ANALYSIS

This section provides the theoretical derivation for the computational complexity comparison between POMP and spectral regularization methods like CaRot (results in main text Table 5).

Let $B$ denote the batch size, $d$ the dimension of the projection layer, and $P$ the total number of model parameters.

**CaRot (Spectral Regularization).**    CaRot imposes an orthogonality constraint on the projection weights to regularize the singular values. Computing the regularization term $\|\mathbf{W}^\top \mathbf{W} - \mathbf{I}\|_F^2$ involves matrix multiplication of the projection layer weights $\mathbf{W} \in \mathbb{R}^{d \times d}$. This operation scales as $\mathcal{O}(d^3)$. For large vision-language models, the projection dimension $d$ can be significant, adding non-trivial computational cost per iteration.

**POMP (Batch-wise Distillation).**    POMP's composite distillation loss operates on the similarity matrices computed within the mini-batch.

- **Feature Distillation:** Element-wise MSE on embeddings: $\mathcal{O}(B \cdot d)$.
- **Relational/Cross-KD:** Operations on $B \times B$ similarity matrices (logits): $\mathcal{O}(B^2)$.
- **Teacher Update:** The WMA update is an element-wise weighted average of parameters, scaling as $\mathcal{O}(P)$. This is identical to the cost of a standard EMA update.

Since the batch size $B$ is generally of similar order or smaller than the projection dimension $d$, and crucially, $B^2 \ll d^3$ for typical values, the overhead of POMP is significantly lower than spectral methods.

## H AI USAGE CLARIFICATION

Large Language Models improved the manuscript's grammar and readability; all research design, analysis, and interpretation were conducted by the authors.

