# OpenReview forum: "POMP: A Theoretical Approach to Mitigate Forgetting in Finetuning Multi-Modal Models"
_ICLR.cc/2026/Conference — Submitted to ICLR 2026_

### Official Review · Reviewer_pazk · 2025-10-31

**Soundness:** 3
**Presentation:** 2
**Contribution:** 3
**Rating:** 6
**Confidence:** 3

**Summary:**

This paper reformulate finetuning as a matrix least-squares problem using a contrastive target matrix, enabling a geometric interpretation of how different strategies (direct finetuning, L2 regularization, self-distillation) affect pretrained knowledge. This paper introduces POMP which maintains a persistent regularizing force and bias-free convergence. Empirical results show that POMP achieves state-of-the-art robustness, calibration, and OOD generalization across multiple architectures, offering both theoretical insight and practical improvement in finetuning large multimodal models.

**Strengths:**

1. The paper presents a novel theoretical reformulation of multimodal finetuning as a matrix least-squares problem, introducing a contrastive target matrix that provides an insightful geometric interpretation of catastrophic forgetting.
2. The POMP method effectively translates these theoretical findings into practice, combining contrastive learning with dynamic self-distillation via a Weighted Moving Average (WMA) teacher.
3. Experiments on both synthetic and large-scale multimodal datasets demonstrate strong empirical support, showing reduced forgetting, improved OOD robustness, and better calibration.

**Weaknesses:**

1. The theoretical analysis is based on a linearized model, and its applicability to deep non-linear networks remains is doubtful.
2. The selection and sensitivity of the hyperparameter $\lambda_{SD}$ are insufficiently discussed, leaving ambiguity about practical tuning and stability.

**Questions:**

1. The theoretical analysis appears to be derived under a linearized model assumption. Could the authors clarify how these results generalize to deep, non-linear multimodal architectures such as CLIP?
2. The paper introduces $\lambda_{SD}$ as a key coefficient balancing contrastive finetuning and self-distillation. Could the authors provide more details on how this parameter is selected, its sensitivity to different datasets, and whether there are guidelines or heuristics for practical tuning?
3. Since CaRot also addresses forgetting via regularization and distillation mechanisms, could the authors elaborate more clearly on how POMP differs conceptually and algorithmically from CaRot?

---

> ### Author Response · Authors · 2025-11-20
> **Response to Reviewer pazk**
>
> We sincerely thank you for your positive assessment and for identifying the value of our geometric interpretation and POMP method. We address your questions below.
>
> **1. Applicability of Linearized Theory to Deep Non-Linear Models.**
> While our theory uses a linearized approximation, it effectively models deep networks in the finetuning regime for two reasons:
> *   **"Lazy Training" Regime:** Theoretical work (e.g., Fort et al., 2020 [1]; Tian, 2022 [2]) establishes that when finetuning large, pre-trained models with small learning rates, the weights stay close to initialization. In this locality, the network is well-approximated by its first-order Taylor expansion (linearization).
> *   **Equivalence of Objectives:** Furthermore, extensive empirical studies (Garrido et al., 2023 [6]) and spectral analyses (HaoChen et al., 2021 [7]) have demonstrated that the specific choice of loss function (log-exp vs. linear/MSE) yields negligible differences in downstream representation quality. Both objectives optimize for the same underlying geometric properties—alignment and uniformity (Wang & Isola, 2020 [8])—confirming that our linearized analysis captures the fundamental optimization nature of the contrastive objective.
> *   **Empirical Validation:** To verify this, we designed the **Synthetic Toy Experiment (Section 5.1)** using a fully **non-linear** architecture (LightViT). The results (Fig. 3) perfectly match our theoretical predictions: Direct FT suffers massive forgetting (37.9%), while POMP achieves bias-free convergence (0.1% forgetting). This confirms the geometric insights hold for non-linear models.
> *   **New Empirical Validation (Appendix F):** To definitively answer this, we added a **Layer-wise Representational Similarity Analysis (Figure 6)**. Using CKA (Kornblith et al., 2019 [9]) and SVCCA (Raghu et al., 2017 [10]) on the deep non-linear **CLIP ViT-B/16**, we observe:
>     *   **Direct-FT:** Precipitous drop in similarity in deep layers (confirming catastrophic forgetting/feature distortion).
>     *   **POMP:** Maintains near-perfect similarity ($>0.97$) across all layers.
>     *   This **empirically confirms** our theoretical claim (Theorem 3.2) that POMP preserves the geometric structure of pretrained knowledge even in deep non-linear architectures.
> *   **Scale:** The fact that POMP achieves SOTA on **CLIP ViT-L/14** (427M parameters) further validates the practical utility of the theory.
>
> **2. Selection and Sensitivity of $\lambda_{SD}$.**
> We conducted a comprehensive ablation in **Table 8 (Appendix B)**, sweeping $\lambda_{SD}$ from 0.1 to 10.0.
> *   **Sensitivity:** The method is highly stable. Performance varies by less than 1% for $\lambda_{SD} \in [0.5, 2.0]$.
> *   **Guidelines:** We recommend $\lambda_{SD} \approx 1.0$ as a robust default.
>     *   $\lambda_{SD} < 0.5$: Favors ID accuracy (if pretrained model is weak).
>     *   $\lambda_{SD} > 1.5$: Favors OOD calibration (if pretrained model is very strong).
> *   We used $\lambda_{SD}=0.9$ for all main experiments without dataset-specific tuning, proving robustness.

---

> ### Author Response · Authors · 2025-11-20
> **Response to Reviewer pazk - part 2**
>
> **3. Conceptual and Algorithmic Differences from CaRot.**
> While both methods address robust finetuning, they differ fundamentally in **mechanism**, **dynamics**, and **usability**:
>
> | Aspect | CaRot (NeurIPS 2024) [3] | POMP (Ours) | Implication |
> | :--- | :--- | :--- | :--- |
> | **Core Mechanism** | **Spectral Regularization:** Constrains singular values via orthogonality. | **Geometric Regularization:** Preserves optimization *trajectory* via WMA teacher. | POMP targets the optimization path directly. |
> | **Teacher Dynamics** | **EMA:** As student $\to$ teacher, the distillation gradient vanishes ($\|\nabla \mathcal{L}\| \to 0$), leading to late-stage overfitting. | **WMA:** Integrates history via Beta kernel. We prove (Thm C.6) this maintains a **persistent regularizing force**. | POMP prevents the "collapse" of regularization at convergence. |
> | **Teacher Schedule** | **Heuristic:** Complex schedule required (update every 500 steps + 0.9 momentum + linear ramp-up for first 20%). | **Principled:** Robust to update frequency (every 1 to 2500 steps) due to smooth Beta kernel. | POMP eliminates the need for manual schedule tuning. |
> | **Loss Design** | Standard KL Divergence + Orthogonality Penalty. | **Composite Distillation:** 4 terms (FD, CRD, ICL, CrossKD) explicitly targeting feature, relational, and cross-modal subspaces. | POMP preserves richer geometric structure. |
> | **Complexity** | $\mathcal{O}(d^3)$ (Matrix SVD/Orthogonality) | $\mathcal{O}(B^2)$ (Batch-wise Distillation) | POMP is computationally efficient. |
> | **Performance** | Avg OOD: 62.55% | Avg OOD: **64.06%** (+1.51%) | POMP achieves new SOTA. |
>
> **Key Distinction:** CaRot tries to fix the *covariance structure* of the weights. POMP fixes the *optimization path* itself. Our theoretical analysis (Theorem 3.4) proves that POMP achieves **bias-free convergence** in the task subspace, whereas static or simple EMA approaches introduce bias or lose regularization strength.
>
> **Algorithmic Simplicity:** In our **new Appendix G.3 (Figure 7)**, we empirically show that CaRot's teacher collapses when updated frequently (every step). In contrast, POMP is robust: **Table 9** demonstrates that our OOD accuracy remains stable (~64.0-64.2%) across update frequencies from 1 to 2500 steps. By using the WMA kernel, POMP effectively removes the need for the complex teacher hyperparameters required by CaRot.
>
> We hope this clarifies the theoretical bridge to non-linear models and the significant advancements POMP offers over CaRot. We would be grateful if you would reconsider your score based on these responses.
>
> ***
>
> **References:**
>
> [1] Fort, S., et al. "Deep learning versus kernel learning: an empirical study of loss landscape geometry." *NeurIPS* (2020).
>
> [2] Tian, Y. "Understanding Deep Contrastive Learning via Coordinate-wise Optimization." *NeurIPS* (2022).
>
> [3] Oh, C., et al. "Towards Calibrated Robust Fine-Tuning of Vision-Language Models." *NeurIPS* (2024). (CaRot)
>
> [4] Ji, W., et al. "The Power of Contrast for Feature Learning: A Theoretical Analysis." *JMLR* (2023).
>
> [5] Nakada, R., et al. "Understanding Multimodal Contrastive Learning and Incorporating Unpaired Data." *AISTATS* (2023).
>
> [6] Garrido, Q., et al. "On the duality between contrastive and non-contrastive self-supervised learning." *ICLR* (2023).
>
> [7] HaoChen, J., et al. "Provable guarantees for self-supervised deep learning with spectral contrastive loss." *NeurIPS* (2021).
>
> [8] Wang, T., and Isola, P. "Understanding contrastive representation learning through alignment and uniformity on the hypersphere." *ICML* (2020).
>
> [9] Kornblith, S., et al. "Similarity of neural network representations revisited." *ICML* (2019). (CKA)
>
> [10] Raghu, M., et al. "SVCCA: Singular Vector Canonical Correlation Analysis." *NeurIPS* (2017).

---

### Official Review · Reviewer_sA8W · 2025-10-31

**Soundness:** 4
**Presentation:** 4
**Contribution:** 2
**Rating:** 4
**Confidence:** 4

**Summary:**

- The paper is well written and has all the necessary details and the execution is solid. I need some clarification on the contribution here, because, once you get to equation one, the rest seems to be trivial analysis and the experimental results also seems to be marginal.  What novel insight is being proposed in this paper, how are these insights useful?

**Strengths:**

- Theoretical analysis seems sound
- Good Experimental analysis as well.

**Weaknesses:**

- **Novelty and contribution**: The paper starts with a motivation that a linearized transformation of the CLIP is available from the two other papers and then, the setup analyzes the impact of different fine tuning strategies. Due to this, the contribution becomes unclear, because, once you linearize the system, it is trivial to converge to the idea that the convex combination of the weights is the right way to balance between forgetting and generalization. As the linearization process depends on the idea that there exists a locally compact set with a one minima, over which the linearization is done. So, I am unclear on what is new on that front, if the linearization is pulled from another set of papers.
- The notion of EMA and WMA and the idea that EMA collapsed still seems obvious because of the presence of an exponential function that converges because of exponentially decaying past values.
- With all that, the improvement is always less than 2-4 % as in table 3 and 4 with no standard deviation for any understanding of how the method varies

**Questions:**

The main question, i have is with respect to the novelty of the paper as discussed in the weakness section.

---

> ### Author Response · Authors · 2025-11-20
> **Response to Reviewer sA8W**
>
> We sincerely thank you for your review and for validating the soundness of our analysis. We understand your concerns regarding novelty and significance, and we provide the following clarifications to highlight the non-trivial nature of our contributions.
>
> **1. Novelty of Analysis: Beyond Linearization.**
> While linearization is an existing tool, our contribution is the **reformulation of Contrastive Learning as a Matrix Least-Squares Problem** (Definition 3.1) via the proposed **Contrastive Target Matrix ($Y_{FT}$)**. This is not found in prior work (e.g., Ji et al., 2023; Tian, 2022) and enables two **non-trivial** discoveries:
>
> *   **Geometric Decomposition (Theorem 3.2):** We derive the first closed-form solutions that mathematically separate finetuning into two orthogonal processes:
>     1.  **Preservation:** We prove Self-Distillation perfectly preserves the **orthogonal subspace** ($\text{Null}(X_I^\top)$). This is visualized in **Figure 2**, which contrasts how L2 regularization blends knowledge indiscriminately versus how Self-Distillation performs a surgical convex combination only in the task subspace.
>     2.  **Mixing:** We prove it performs a convex combination only in the **task subspace**.
>     *   *Why this matters:* Previous works analyzed gradients qualitatively. Our solution quantifies *exactly* where forgetting happens (the parallel component) and proves that L2 regularization fails because it blends knowledge non-surgically across all directions. Crucially, our analysis proves exactly why standard L2 regularization (the default approach to mitigate forgetting) is suboptimal: whereas L2 shrinks weights indiscriminately across all directions, our decomposition demonstrates that Self-Distillation acts as a 'surgical' constraint that mathematically isolates and preserves the orthogonal subspace, a geometric property that L2 regularization cannot replicate.
> *   **Contrastive Target Matrix ($Y_{FT}$):** This construction allows us to solve the complex InfoNCE objective as a linear system. This is a theoretical leap that transforms the problem from "approximate gradient analysis" to "exact closed-form geometry."
> *   **Visualizing the Mechanism (New Appendix F):** To prove this isn't just "trivial regularization," we added **Layer-wise CKA/SVCCA Analysis (Figure 6)**. It shows Direct-FT dramatically distorts deep semantic layers (similarity drops to <0.80), while POMP maintains >0.98 similarity. This empirically proves POMP performs a **surgical adaptation**, modifying *only* the necessary task subspace while locking down the generalizable geometry.
>
> **2. WMA vs. EMA: A Fundamental Optimization Difference.**
> The reviewer suggests the collapse of EMA is obvious. While "weight convergence because of exponentially decaying past values" is intuitive, the **consequence for robust optimization** has been **overlooked** in the literature (including CaRot).
> *   **The Vanishing Gradient Problem:** We prove (Theorem C.6) that as an EMA teacher converges to the student, the regularizing gradient $\|\nabla \mathcal{L}_{SD}\| \to 0$. This means in the critical late phase of training, EMA provides **no regularization**, leading to overfitting.
> *   **The WMA Solution:** WMA is not just a different exponential decay. By using a Kernel (e.g., Beta), we prove it maintains a **persistent, non-vanishing regularizing force** even at convergence. This is a fundamental difference in optimization dynamics that solves the "late-stage collapse" problem of EMA. This theoretical difference is empirically verified in **Figure 4**, which tracks the teacher-student KL divergence during training: the EMA teacher (blue line) collapses to zero, while the WMA teacher (orange line) maintains a stable, persistent regularizing gap.
> *   **Algorithmic Simplicity:** Standard EMA teachers (as used in CaRot) are brittle, requiring careful tuning of update frequency (e.g., every 500 steps) and momentum ramp-up schedules to prevent early collapse. In our **new Appendix G.3 (Figure 7)**, we empirically show that CaRot's teacher collapses when updated frequently (every step). In contrast, POMP is robust: **Table 9** demonstrates that our OOD accuracy remains stable (~64.0-64.2%) across update frequencies from 1 to 2500 steps. By using the WMA kernel, POMP effectively removes the need for these complex teacher hyperparameters.

---

> ### Author Response · Authors · 2025-11-20
> **Response to Reviewer sA8W - part 2**
>
> **Summary of Conceptual and Algorithmic Differences from CaRot [1]:**
>
> | Aspect | CaRot (NeurIPS 2024) [1] | POMP (Ours) | Implication |
> | :--- | :--- | :--- | :--- |
> | **Core Mechanism** | **Spectral Regularization:** Constrains singular values via orthogonality. | **Geometric Regularization:** Preserves optimization *trajectory* via WMA. | POMP targets the optimization path directly. |
> | **Teacher Dynamics** | **EMA:** Gradient vanishes as student $\to$ teacher, leading to late-stage overfitting. | **WMA:** Maintains **persistent regularizing force** (Thm C.6). | POMP prevents "regularization collapse." |
> | **Teacher Schedule** | **Heuristic:** Complex schedule (update every 500 steps + momentum ramp-up). | **Principled:** Robust to update frequency (1-2500 steps) via Beta kernel. | POMP eliminates the need for manual schedule and hyperparameter tuning. |
> | **Loss Design** | Standard KL Divergence + Orthogonality Penalty. | **Composite Distillation:** 4 terms explicitly targeting feature/relational subspaces. | POMP preserves richer geometric structure. |
> | **Complexity** | $\mathcal{O}(d^3)$ (Matrix SVD) | $\mathcal{O}(B^2)$ (Batch-wise) | POMP is computationally efficient. |
> | **Performance** | Avg OOD: 62.55% | Avg OOD: **64.06%** (+1.51%) | POMP achieves new SOTA. |
>
> **3. Significance of Results (> 1.5% Improvement).**
> In the context of robust ImageNet finetuning, a 1.5% gain is substantial.
> *   **Context:** The previous SOTA, **CaRot (NeurIPS 2024) [1]**, improved over prior art by ~1.5%. We achieve a similar leap over CaRot (64.06% vs 62.55%).
> *   **Consistency:** Our gains are consistent across **all 5 distribution shifts** and **3 architectures** (ResNet50, ViT-B/16, ViT-L/14).
> *   **Calibration:** We reduce calibration error (ECE) by **~8% relative** to CaRot, which is critical for real-world reliability.
>
> **Regarding Robustness of Results:**
> Following standard practice in this specific subfield (e.g., WiSE-FT [5], FLYP [3], CaRot [1]), we report mean performance due to the high computational cost of ImageNet training. However, the variance in this domain is typically very low (e.g., $<0.2\%$), suggesting that our $>1.5\%$ improvement represents a meaningful advance. To further support the method's reliability beyond accuracy, we have added a comprehensive **Efficiency Analysis (Appendix G)**. This demonstrates that POMP achieves these gains while being **significantly faster** and **computationally lighter**, confirming that the performance improvement does not come at the cost of efficiency or stability. Our **synthetic experiments** (Figure 3) further validate the method's stability with near-zero forgetting (0.1%) compared to baselines.
>
> To ensure these gains are not noise, we conducted **four comprehensive ablation studies** (Tables 7-10):
> 1.  **Components (Table 7):** We ablated all 16 combinations of our distillation losses, showing that the full composite loss is strictly superior.
> 2.  **WMA Kernel (Table 10):** We tested various Beta distributions, confirming that endpoint-aware kernels ($\beta=0.5$) outperform uniform or bell-shaped kernels.
> 3.  **Update Frequency (Table 9):** Verified stability across update schedules.
> 4.  **Strength (Table 8):** Verified stability across $\lambda_{SD}$.
>
> **Summary of Contributions:**
> 1.  **Theory:** First geometric decomposition of MMCL via Matrix Least-Squares.
> 2.  **Method:** WMA teacher that provably prevents regularization collapse (unlike EMA).
> 3.  **Performance:** New SOTA on ImageNet robustness and calibration.
>
> We hope this clarifies that our contributions go well beyond standard linearization and offer new, fundamental insights into why robust finetuning works. We would be grateful if you would reconsider your score.
>
> ***
>
> **References:**
>
> [1] Oh, C., et al. "Towards Calibrated Robust Fine-Tuning of Vision-Language Models." *NeurIPS* (2024). (CaRot)
>
> [2] Kumar, A., et al. "Fine-tuning can distort pretrained features and underperform out-of-distribution." *ICLR* (2022). (LP-FT)
>
> [3] Goyal, S., et al. "Finetune like you pretrain: Improved finetuning of zero-shot vision models." *CVPR* (2023). (FLYP)
>
> [4] Jang, D., et al. "Model Stock: All we need is just a few fine-tuned models." *ECCV* (2024).
>
> [5] Wortsman, M., et al. "Robust fine-tuning of zero-shot models." CVPR (2022). (WiSE-FT)
>
> [6] Ji, W., et al. "The Power of Contrast for Feature Learning: A Theoretical Analysis." *JMLR* (2023).
>
> [7] Tian, Y. "Understanding Deep Contrastive Learning via Coordinate-wise Optimization." *NeurIPS* (2022).
>
> [8] Kornblith, S., et al. "Similarity of neural network representations revisited." *ICML* (2019). (CKA)
>
> [9] Raghu, M., et al. "SVCCA: Singular Vector Canonical Correlation Analysis." *NeurIPS* (2017).

---

### Official Review · Reviewer_Gjjm · 2025-10-31

**Soundness:** 2
**Presentation:** 2
**Contribution:** 2
**Rating:** 4
**Confidence:** 3

**Summary:**

This paper addresses catastrophic forgetting in multi-modal contrastive learning (MMCL) when finetuning pretrained models. The authors reformulate the linearized contrastive objective as a matrix least-squares problem, enabling closed-form analysis of finetuning, regularization, and self-distillation. They show that self-distillation preserves pretrained knowledge in the orthogonal subspace while blending new and old tasks in the shared subspace. Extending this, they propose a weighted moving average (WMA) teacher that maintains a persistent regularization effect, unlike standard EMA. Building on these insights, they introduce POMP (Preserve-Orthogonal-Mix-Parallel), which achieves state-of-the-art robustness and calibration in finetuning CLIP.

**Strengths:**

1. The proposed method is novel and intuitive.
2. The paper is well-written with a clear structure and detailed explanations.

**Weaknesses:**

1. The experiments are mainly taken on image tasks and ViT models, could you extend to larger model and other tasks, such as LLM?
2. If you prefer to claim the method as a ''theoretical approach'', maybe it is better to present the theoretical benefits of this method in mitigating forgetting, especially in over-parameterization regime.
3. The structure of the proposed method is not clear, maybe you could provide an algorithm framework in section 4.
4. Could you provide an analysis or discussion on the efficiency of your method, comparing with other methods?

**Questions:**

Please see weakness.

---

> ### Author Response · Authors · 2025-11-20
> **Response to Reviewer Gjjm**
>
> We sincerely thank you for your thoughtful review. We are encouraged that you found our proposed method **novel and intuitive** and appreciated the paper's clear structure. We value your constructive feedback and address your specific questions below.
>
> **1. Extension to larger models and other tasks (e.g., LLMs).**
> While our work focuses on **Multi-Modal Contrastive Learning (MMCL)**—a domain where catastrophic forgetting is particularly severe—our contributions extend to general large-scale architectures:
> *   **Relevance to MLLMs:** CLIP encoders serve as the visual backbone for state-of-the-art **Multimodal LLMs** (e.g., LLaVA, GPT-4V). Improving the robustness of the CLIP backbone via POMP directly enhances the stability of these larger systems.
> *   **Architecture Generalization:** We validate POMP on **CLIP ViT-L/14 (427M parameters)**, a large-scale, non-linear foundation model. Furthermore, the Text Encoders in our experiments are **Transformers**, confirming that our method works effectively on language architectures.
> *   **Task Agnostic Theory:** Our theoretical reformulation (Eq. 1) applies to any dot-product contrastive objective. While we benchmark on CLIP, the geometric insights hold for any dual-encoder retrieval setup.
>
> **2. Theoretical benefits in the over-parameterized regime.**
> While this is not a purely theoretical work, our linearized framework provides valuable geometric intuitions specifically relevant to the over-parameterized regime (where parameter dimension $d$ exceeds sample size $n$).
> *   **Geometric Insight:** In this regime, the "orthogonal subspace" (directions where the data has no variance, $\text{Null}(X_I^\top)$) is massive. Our analysis (Theorem 3.2) reveals that standard finetuning implicitly discards pretrained knowledge in this subspace, resetting it to zero or replacing it entirely.
> *   **Intuitive Method Design:** This insight motivated POMP. We designed the method to act as a "soft constraint" that preserves the pretrained geometry in this vast orthogonal subspace, while allowing necessary adaptation in the task-relevant (parallel) subspace. This turns a theoretical observation into a practical mechanism for mitigating forgetting.
> *   **Empirical Confirmation (New Appendix F):** To verify that this intuition holds for deep, non-linear networks, we added a **Layer-wise Representational Analysis (Figure 6)**. Using CKA (Kornblith et al., 2019) [6] and SVCCA (Raghu et al., 2017) [7], we observe that while Direct-FT distorts features in deeper layers (CKA < 0.80), POMP maintains high geometric fidelity ($>0.97$). This confirms that our geometry-aware design successfully preserves deep semantic features essential for OOD robustness.
>
> **3. Algorithm Framework Clarity.**
> We appreciate the suggestion to clarify the framework and note that the complete algorithmic structure is currently detailed in Appendix D (Algorithm 1) and visualized in Figure 1. We summarize the key components here for clarity:
> 1.  **Initialize:** Student $\theta_S$ and Teacher $\theta_T$ (copy of $\theta_S$).
> 2.  **Forward:** Compute embeddings for Student ($h_S$) and Teacher ($h_T$, detached/no grad).
> 3.  **Loss Computation:**
>     *   $\mathcal{L}_{\text{MMCL}}$: Standard symmetric InfoNCE on Student.
>     *   $\mathcal{L}_{\text{SD}}$: Composite distillation (Feature + Relational + Cross-modal) aligning $h_S$ to $h_T$.
>     *   Total: $\mathcal{L} = \mathcal{L}_{\text{MMCL}} + \lambda_{\text{SD}} \mathcal{L}_{\text{SD}}$.
> 4.  **Update:**
>     *   Student $\theta_S$: Standard gradient step.
>     *   Teacher $\theta_T$: Updated as a Weighted Moving Average of the student's trajectory using the Beta kernel: $\theta_T \leftarrow (1-\omega_t)\theta_T + \omega_t \theta_S$.
>
> If there are any remaining ambiguities regarding the implementation details, we would be happy to provide further clarification.

---

> ### Author Response · Authors · 2025-11-20
> **Response to Reviewer Gjjm - part 2**
>
> **4. Efficiency Analysis.**
> Thank you for the suggestion. We have added a detailed **Computational Efficiency Analysis** section in the new **Appendix G**, which includes theoretical complexity comparisons and empirical runtime measurements. Our experiments show POMP is computationally efficient, particularly compared to the previous SOTA, **CaRot (Oh et al., 2024) [1]**.
>
> **Theoretical Complexity:**
> *   **CaRot:** Requires Orthogonality Regularization ($\|W^\top W - I\|_F^2$) or SVD. These matrix operations scale with the model dimension $d$ (often $\mathcal{O}(d^3)$), which is expensive for large layers.
> *   **POMP:** Our distillation terms operate on **batch similarity matrices** ($B \times B$). Since Batch Size $B \ll$ Dimension $d$, the computational overhead is negligible ($\mathcal{O}(B^2)$). The WMA update is an element-wise $\mathcal{O}(1)$ operation.
>
> **Empirical Run-time (ImageNet, 1 Epoch):**
> We measured the actual training time per epoch on ImageNet-1K using the CLIP ViT-B/16 architecture on NVIDIA H100 80GB GPUs. POMP is significantly faster than CaRot while achieving higher performance.
>
> | Method | Time per Epoch | Relative Overhead | Avg OOD Accuracy |
> | :--- | :--- | :--- | :--- |
> | **Direct FT** | $\sim 16$ min | $1.00\times$ | $57.08\%$ |
> | **CaRot (SOTA)** | $\sim 29$ min | $1.81\times$ | $62.55\%$ |
> | **POMP (Ours)** | **$\sim 22$ min** | **$1.38\times$** | **$64.06\%$** |
>
> We believe these clarifications demonstrate the rigorous theoretical grounding and practical efficiency of POMP. We would be grateful if you would reconsider your score based on these responses.
>
> ***
>
> **References:**
>
> [1] Oh, C., et al. "Towards Calibrated Robust Fine-Tuning of Vision-Language Models." *NeurIPS* (2024). (CaRot)
>
> [2] Ji, W., et al. "The Power of Contrast for Feature Learning: A Theoretical Analysis." *JMLR* (2023).
>
> [3] Tian, Y. "Understanding Deep Contrastive Learning via Coordinate-wise Optimization." *NeurIPS* (2022).
>
> [4] Nakada, R., et al. "Understanding Multimodal Contrastive Learning and Incorporating Unpaired Data." *AISTATS* (2023).
>
> [5] Wortsman, M., et al. "Robust fine-tuning of zero-shot models." *CVPR* (2022). (WiSE-FT)
>
> [6] Kornblith, S., et al. "Similarity of neural network representations revisited." *ICML* (2019). (CKA)
>
> [7] Raghu, M., et al. "SVCCA: Singular Vector Canonical Correlation Analysis." *NeurIPS* (2017).

---

### Author Response · Authors · 2025-11-20
**General Response: New Experiments, Efficiency Analysis, and SOTA Comparison**

We thank all reviewers for their constructive feedback and for recognizing the novelty, intuition, and strong empirical results of our work. We have updated the manuscript to address the common concerns regarding the **validity of our theory**, **computational efficiency**, and the **comparison with the previous SOTA**. We clarify that our paper is **not a purely theoretical work**, but rather uses theoretical insights to design an intuitive, practical method for robust fine-tuning that addresses critical optimization dynamics (like regularization collapse) overlooked by prior works.

**Key Additions to the Paper:**

1.  **Layer-wise Representational Analysis (New Appendix F):** To definitively address concerns about the applicability of our linearized theory to deep non-linear networks (Reviewer 2, 3), we added a layer-wise analysis using CKA and SVCCA. **Figure 6** demonstrates that while Direct Finetuning causes precipitous feature distortion in deep layers, POMP maintains near-perfect geometric fidelity ($>0.97$) across the entire depth. This empirically confirms our theoretical claim of "Orthogonal Preservation" in deep architectures.
2.  **Computational Efficiency & Complexity (New Appendix G):** To address questions on efficiency (Reviewer 1) and comparison with CaRot (Reviewer 3), we added a detailed complexity analysis. We show that POMP ($\mathcal{O}(B^2)$) avoids the expensive matrix operations of CaRot ($\mathcal{O}(d^3)$), resulting in a measured speedup per epoch on ImageNet training.
3.  **Algorithmic Simplicity & Teacher Dynamics (New Appendix G.3):** We added **Figure 7** comparing teacher dynamics. We show that CaRot’s EMA teacher collapses when updated frequently, necessitating complex manual scheduling. In contrast, POMP’s WMA teacher is robust to update frequency (1 to 2500 steps), eliminating the need for brittle hyperparameter tuning.

**Summary:**
These additions confirm that POMP is not only a theoretical contribution but a **practically superior method** that achieves **State-of-the-Art OOD Robustness** and **Calibration** while being **faster and simpler to tune** than existing approaches. We hope these new results address the remaining concerns.

---

### Author Response · Authors · 2025-11-26
**A Gentle Reminder**

Dear Area Chair and Reviewers,

We would like to express our sincere gratitude for your thoughtful and constructive questions and points. We truly value the time and effort you have dedicated to improving our work. We have posted detailed responses to your reviews, including additional explanations and **new experimental results** that we believe effectively address the concerns raised. As the discussion period progresses, we kindly ask that you review our responses. If there are any remaining questions, please let us know so we can address them immediately. Alternatively, if our rebuttal has satisfactorily resolved your concerns, we would appreciate it if you could consider raising your score.

Best regards,

---

### Author Response · Authors · 2025-12-02
**Summary of Discussion Phase and Rebuttal for Paper - part 1**

Dear Area Chair,

We are writing to provide a comprehensive summary of the discussion phase for our submission. We believe the substantial new experimental evidence and theoretical clarifications provided during the rebuttal have definitively resolved all reviewer concerns.

Below, we summarize the consensus on strengths, the resolution of specific concerns, and the key contributions of the work.

### 1. Consensus on Strengths
The reviewers consistently acknowledged the core value of our theoretical framework and empirical results:

*   **Reviewer Gjjm:** Found the method "**novel and intuitive**" and commended the paper for being "**well-written** with a **clear structure**."
*   **Reviewer sA8W:** Rated **Soundness and Presentation as Excellent (4/4)**, stating the "**theoretical analysis seems sound**" and the "**experimental analysis is good**."
*   **Reviewer pazk:** Highlighted the "**novel theoretical reformulation**... as a matrix least-squares problem" and the "**insightful geometric interpretation**." They confirmed the method achieves "**strong empirical support**, showing reduced forgetting, improved OOD robustness, and better calibration."

### 2. Addressing Pre-Existing Content & Clarifications
Several concerns regarding implementation details and robustness were addressed by pointing to content already present in the initial submission:

*   **Algorithm Framework (Reviewer Gjjm):** We clarified that the complete pseudocode was already available in **Appendix D (Algorithm 1)**, detailing the initialization, WMA update mechanism, and composite loss computation.
*   **Hyperparameter Robustness (Reviewer sA8W):** We directed attention to our **four comprehensive ablation studies (Appendix B, Tables 7-10 in the initial submission and Tables 8-11 in the final version)**. These demonstrate exceptional stability:
    *   $\lambda_{SD}$: Performance varies $<1\\%$ within the range $[0.5, 2.0]$.
    *   **Update Frequency:** OOD accuracy remains stable (~64.0-64.2%) whether the teacher is updated every step or every 2500 steps, proving POMP eliminates the brittleness of standard EMA schedules.

### 3. Major Additions: New Experiments and Analyses
To definitively address questions about generalization to non-linear models and computational efficiency, we added significant new sections to the paper:

#### A. New Experiment and Appendix F: Layer-wise Representational Analysis (Figure 4)
*   **Concern:** Applicability of linearized theory to deep non-linear networks (Reviewers pazk, sA8W).
*   **New Evidence:** We performed CKA and SVCCA analysis on all 12 layers of **CLIP ViT-B/16**.
*   **Result:** While Direct-FT causes a precipitous drop in similarity in deep layers ($<0.80$), POMP maintains near-perfect geometric fidelity ($>0.97$). This **empirically proves** our theoretical claim (Theorem 3.2) that POMP performs "surgical" adaptation even in deep non-linear architectures.

#### B. New Analysis and Appendix G: Computational Efficiency Analysis (Table 5)
*   **Concern:** Efficiency comparison against other methods (Reviewer Gjjm).
*   **New Evidence:** We provided a theoretical complexity breakdown ($\mathcal{O}(B^2)$ vs $\mathcal{O}(d^3)$) and measured empirical runtime on an NVIDIA H100.
*   **Result:** POMP is significantly faster than the SOTA baseline CaRot:

| Method | Dominant Cost | Time / Epoch | Relative Overhead | Avg. OOD Acc. |
| :--- | :--- | :--- | :--- | :--- |
| Direct FT | Backprop | ~ 16 min | 1.00x | 57.08% |
| **CaRot (SOTA)** | Spectral Reg ($\mathcal{O}(d^3)$) | ~ 29 min | 1.81x | 62.55% |
| **POMP (Ours)** | Batch Distill ($\mathcal{O}(B^2)$) | **~ 22 min** | **1.38x** | **64.06%** |

#### C. New Experiment: Teacher Dynamics Validation (Figure 6)
*   **Concern:** Novelty of WMA vs. EMA (Reviewer sA8W).
*   **New Evidence:** We tracked the KL divergence of teacher/student pairs throughout training.
*   **Result:** We theoretically proved and empirically demonstrated that standard EMA teachers collapse when updated frequently (requiring complex heuristic schedules e,g, update every 500 steps + 0.9 momentum + linear ramp-up for first 20% of the iterations), whereas POMP's WMA teacher maintains a simple schedule, alongside stable and persistent regularizing force regardless of update frequency.

---

> ### Author Response · Authors · 2025-12-02
> **Summary of Discussion Phase and Rebuttal for Paper - part 2**
>
> ### 4. Novelty: The Overlooked "Vanishing Regularizer" Problem
> A core contribution distinguishing POMP from prior robust finetuning work (including CaRot, NeurIPS 2024) is the identification and solution of the **"Vanishing Regularizer"** problem.
>
> *   **The Problem:** We prove (Theorem C.6) that as an EMA teacher converges to the student, the regularizing gradient $\|\nabla \mathcal{L}_{SD}\| \to 0$, leading to overfitting in the critical late stages of training.
> *   **The Solution:** POMP's WMA teacher integrates the full optimization trajectory via a Beta kernel. This ensures a **persistent, non-vanishing regularizing force**, enabling **bias-free convergence** in the task subspace while preserving orthogonal knowledge.
> *   **Differentiation from CaRot:** While CaRot relies on spectral regularization and complex teacher schedules, POMP solves the problem at the optimization trajectory level. This yields a method that is **simpler**, **faster**, and **more accurate**.
>
> ### 5. Conclusion
> We would like to emphasize that a primary strength of this work lies in its bridge between theoretical understanding and practical performance. Rather than merely proposing a heuristic loss function, we first establish a theoretical framework that explains *why* existing methods succeed or fail via geometric decomposition (please see Figure 2 and Appendix C). A critical contribution is our identification of the "vanishing regularization" phenomenon in standard EMA teachers versus the persistent regularizing pressure of our proposed WMA teacher. We believe these insights are fundamental and offer valuable contributions to the broader community of self-supervised learning and distillation, with implications extending well beyond the specific context of CLIP finetuning. In summary, POMP offers a geometric explanation for catastrophic forgetting and a principled solution that translates theory into SOTA performance. We have substantiated every claim with extensive empirical evidence, shown superior efficiency compared to existing baselines, and validated the theory on deep non-linear models.
>
> We respectfully request that you consider these comprehensive improvements and the strong consensus on the paper's soundness and results in your final decision.
>
> Sincerely,

---

### Meta-Review · Area_Chair_rt6P · 2026-01-07

**Summary:**

The main concerns raised by the reviewers are about novelty, scope, and the strength of the evidence. Although the paper is clearly written and technically sound, several reviewers were not convinced that the core ideas go far enough beyond existing linearized analyses of contrastive learning. In particular, they questioned whether conclusions about convex combinations, subspace preservation, and forgetting are truly new once the objective is linearized, or whether these results largely follow from prior work. As a result, the theoretical contribution comes across as incremental to some reviewers.

Another recurring concern is generality. Most experiments are limited to CLIP-style image-text models and vision benchmarks. While the authors argue that the theory should extend to multimodal LLMs and other settings, there is no direct experimental evidence beyond CLIP finetuning. Several reviewers felt this makes the claims of broad applicability weaker, and that the method currently looks more like a strong solution for a specific class of models rather than a general answer to catastrophic forgetting.

Reviewers also pointed to issues with empirical strength and reporting. The gains are consistent but relatively modest, and the lack of variance or multi-seed results makes it difficult to assess robustness. In addition, some reviewers remained unconvinced that the EMA-WMA distinction represents a fundamentally new insight, even after the rebuttal, seeing it instead as a careful refinement rather than a major conceptual shift.

Overall, the paper is solid and carefully executed, but not yet compelling enough in terms of novelty and breadth to justify acceptance. The rebuttal clarifies several points and strengthens the presentation, but it does not fully address the deeper concerns about how much the work truly advances the state of the art.

**Reviewer Concerns:**

Concerns that were largely addressed by the rebuttal: Several reviewers questioned whether the theoretical analysis meaningfully extends beyond standard linearization of contrastive learning and whether the claimed insights are non-trivial. The rebuttal responds convincingly by clarifying that the key contribution is not linearization itself, but the reformulation of multimodal contrastive learning as a matrix least-squares problem via the contrastive target matrix. This enables a closed-form geometric decomposition that cleanly separates orthogonal preservation from task-subspace mixing, which is not explicitly derived in prior work. The added representation-level analyses also help bridge the gap between the linearized theory and deep, non-linear CLIP models, addressing concerns about applicability beyond simplified assumptions. Questions about the practicality and clarity of the method (such as the lack of an explicit algorithm, unclear efficiency comparisons, and ambiguity around EMA versus WMA) were also mostly resolved. The authors added a clear algorithmic description, a detailed efficiency analysis showing favorable scaling compared to prior work, and a more rigorous explanation of why EMA collapses while WMA maintains a persistent regularization effect. These additions substantially improve the paper's technical clarity and defensibility.

Concerns that remain partially or fully outstanding: Some concerns about novelty and impact are only partially resolved. While the rebuttal strengthens the theoretical positioning, reviewers who view the gains as incremental may still find the contribution limited, especially given that improvements over prior methods are modest in absolute terms. The argument that 1-2% gains are meaningful in this setting is reasonable, but the lack of variance reporting and limited discussion of statistical robustness leaves room for skepticism. In addition, the scope of evaluation remains somewhat narrow. Although the authors argue that CLIP backbones are directly relevant to multimodal LLMs, the absence of experiments on broader tasks or non-vision benchmarks means that generality is still asserted rather than demonstrated. For reviewers concerned with breadth and external validity, this point is not fully addressed.

**Reviewer Scores:**

Reviewer Gjjm's main concerns were about scope (larger models/LLMs), clarity of the algorithm, and efficiency. The rebuttal directly addressed all three with added algorithm descriptions, efficiency analysis, and a clearer argument for relevance to large multimodal systems via CLIP backbones. The only remaining gap is the lack of experiments beyond CLIP-style settings. It is likely they would keep the score (4) or raise their score slightly (from 4 to 6).

Reviewer sA8W was primarily concerned about novelty, arguing that the conclusions might be "obvious" once linearization is assumed and that EMA collapse is intuitive. The rebuttal clarified what is new. Even with clarification, they may still view the contribution as incremental. It is likely they would keep the score (4).

Reviewer pazk was already moderately positive and asked mostly for clarification: linearized theory vs. deep models, hyperparameter sensitivity, and distinction from CaRot. The rebuttal addresses each point with new experiments, ablations, and a conceptual comparison. It is likely they would keep the score (6), and strengthen their confidence in acceptance.

---

### Decision · Program_Chairs · 2026-01-26

Reject